# SnRK1-triggered switch of bZIP63 dimerization mediates the low-energy response in plants

Andrea Mair[1], Lorenzo Pedrotti[2], Bernhard Wurzinger[1], Dorothea Anrather[3], Andrea Simeunovic[1], Christoph Weiste[2], Concetta Valerio[4], Katrin Dietrich[2], Tobias Kirchler[5], Thomas Nägele[1], Jesús Vicente Carbajosa[6], Johannes Hanson[7,8], Elena Baena-González[4], Christina Chaban[5], Wolfram Weckwerth[1], Wolfgang Dröge-Laser[2], Markus Teige[1,9]*

[1]Department of Ecogenomics and Systems Biology, University of Vienna, Vienna, Austria; [2]Pharmaceutical Biology, Julius-von-Sachs-Institute, University of Würzburg, Würzburg, Germany; [3]Mass Spectrometry Facility, Max F. Perutz Laboratories, University of Vienna, Vienna, Austria; [4]Instituto Gulbenkian de Ciência, Oeiras, Portugal; [5]Department of Plant Physiology, Center for Plant Molecular Biology, University of Tübingen, Tübingen, Germany; [6]Centro de Biotecnología y Genómica de Plantas, Universidad Politécnica de Madrid, Madrid, Spain; [7]Department of Molecular Plant Physiology, Utrecht University, Utrecht, Netherlands; [8]Department of Plant Physiology, Umea Plant Science Center, Umeå University, Umea, Sweden; [9]Department of Applied Genetics and Cell Biology, University of Natural Resources and Life Sciences, Vienna, Austria

*For correspondence: markus. teige@univie.ac.at

Competing interests: The authors declare that no competing interests exist.

**Abstract** Metabolic adjustment to changing environmental conditions, particularly balancing of growth and defense responses, is crucial for all organisms to survive. The evolutionary conserved AMPK/Snf1/SnRK1 kinases are well-known metabolic master regulators in the low-energy response in animals, yeast and plants. They act at two different levels: by modulating the activity of key metabolic enzymes, and by massive transcriptional reprogramming. While the first part is well established, the latter function is only partially understood in animals and not at all in plants. Here we identified the *Arabidopsis* transcription factor bZIP63 as key regulator of the starvation response and direct target of the SnRK1 kinase. Phosphorylation of bZIP63 by SnRK1 changed its dimerization preference, thereby affecting target gene expression and ultimately primary metabolism. A *bzip63* knock-out mutant exhibited starvation-related phenotypes, which could be functionally complemented by wild type bZIP63, but not by a version harboring point mutations in the identified SnRK1 target sites.

## Introduction

Flexibility in the regulation of gene expression is crucial for all organisms to adjust their metabolism to changing growth conditions. Particularly under stress, available energy resources need to be balanced between defense and growth. The SUCROSE NON-FERMENTING RELATED KINASE 1 (SnRK1) in plants and its orthologs, the sucrose-non-fermenting 1 (Snf1) kinase in yeast and the AMP-dependent protein kinase (AMPK) in mammals, are well-known and crucial master regulators of energy homeostasis. SnRK1 is involved in the regulation of plant metabolism, development, and stress response (*Polge and Thomas, 2007*; *Baena-González and Sheen, 2008*), Snf1 is required for the

**eLife digest** Organisms need to adjust their metabolism in response to changing environmental conditions to ensure that they balance their energy intake with the demands of growth and reproduction. In plants, an enzyme called SnRK1 plays a crucial role in responses to starvation in two ways: by altering the activities of enzymes involved in metabolism and by regulating the expression of genes. To perform the second job, SnRK1 is thought to control the activity of proteins called transcription factors—which alter the expression of genes by binding to DNA—but it is not known which ones.

SnRK1 has 'kinase' activity, that is, it can alter the activities of other proteins by adding small molecules called phosphates to them. It has been suggested that a group of transcription factors called the bZIP proteins may be regulated by SnRK1. Two bZIP proteins work together to switch on a gene, and the combination of bZIP proteins that interact can influence which genes are switched on. Here, Mair et al. studied the role of a bZIP protein called bZIP63 during starvation in the plant *Arabidopsis*.

The experiments show that bZIP63 is involved in controlling responses to starvation. Furthermore, its activity is regulated by SnRK1, which adds phosphates to three specific locations on the protein. These phosphates alter the ability of bZIP63 to interact with other bZIP proteins, leading to changes in gene expression during starvation.

Mair et al. triggered starvation in *Arabidopsis* plants by keeping the plants in darkness for several days. The leaves of normal plants turn yellow in response to starvation, but the leaves of mutant plants that lacked bZIP63 remained green. In contrast, plants containing higher amounts of this bZIP protein showed the opposite effect and their leaves turned yellow much more quickly than normal plants. The mutant plants that lacked bZIP63 could be rescued by the normal protein, but not by another version of the protein that SnRK1 is unable to add phosphates to.

These data suggest that SnRK1 regulates bZIP63 activity to alter metabolism in response to starvation. Mair et al. propose a model in which the ability of bZIP63 to interact with other bZIPs is normally rather low. However, when the plants are starved, SnRK1 adds phosphates to bZIP63, which increases its ability to bind to other bZIP proteins and leads to changes in gene expression. The bZIP proteins are also found in animals; therefore a future challenge is to find out whether these proteins are also regulated in a similar way.

switch from fermentative to oxidative metabolism in the absence of glucose (*Hedbacker and Carlson, 2008*), and AMPK regulates glucose, lipid, and protein metabolism, mitochondrial biogenesis, and feeding behavior in animals (*Hardie et al., 2012*). They are generally activated under energy starvation conditions and trigger metabolic reprograming to slow down energy-consuming processes and turn on pathways for alternative energy production in order to survive the stress conditions (*Hardie, 2007*; *Tome et al., 2014*). This happens, in two ways: by direct phosphorylation and modulation of the activity of key enzymes in nitrogen, carbon, or fatty acid metabolism (*Sugden et al., 1999*; *Kulma et al., 2004*; *Harthill et al., 2006*), and by massive transcriptional reprogramming (*Polge and Thomas, 2007*; *Baena-González and Sheen, 2008*; *McGee and Hargreaves, 2008*). Especially in plants, the latter aspect, the regulation of transcription, is still poorly understood. In *Arabidopsis* protoplasts, transient overexpression of AKIN10, a catalytic subunit of the SnRK1 complex, resulted in a transcriptional profile reminiscent of various starvation conditions and led to the identification of 1021 putative SnRK1 target genes (*Baena-González et al., 2007*). However, the transcription factors mediating the transcriptional response of SnRK1 to energy starvation are still unknown. Based on reporter gene activation assays in protoplasts (*Baena-González et al., 2007*) and modelling of microarray data (*Usadel et al., 2008*), some members of the C/S1 group of basic leucine zipper (bZIP) transcription factors (TFs)—foremost bZIP11 and bZIP1 from the S1 group—were speculated to be involved in this process. Yet, a direct regulation of these bZIPs by SnRK1 has never been shown.

bZIP proteins form a large and highly conserved group of eukaryotic TFs. They bind the DNA as dimers and are characterized by a basic region for specific DNA binding and a leucine zipper for dimerization (*Deppmann et al., 2006*; *Reinke et al., 2013*). They are involved in a multitude of cellular

processes, including cell proliferation and differentiation, metabolism, stress response, and apoptosis (*Mayr and Montminy, 2001*; *Jakoby et al., 2002*; *Motohashi et al., 2002*; *Rodrigues-Pousada et al., 2010*; *Tsukada et al., 2011*). The diversity and flexibility of transcriptional regulation by bZIP TFs can at least partially be attributed to their potential to form variable dimer combinations, which bind to different consensus target sites (*Deppmann et al., 2006*; *Tsukada et al., 2011*). While the leucine zipper determines the possible dimer combinations (*Deppmann et al., 2006*; *Reinke et al., 2013*), the actual in vivo dimer composition is further influenced by factors such as protein availability, binding of regulatory proteins, or post-translational modifications (*Kim et al., 2007*; *Schuetze et al., 2008*; *Lee et al., 2010*). Since the initial discovery that the mammalian bZIP cAMP response element binding protein (CREB) is regulated by reversible phosphorylation, many bZIP TFs were reported to be phosphorylated (*Holmberg et al., 2002*; *Schuetze et al., 2008*; *Tsukada et al., 2011*). However, particularly in plants, the functional consequences of these phosphorylation events often remained unclear. For example, it has been known for several years that abscisic acid (ABA)-dependent phosphorylation of some ABA-responsive element binding proteins (AREBs) by SnRK2 kinases increases their transcriptional activity (*Furihata et al., 2006*), yet the underlying mechanism of this activation is still unknown. It is also surprising that, while many examples for phosphorylation-dependent regulation of bZIP activity, DNA-binding, subcellular localization, stability, and interaction with regulatory proteins are known (*Schuetze et al., 2008*; *Tsukada et al., 2011*), reports on the regulation of dimerization are scarce. So far, only three publications (*Kim et al., 2007*; *Guo et al., 2010*; *Lee et al., 2010*) showed compelling evidence for phosphorylation-dependent changes in bZIP dimerization in animals. Still, even in these cases it is often not entirely clear whether bZIP phosphorylation affects dimerization directly or indirectly by enhancing DNA binding.

bZIP63 is a member of the C-group of *Arabidopsis* bZIPs, which was proposed to play a role in energy metabolism, seed maturation, and germination under osmotic stress (*Jakoby et al., 2002*; *Correa et al., 2008*; *Veerabagu et al., 2014*). Its transcriptional profile indicates that bZIP63 could be involved in the (energy) starvation response, as transcription and mRNA stability are repressed by sugars and ABA and mRNA levels increase in the night and even more during extended night treatments (*Matiolli et al., 2011*; *Kunz et al., 2014*). A small set of potential target genes for bZIP63 has been identified, including genes involved in amino acid metabolism (*ASN1*/*DIN6* = ASPARAGINE SYNTHETASE 1, *ProDH* = PROLINE DEHYDROGENASE), energy starvation response (*DIN10* = RAFFINOSE SYNTHASE 6), and senescence (*SEN1* = SENESCENCE 1) (*Baena-González et al., 2007*; *Dietrich et al., 2011*; *Matiolli et al., 2011*; *Veerabagu et al., 2014*). The C-group bZIPs form a dimerization network with the S1-group in plants, in which bZIP63 can interact with all members (*Ehlert et al., 2006*; *Kang et al., 2010*). Three of its dimerization partners from the S1-group—bZIP1, bZIP11, and bZIP53—were shown to be important metabolic regulators, especially under energy starvation conditions, and to regulate the expression of *ASN1* and *ProDH* as well (*Hanson et al., 2008*; *Dietrich et al., 2011*; *Ma et al., 2011*). Furthermore, bZIP1 was recently confirmed as a transcriptional master regulator of the rapid response to nutrient signals controlling mainly genes involved in amino acid metabolism and cell death/phosphorus metabolism as primary targets (*Para et al., 2014*). In that study the authors also speculate about a posttranslational modification of bZIP1 or its binding partners.

Here we show that bZIP63 is an important metabolic regulator, especially under stress/starvation conditions, and that bZIP63 is phosphorylated at multiple sites in vivo in a sugar- and energy-dependent manner. In an unbiased approach, we identified SnRK1 as one of the kinases responsible for bZIP63 phosphorylation and found that it targets three highly conserved serine residues in the N- and C-terminus of bZIP63. Moreover, we demonstrate that the phosphorylation of these sites is crucial for bZIP63's dimerization and activity in planta and propose a molecular model for a phosphorylation-triggered switch of bZIP63 dimerization partners, which ultimately regulates metabolic reprogramming.

## Results

### bZIP63 controls dark-induced senescence and primary metabolism

To better understand the role of bZIP63 in the plant we first tested whether bZIP63 has a similar phenotype as its dimerization partners bZIP1 and bZIP11. Prolonged darkness was shown to induce

increased chlorophyll loss in plants overexpressing bZIP1 (*Dietrich et al., 2011*). Therefore, we incubated a bZIP63 knock-out (ko), two independent overexpressor (ox) lines (*Figure 1—figure supplement 1*), and their respective wild types (wt) in the dark and determined the percentage of the green leaf area as a measure for chlorophyll content (*Figure 1*; *Figure 1—figure supplement 2A*). This method was preferred over direct chlorophyll measurements (*Figure 1—figure supplement 2B*) as it is not affected by the water loss in senescing leaves. While no differences were observed before dark treatment, significant differences were visible after 9 days in darkness (*Figure1—figure supplement 2C–E*). Similar to the bZIP1 ox, the bZIP63 ox lines (ox#2 and ox#3) had a significantly

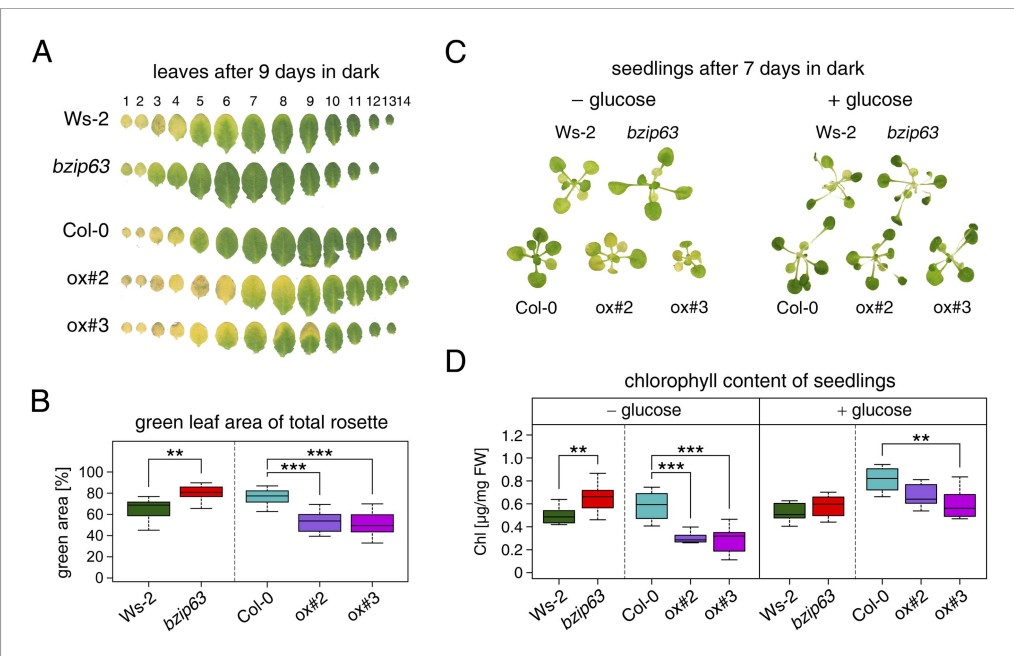

**Figure 1**. *bZIP63* mutants have a phenotype in dark-induced senescence. (**A**) and (**B**) Dark-induced senescence phenotype of 4.5 week-old soil grown plants. Comparison of a *bzip63* line in the Wassilewskya (Ws-2), and two bZIP63 ox lines in the Columbia (Col-0) background, after 9 days in darkness. (**A**) Representative leaf series. (**B**) Box-and-whiskers plot of the total green leaf area of eight biological replicates as determined with ImageJ. See *Figure 1—figure supplement 1* for molecular characterization of the bZIP63 mutant lines, *Figure 1—figure supplement 2* for a scheme of the determination of the green leaf area, quantitative chlorophyll measurements, controls and green area of individual leaves, *Figure1—source data 1* for the used ImageJ macro, and *Figure 1—figure supplement 3* for expression of senescence marker genes in wt and *bzip63*. (**C**) and (**D**) Sugar rescue of the dark-induced senescence phenotype. Seedlings were grown for 12 days on ½ MS agar containing 0.5% sucrose, transferred to ½ MS agar containing 0% or 2% glucose, and grown for another 6 days before incubation in the dark for 7 days. (**A**) Representative seedlings. (**B**) Box-and-whiskers plot of the chlorophyll content of 8 seedlings per line and condition. See *Figure 1—figure supplement 4* for pictures and chlorophyll content of seedlings before dark incubation. p-values from T-tests between mutants and wt < 0.05, < 0.01, and < 0.001 are indicated by *, **, and ***, respectively.

The following source data and figure supplements are available for figure 1:

**Source data 1**. ImageJ macro for determination of leaf area and green leaf area.

**Figure supplement 1**. Molecular characterization of the *bzip63* line and expression of bZIP63 in the ko and ox lines.

**Figure supplement 2**. Phenotype of *bZIP63* mutants in dark-induced senescence.

**Figure supplement 3**. Expression of senescence marker genes during prolonged darkness.

**Figure supplement 4**. Effect of sugar on bZIP63 mutants in light.

higher percentage of yellow leaf area than the wt. In contrast, *bzip63* plants displayed a stay-green phenotype (*Figure 1A,B*; *Figure1—figure supplement 2F*). RT-qPCRs of different senescence marker genes confirmed that the phenotype is due to dark-induced and not to natural senescence (*Figure 1—figure supplement 3*). Transcriptional responses in dark-induced senescence show clear similarities to starvation-induced senescence in cell suspension culture, which likely results from carbon depletion in both systems (*Buchanan-Wollaston et al., 2005*). We therefore tested whether addition of sugar could rescue this phenotype by performing the dark-induced senescence experiment with seedlings grown on agar plates with and without addition of sugar. Indeed, we found that the phenotype could be rescued by the addition of glucose (*Figure 1C,D Figure 1—figure supplement 4*), supporting the suggested role of bZIP63 in energy/carbon starvation response.

The notion that several hetero-dimerization partners of bZIP63, including bZIP1 and bZIP11, are important metabolic regulators under starvation conditions (*Dietrich et al., 2011*; *Ma et al., 2011*) prompted us to perform an unbiased metabolomics analysis of *bzip63* and ox#3 plants and their respective wt lines. Leaves of 5 week-old plants were harvested after 6 hr of light and extended night and analyzed for changes in the primary carbon and nitrogen metabolism using gas chromatography coupled to mass spectrometry (*Figure 2*; *Figure 2—figure supplement 1* and *Figure 2—source data 1*). Intriguingly, almost all amino acid levels were increased in the bZIP63 ko and decreased in the ox plants. This effect was even more pronounced after the extended night treatment. The biggest differences were observed for proline and the entire glutamate family. This accumulation of amino acids, particularly of proline, is striking in view of the observed senescence phenotype as it has been suggested that proline serves as an alternative energy source, especially under low carbon conditions (*Szabados and Savouré, 2010*; *Szal and Podgórska, 2012*).

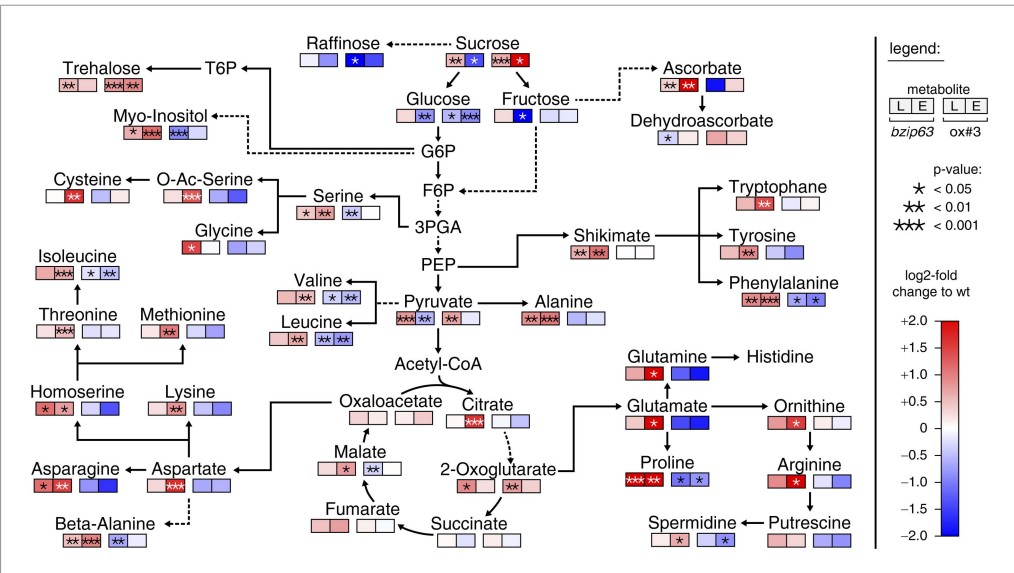

**Figure 2**. *bZIP63* mutants have an altered primary metabolism. Metabolic phenotype of 5 week-old soil grown plants after 6 hr light (L) and extended night (E). Log-2 fold changes of metabolite levels in ko and ox compared to their respective wt, displayed on a simplified map of the central primary metabolism. Values are means of five biological replicates. p-values from T-tests between mutants and wt < 0.05, < 0.01, and < 0.001 are indicated by *, ** and ***, respectively. For more details including mean values and SD see *Figure 2—figure supplement 1* and *Figure 2—source data 1*.

The following source data and figure supplement are available for figure 2:

**Source data 1**. Excel table of relative metabolite levels in *bZIP63* mutants and p-values from T-tests.

**Figure supplement 1**. Metabolic changes in *bZIP63* ko and ox plants.

# bZIP63 is phosphorylated at multiple sites in an energy-dependent manner

*Kirchler et al. (2010)* showed that bZIP63 can be phosphorylated in vitro by crude *Arabidopsis* extracts. To test whether bZIP63 is also phosphorylated in vivo, we treated total leaf protein extracts of ox#3 plants, expressing GFP-tagged bZIP63, with lambda protein phosphatase (λPP) and separated treated and untreated extracts on 2D gels by isoelectric focusing (IEF) in the first, and SDS-PAGE in the second dimension (*Figure 3A*). λPP treatment induced a clear shift of bZIP63 towards the basic region of the IEF strip, thus indicating dephosphorylation of the protein.

As bZIP63 expression is strongly regulated by the day/night cycle and by sugars (*Matiolli et al., 2011*; *Kunz et al., 2014*), we investigated its phosphorylation status under these conditions, applying the Phos-tag technique to enhance phosphorylation-induced mobility shifts in 1D SDS-PAGE (*Kinoshita et al., 2006*). Comparison of protein extracts from seedling cultures after 6 hr extended night in the presence or absence of sucrose or leaves harvested after 6 hr of light or extended night revealed strong differences in the phosphorylation patterns of bZIP63 (*Figure 3B*; *Figure 3—figure supplement 1*). Compared to the recombinantly expressed (not phosphorylated) bZIP63-YFP, the majority of plant-expressed bZIP63-GFP appeared in several slower migrating forms, thus indicating

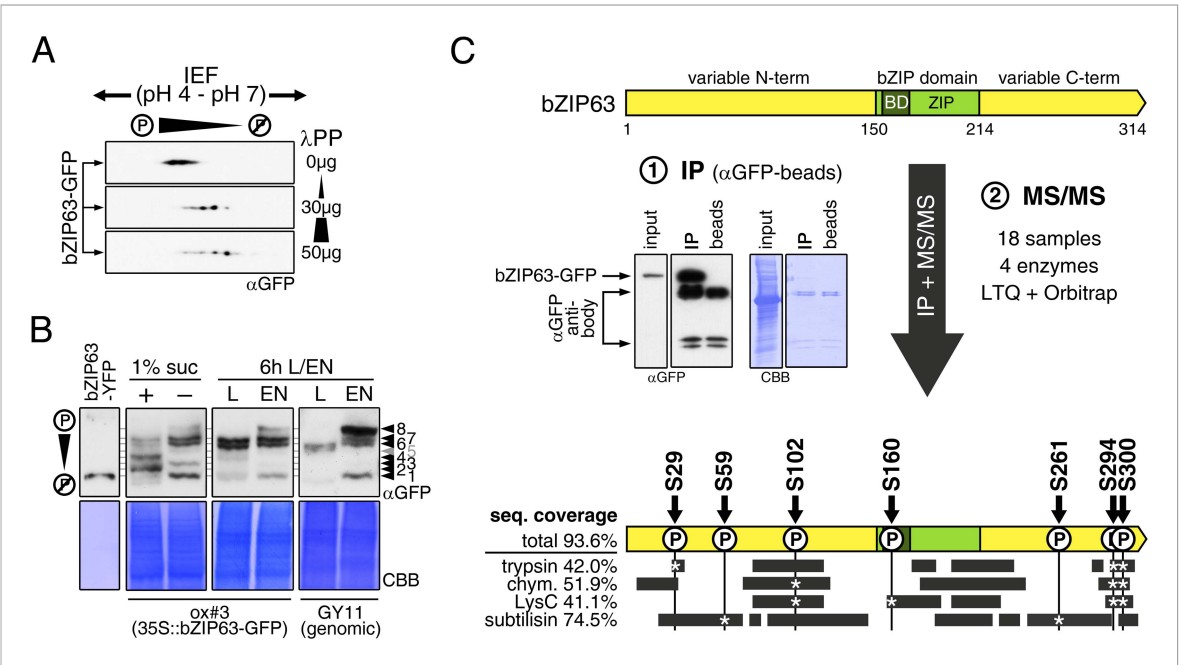

**Figure 3**. bZIP63 is phosphorylated at multiple sites in an energy-dependent manner in vivo. (**A**) 2D gel western blots (αGFP) of lambda phosphatase (λPP)-treated protein extracts from adult soil-grown plants expressing bZIP63-GFP (ox#3). (**B**) Phos-tag gel western blots showing the in vivo phosphorylation state of bZIP63 in seedlings after 6 hr extended night in the presence (+) or absence (−) of 1% sucrose and in 5 week-old soil-grown plants after 6 hr light (L) or extended night (EN). Plants either expressed 35S::bZIP63-GFP (ox#3) or a genomic fragment of bZIP63 (GY11) with a YFP-tag. Recombinant bZIP63-YFP was used as a nonphosphorylated control. Numbered arrowheads on the right mark the position of each observed bZIP63 band for easy reference with other figures (see *Figure 3—figure supplement 1* for a comparative image of all Phos-tag western blots). For more information on the genomic line see *Figure 7* and *Figure 7—figure supplement 1*. (**C**) Identification of in vivo phosphorylation sites by immunoprecipitation (IP) and tandem mass spectrometry (MS/MS). An exemplary western blot of the IP is shown. The scheme at the bottom shows the positions of the identified in vivo phosphorylation sites and the total sequence coverage reached with each proteolytic enzyme (grey bars). Asterisks mark the identification of a phospho-site. For more information on samples and (phospho-) peptides see *Figure 3—figure supplement 2*. IEF, isoelectric focusing; CBB, coomassie brilliant blue; BD, basic domain; ZIP, leucine zipper; LTQ, linear ion trap quadrupole; chym., chymotrypsin.

The following figure supplements are available for figure 3:

**Figure supplement 1**. Comparison of Phos-tag western blots from different figures.

**Figure supplement 2**. Overview over identified phospho-peptides of bZIP63.

multiple and differentially phosphorylated forms. In total, we were able to distinguish eight different forms on the Phos-tag gels, depending on the conditions (*Figure 3B*; *Figure 3—figure supplement 1*). However, it is important to note, that not every phosphorylation event results in an equal shift. In the light, two strong bands were visible, possibly reflecting the two spots on the 2D gel (*Figure 3B*, labelled as band 6 and band 7, respectively). Under extended night conditions, an additional band appeared (labelled as band 8), indicating increased phosphorylation of bZIP63. Notably, the ratio of the most phosphorylated form of bZIP63 (band 8) to the two less phosphorylated forms (band 6 and 7) was different between the ox (ox#3) and a genomic complementation line of *bzip63* (GY11, detailed description follows later). In the genomic complementation line, the most phosphorylated form of bZIP63 was the strongest band under extended night conditions, which might be a result of the different bZIP63 amounts in relation to the endogenous kinase(s), and likely better reflects the phosphorylation state of the endogenous protein. Interestingly, in seedlings grown in liquid culture, the extended night-triggered phosphorylation of bZIP63 could be abolished by the addition of 1% of sucrose (*Figure 3B*). In fact, phosphorylation was even lower here than in the light. These data indicate that bZIP63 has a basal level of phosphorylation in the plant and undergoes hyper-phosphorylation under starvation conditions. In contrast, addition of external sugars leads to a reduced phosphorylation status of bZIP63.

In order to identify the in vivo phosphorylated residues, bZIP63-GFP was immunoprecipitated from total leaf extracts using bead-coupled anti-GFP antibodies, subjected to proteolytic digest and analyzed by liquid chromatography coupled to tandem mass spectrometry (LC-MS/MS) (*Figure 3C*). To achieve maximum sequence coverage of the protein we combined proteolytic digests from four proteolytic enzymes (trypsin, chymotrypsin, LysC, and subtilisin). This approach resulted in a total sequence coverage of 93.6% and the identification of several phospho-peptides, indicating that bZIP63 is phosphorylated at up to seven serines (S29, S59, S102, S160, S261, S294, and S300) in vivo (*Figure 3C*; *Figure 3—figure supplement 2*). Two of these sites—S29 and S300—were also found in a recent phospho-proteomics study (*Umezawa et al., 2013*). Notably, only three of the seven sites—S29, S294, and S300—were found by tryptic protein digest, underpinning the advantage of alternative proteolytic digests for phospho-peptide identification in targeted proteomics.

## SnRK1, CDPKs and CKII are potential upstream kinases of bZIP63

To identify potential upstream kinases of bZIP63, we performed in-gel kinase assays with plant protein extracts from roots of hydroponically grown *Arabidopsis* plants or root cell culture using recombinantly expressed bZIP63 as substrate. Root material was chosen because in leaf extracts the large amount of Ribulose-1,5-bisphosphate carboxylase/oxygenase (RuBisCO) interfered with some of the kinase signals and poses a considerable problem during MS-based protein identification. Three strong bands of about 40, 50, and 55 kDa, respectively, were visible on the autoradiogram (*Figure 4A*), indicating that at least three kinases can phosphorylate bZIP63 in vitro. These bands were not visible on a gel without substrate, excluding the possibility that they originate from kinase auto-phosphorylation (*Figure 4—figure supplement 1*). To reduce the sample complexity and to enrich low abundant bZIP63-binding proteins before kinase identification, we affinity purified plant protein extracts on immobilized bZIP63. The eluted fractions were tested in an in-gel kinase assay for kinase activity towards bZIP63 and loaded on a normal SDS-PAGE gel without substrate for kinase identification. Bands corresponding in molecular weight (MW) to the signal from the in-gel kinase assay were excised from the gel, digested with trypsin and analyzed by LC-MS/MS (*Figure 4B,C*). In total, 27 protein kinases and kinase complex subunits were identified in four independent experiments (*Figure 4—source data 1*; *Figure 4—source data 2*). From those, proteins which did not match the expected MW or for which it had been shown experimentally that they are not localized in the nucleus were excluded. Only proteins which were identified in more than one sample with at least one proteotypic peptide were considered to be high confidence candidates, resulting in six protein kinases and two kinase regulatory subunits (*Figure 4C*; *Figure 4—source data 1*; *Figure 4—source data 2*). We identified the two main catalytic subunits of SnRK1, AKIN10 and AKIN11, as well as the regulatory subunit SNF4. Several members of the calcium dependent protein kinase (CDPK) family were found, but only CPK3 was identified with high confidence. In addition, two catalytic (CKA1 and CKA2) and one regulatory subunit (CKB1) of casein kinase II (CKII) were found, as well as Casein kinase 1-like protein 2 (CKL2). The SnRK1 kinases, CDPKs, and CKL2, correspond in MW

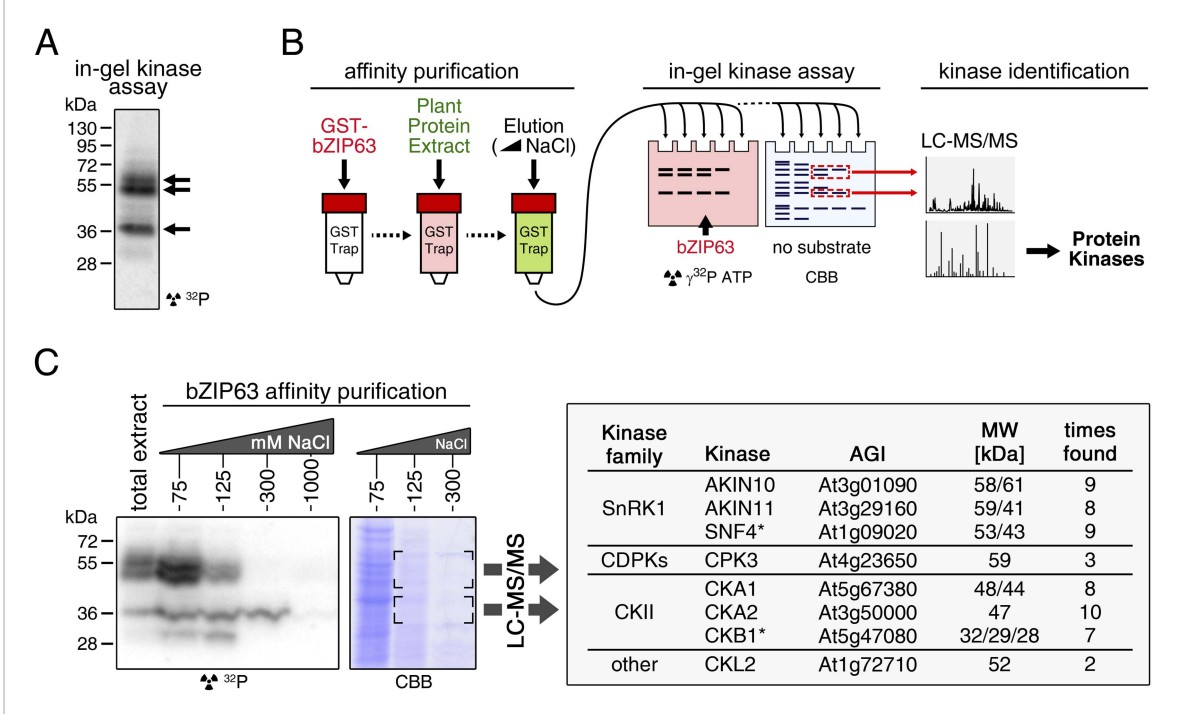

**Figure 4**. Several different kinases can phosphorylate bZIP63. (**A**) In-gel kinase assay with a root protein extract from hydroponically grown wild type plants and bZIP63 as substrate. Arrows indicate the positions of potential bZIP63 kinases. (**B**) Scheme of the kinase identification process. (**C**) In-gel kinase assay with samples from affinity purification of a root protein extract with bZIP63 and bZIP63 as a substrate (left) and a list of catalytic and regulatory (*) kinase subunits identified with high confidence (right). The list also contains the gene identifier (AGI), molecular weight (MW), and number of samples in which the protein was found. For controls and a list of all identified kinases and kinase peptides see *Figure 4—figure supplement 1* and *Figure 4—source data 1*, *2*. CBB, Coomassie brilliant blue.

The following source data and figure supplement are available for figure 4:

**Source data 1**. Overview over the kinases identified by LC-MS/MS after affinity purification with bZIP63.

**Source data 2**. Excel table containing a detailed overview over the identified kinases and analyzed samples as well as all peptides found for the kinase subunit.

**Figure supplement 1**. Auto-phosphorylation from the protein extracts is negligible in in-gel kinase assays.

to the two upper bands, while the lower band corresponds to the CKII kinase subunits. As SnRK1 was previously reported to enhance the activity of several C/S1 group bZIPs (*Baena-González et al., 2007*) and was suggested to be activated under energy starvation conditions (*Baena-González and Sheen, 2008*)—which could explain the observed hyper-phosphorylation of bZIP63 in extended night—we focused our further analysis on AKIN10 and AKIN11.

## The SnRK1 kinase AKIN10 interacts with and phosphorylates bZIP63 in vivo

To confirm that AKIN10 and AKIN11 phosphorylate bZIP63, we first performed an in-gel kinase assay with protein extracts of wt and *akin10* seedlings in the presence of EGTA, to reduce the signal from CDPKs of the same MW (*Figure 5A*). In both root and leaf extracts of *akin10* plants one band at the expected MW of AKIN10 nearly disappeared (see *Figure 5—figure supplements 1–3* for characterization of the *akin10* line). As AKIN11 has approximately the same MW as AKIN10, the remaining signal likely originates from AKIN11. In vitro kinase assays, with equal amounts of both kinases, showed that both kinases can phosphorylate bZIP63. However, AKIN10 phosphorylates bZIP63 much stronger than AKIN11 does (*Figure 5B*). Addition of the SnRK1 upstream kinase SnAK2

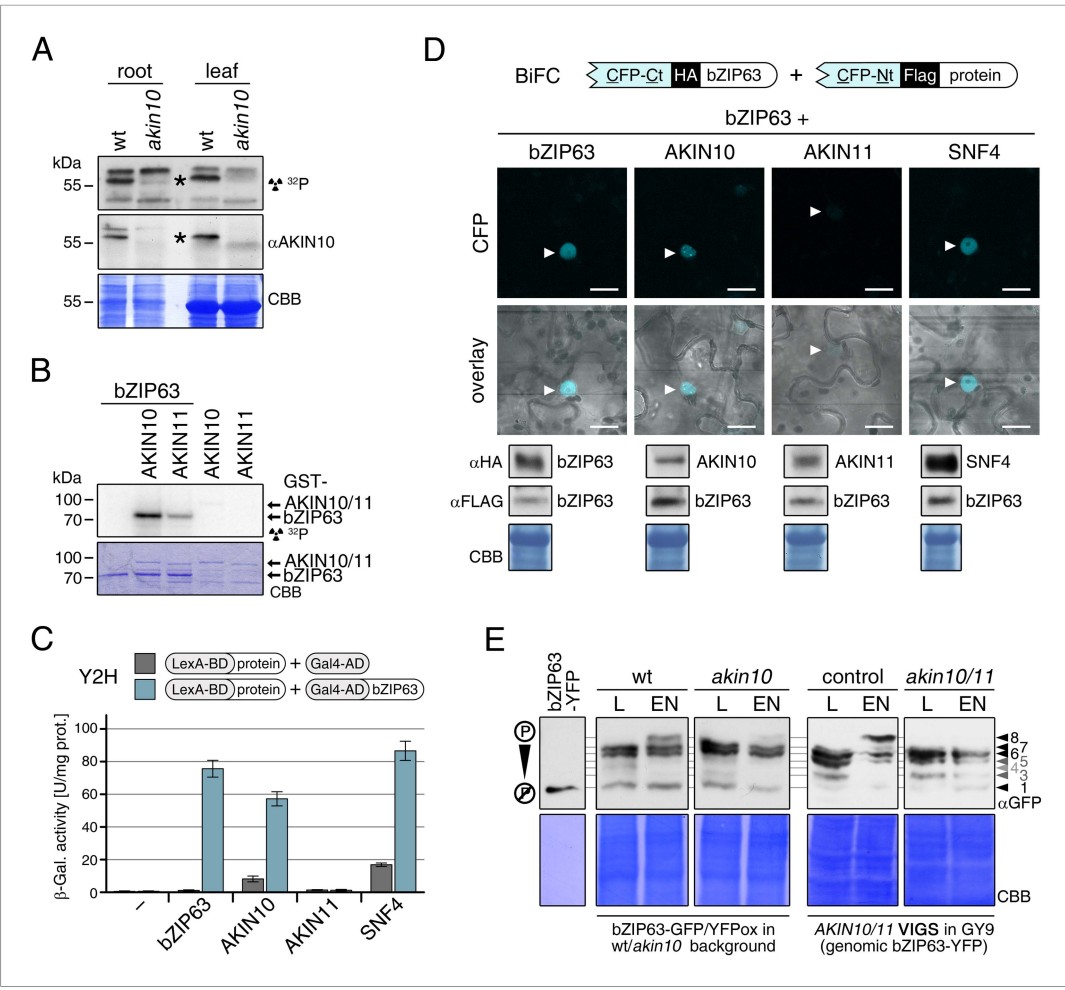

**Figure 5**. The SnRK1 kinase AKIN10 phosphorylates bZIP63 and interacts with bZIP63 in vivo. (**A**) In-gel kinase assay with protein extracts from wt and *akin10* plants and bZIP63 as a substrate (top), western blot against AKIN10 (αAKIN10, middle), and Coomassie brilliant blue stain (CBB, bottom). Asterisks mark the position of AKIN10. For characterization of the *akin10* line see *Figure 5—figure supplements 1–3*. (**B**) In vitro kinase assay with recombinant AKIN10/AKIN11 and bZIP63 as a substrate. See also *Figure 5—figure supplement 4* for kinase assays including the SnRK1 upstream kinase SnAK2. (**C**) and (**D**) Interaction of SnRK1 subunits with bZIP63. Homo-dimerization of bZIP63 was used as a positive control. (**C**) Yeast two-hybrid (Y2H) assay with auto-activation in grey and interaction with bZIP63 in blue. Bars represent means ± SD of eight biological replicates. For Y2H with AKINβ1 and β2 see *Figure 5—figure supplement 5*. (**D**) Laser scanning microscopy images of bimolecular fluorescence complementation (BiFC) in transiently transformed *Nicotiana tabacum* leaves (top). Arrowheads indicate the position of the nucleus. Size bar = 20 μm. Expression of the fusion proteins was verified by western blots (αHA, αFlag, bottom). (**E**) Phos-tag gel western blots showing the in vivo phosphorylation state of bZIP63 in 4–5 week-old soil grown plants after 6 hr light (L) or extended night (EN) in the presence and absence of AKIN10 alone or both AKIN10 and AKIN11. Plants overexpressing bZIP63-YFP in the *akin10* line were compared to plants overexpressing bZIP63-GFP in the wt background (ox#3) (left). Additionally, AKIN10 and AKIN11 were knocked down (*akin10/11*) by virus-induced gene silencing (VIGS) in plants expressing genomic bZIP63 with a YFP-tag (GY9 line). See *Figure 5—figure supplement 7* for images and a western blots of the VIGS plants. Recombinant bZIP63-YFP was used as a nonphosphorylated control. Numbered arrowheads on the right mark the position of each observed bZIP63 band for easy reference with other figures (see *Figure 3—figure supplement 1* for a comparative image of all Phos-tag western blots). A comparison of the phosphorylation state of bZIP63 in seedlings can be found in *Figure 5—figure supplement 6*.

The following figure supplements are available for figure 5:

**Figure supplement 1**. Molecular characterization of the *akin10* line.

*Figure 5. continued on next page*

*Figure 5. Continued*

**Figure supplement 2**. Phenotype and gene expression of selected AKIN10 target genes in the *akin10* line.

**Figure supplement 3**. Expression of bZIPs in the *akin10* line.

**Figure supplement 4**. SnAK2 increases the kinase activity of AKIN10 and AKIN11 but does not phosphorylate bZIP63.

**Figure supplement 5**. AKINβ1 and AKINβ2 do not interact with bZIP63.

**Figure supplement 6**. Altered sugar-dependent in vivo phosphorylation of bZIP63 in seedlings.

**Figure supplement 7**. *AKIN10/AKIN11* VIGS plants.

---

increased the activity of AKIN10 and AKIN11 about sevenfold (see also *Crozet et al., 2010*), but had no effect on the ratio between the signal intensities (*Figure 5—figure supplement 4*). In vivo interaction assays with the identified SnRK1 complex subunits supported the findings of the kinase assays. In both yeast two-hybrid (Y2H) (*Figure 5C*) and bimolecular fluorescence complementation (BiFC) assays (*Figure 5D*) AKIN10 and the regulatory subunit SNF4 interacted strongly with bZIP63, to a comparable level as measured for the bZIP63 homo-dimer, which was used as positive control (*Walter et al., 2004*). In contrast, AKIN11 and the two regulatory subunits AKINβ1 and AKINβ2 showed almost no interaction with bZIP63 (*Figure 5C,D*; *Figure 5—figure supplement 5*). However, it has to be considered that AKIN10 and AKIN11 are part of a trimeric complex including SNF4 (*Emanuelle et al., 2015*), which is neglected in these two assays. It is therefore still possible that AKIN11 interacts indirectly with bZIP63 via the regulatory subunit of the SnRK1 complex and plays a minor role in bZIP63 phosphorylation in the plant.

To verify that SnRK1 plays an important role in the in the in vivo phosphorylation of bZIP63 we compared the phosphorylation state of bZIP63 in plants overexpressing bZIP63-GFP or -YFP in the wt and *akin10* background, respectively. As SnRK1 has been suggested to act as a major regulator in the energy deprivation response, we again compared leaf protein extracts after 6 hr light and extended night and found that the hyper-phosphorylated form (band 8) of bZIP63, observed in the wt in extended night, was much weaker in the *akin10* background (*Figure 5E*). The same effect was observed in seedling cultures after 6 hr of extended night (*Figure 5—figure supplement 6*). To see whether the weak phosphorylation remaining in the *akin10* mutant would be further reduced in an *akin10/11* double mutant, we employed virus-induced gene silencing (VIGS) to knock down AKIN10 and AKIN11 in a genomic *bzip63* complementation line (GY9). Resulting plants showed strongly reduced growth and accumulation of anthocyanins (*Figure 5—figure supplement 7*), as previously described in *Baena-González et al. (2007)*. Like in the *akin10* line, there was no difference in bZIP63 phosphorylation between *akin10/11* and control plants in light, but the hyper-phosphorylated form of bZIP63 (band 8) in the extended night was now almost completely gone (*Figure 5E*). This shows clearly, that AKIN10 and—to a lower extent—AKIN11 are the major kinases responsible for bZIP63 hyper-phosphorylation under starvation conditions.

Additional evidence for in vivo phosphorylation of bZIP63 by SnRK1 emerges from a recent phosphoproteomics study in which one of the in vivo phosphorylation sites in bZIP63 (S300) was found to be more abundant in an AKIN10 ox line and less abundant in the ko line after extended night treatment, respectively (Nukarinen et al., unpublished).

## AKIN10 phosphorylates three conserved and functionally important serine residues in bZIP63

Next, to elucidate which of the seven in vivo phosphorylation sites can be phosphorylated by AKIN10, we performed in vitro kinase assays using the wt version of bZIP63 and different serine to alanine (S/A) mutants as substrates (*Figure 6A*; *Figure 6—figure supplements 1, 2*). Differences in phosphorylation were observed for proteins with mutations in S29, S294, and S300. In detail, the signal from S29A was strongly decreased in full length bZIP63 and completely gone in the N-terminal peptides that appeared as lower

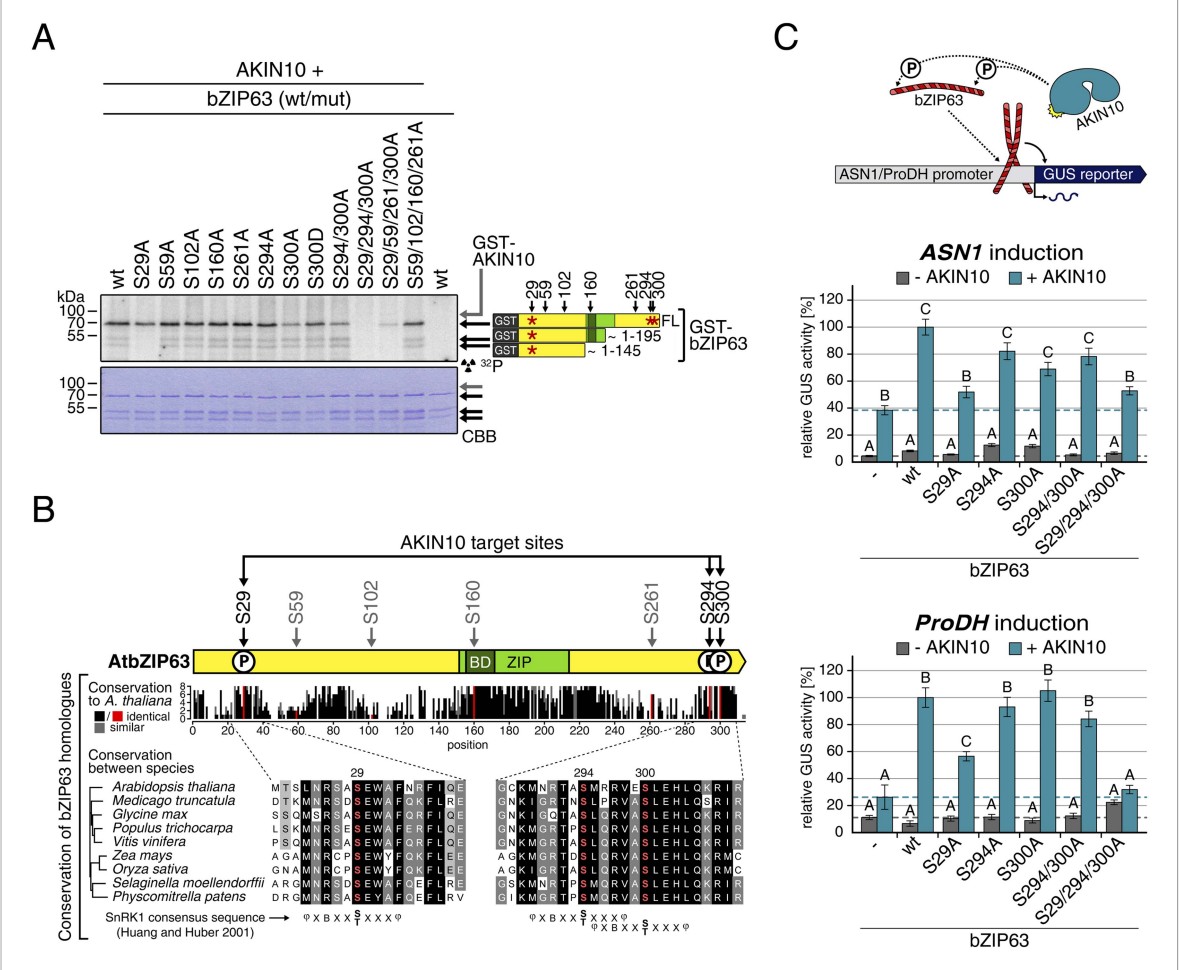

**Figure 6**. AKIN10 targets three highly conserved and functionally important serine residues in bZIP63. (**A**) In vitro kinase assay of wt and S/A mutants of GST-tagged bZIP63 with AKIN10. Positions of full length (FL) and N-terminal fragments of bZIP63 are marked by black arrows. The scheme on the right shows the position of the in vivo phosphorylation sites and the in vitro target sites of AKIN10 (red asterisk) on bZIP63. See *Figure 6—figure supplements 1, 2* for controls and Phos-tag gel of kinase assays, respectively. (**B**) Conservation of phosphorylation sites in bZIP63. Sequences of bZIP63 homologues from eight species were aligned with ClustalΩ. The scheme on top indicates the positions of the in vivo phosphorylation and AKIN10 target sites on bZIP63. The histogram below shows the sequence identity (red/black) and similarity (grey) to *A. thaliana* bZIP63. Red bars represent the in vivo phosphorylation sites. Below, the alignment of the sequence surrounding the AKIN10 target sites and the SnRK1 consensus motif (*Huang and Huber, 2001*) are shown. The grey/black shading indicates the degree of conservation, phosphorylation sites are in red. For alignment of non-AKIN10 target sites and full sequence alignment see *Figure 6—figure supplement 3*, for sequences in fasta format see *Figure 6—source data 1*. (**C**) Promoter activation assays in protoplasts with an ASN1/ProDH promoter-driven GUS reporter. Activation by bZIP63 wt and S/A mutants without (grey) or with (blue) co-transformation of AKIN10 is shown. Bars are means ± SD of 4 biological replicates, given in % of the activity of wt bZIP63 with AKIN10. Horizontal dashed lines indicate the signal in the control without bZIP63. Letters indicate significant differences as determined by ANOVA and pairwise T-testing (p < 0.05). See *Figure 6—figure supplement 4* for a western blot control. CBB, Coomassie brilliant blue.

The following source data and figure supplements are available for figure 6:

**Source data 1**. Sequences of the bZIP63 homologues.

**Figure supplement 1**. AKIN10 phosphorylates bZIP63 but not GST.

**Figure supplement 2**. AKIN10 phosphorylates S29, S294, and S300 on bZIP63.

**Figure supplement 3**. The AKIN10 target sites and S160 in the bZIP domain are highly conserved.

**Figure supplement 4**. Expression of bZIP63 and AKIN10 in the promoter activation assays.

MW degradation products in these assays. The S300A mutation led to an even stronger decrease of the signal, comparable to the S294/300A double mutant. Even though the S294A single mutant showed a similar signal as the wt, all three serines had to be mutated to completely abolish phosphorylation, suggesting that all of them are in vitro targets for AKIN10, with S294 being the weakest. Importantly, all three sites match the SnRK1 consensus sequence (*Huang and Huber, 2001*) (*Figure 6B*).

A comparison of *Arabidopsis* bZIP63 with orthologs from eight other plant species, ranging from mosses to higher plants, showed that the three putative AKIN10 target sites are highly conserved throughout evolution (*Figure 6B*; *Figure 6—figure supplement 3* and *Figure 6—source data 1*). As the origin of SnRK1 dates back even further, to the common ancestor of plants and animals (*Bayer et al., 2014*), it is likely that phosphorylation of bZIP63 by AKIN10 poses an ancient and important regulatory mechanism. We therefore set out to test the functional relevance of AKIN10-mediated phosphorylation of bZIP63. To this end, we tested the transcriptional activity of bZIP63 in protoplast-based promoter activation assays using the *ASN1* or *ProDH* promoter fused to the beta galactosidase (GUS) reporter (*Figure 6C*; *Figure 6—figure supplement 4*). In both cases, co-transformation of wt bZIP63 and AKIN10 strongly induced reporter gene expression, while transformation of bZIP63 alone was not sufficient for significant induction. Transformation of AKIN10 alone also led to a weak induction of the reporters, which could be explained by the action of endogenous bZIPs. Mutation of S29 or all three AKIN10 target sites on bZIP63 to alanine reduced the reporter activation almost to background level. In contrast, mutation of the two C-terminal serines, S294 and S300, had only a weak negative effect on *ASN1* and no significant effect on *ProDH* activation. Taken together, our data suggest that AKIN10 phosphorylates bZIP63 at up to three conserved sites, namely S29, S294, and S300 and phosphorylation by AKIN10, especially at S29, is crucial for bZIP63 TF activity.

## The AKIN10 target sites play an important role for bZIP63 function in planta

To determine the impact of bZIP63 phosphorylation in planta, we transformed the *bzip63* mutant with genomic constructs of bZIP63 containing either the wt sequence (GY lines), or a S/A mutation of S29, S294, and S300, respectively (GAY lines) with a C-terminal YFP tag (*Figure 7A*; *Figure 7—figure supplement 1A–C*). The wt construct showed strong phosphorylation in the light (bands 6 and 7) and an even stronger phosphorylation in extended night (band 8), as well as reduced phosphorylation in the presence of sucrose (bands 1, 2, and 4) (*Figure 7B*; *Figure 7—figure supplement 1D*). In contrast, the S/A construct showed only weak phosphorylation (only bands 1 and 3 visible) and most importantly no difference between all tested conditions. This indicates that S29, S294 and S300 are the major in vivo phosphorylation sites on bZIP63, which are also responsible for the observed condition-dependent shift in bZIP63 phosphorylation.

Next, we tested whether complementation of the observed *bzip63* phenotypes depends on the bZIP63 phosphorylation status, using two independent GY and GAY lines. The dark-induced senescence phenotype of *bzip63* was complemented in the GY lines, but not in the GAY lines (*Figure 7C,D*; *Figure 7—figure supplement 2*). After 9 days in darkness, the ko and GAY lines showed visibly less chlorosis and had a higher percentage of green leaf area as compared to the wt and GY lines. Metabolite profiling of leaves harvested after 6 hr of light also revealed marked differences between the GY and GAY lines (*Figure 7E,F*; *Figure 7—source data 1*). The metabolite profile of the GAY lines was similar to that of *bzip63* plants. In contrast, in the GY lines the metabolic changes between mutant and wt were mostly weaker than in *bzip63* or even resembled those observed for ox#3 (*Figure 7E*). In a principal component analysis the GAY lines grouped together with *bzip63*, while the GY lines were closer to the two wt lines and the ox (*Figure 7F*).

To test the effect of the S/A mutation on the expression of bZIP63 target genes, we performed RT-qPCR of *ASN1*, *DIN10*, and *ProDH* (*Figure 7G*)—three suggested AKIN10 target genes (*Baena-González et al., 2007*). The expression of all three genes increased steadily during a 4 hr extended night treatment, but the increase was delayed in the *bzip63* mutant as compared to the wt. At the 4 hr time point we could observe a clear difference between wt and ko. We therefore chose this time point to quantify *ASN1*, *DIN10*, and *ProDH* transcripts in one GY and GAY line. As expected, the GY line had the same or even more transcript than the wt for all three genes, while the GAY line had significantly lower expression levels, similar to *bzip63*.

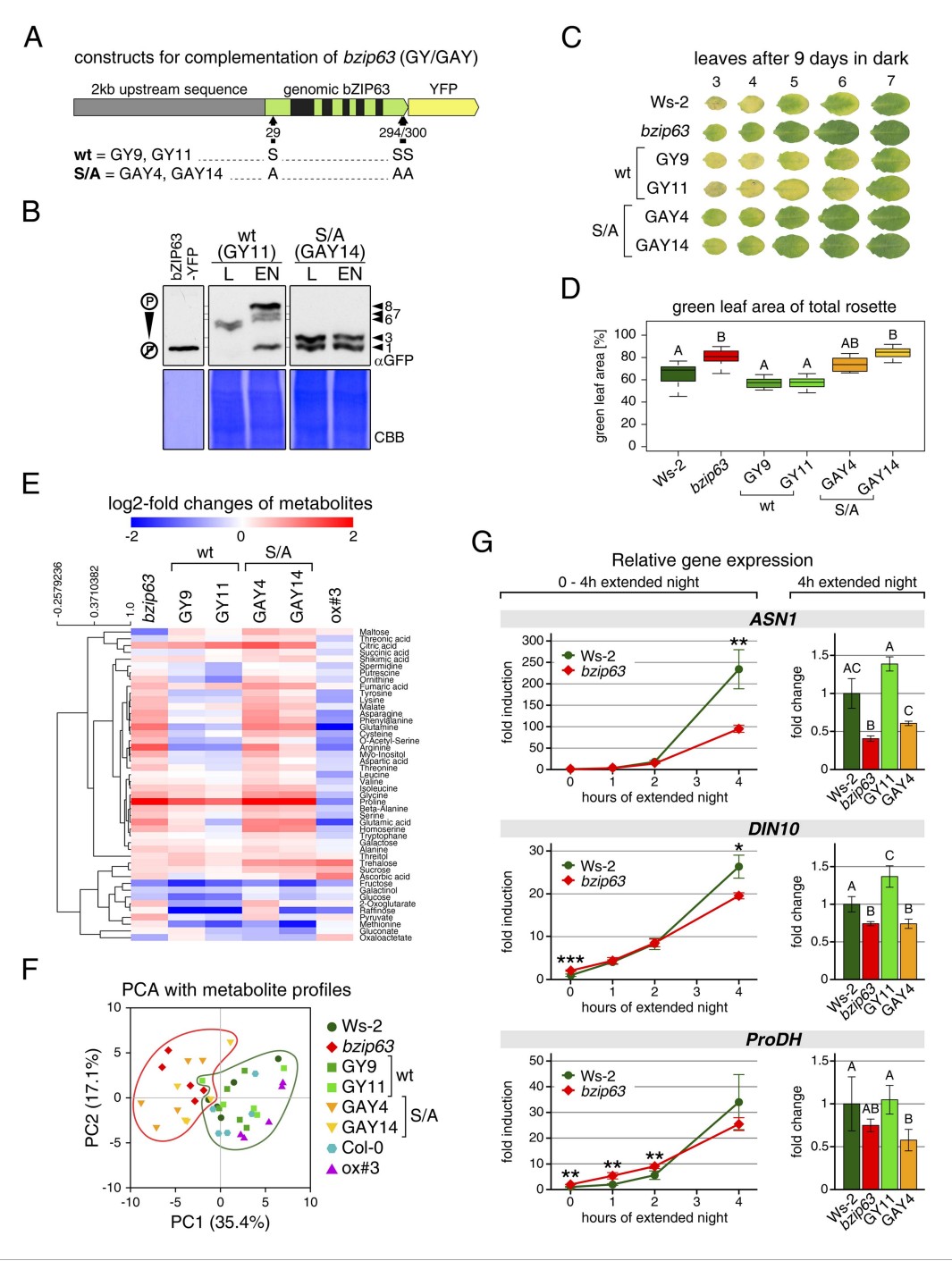

**Figure 7**. The *bzip63* phenotype can be complemented by wt bZIP63, but not by bZIP63 harboring S/A mutations of the AKIN10 target sites. (**A**) Genomic complementation constructs. Exons are green, introns black. See *Figure 7—figure supplement 1* for characterization of the complementation lines. (**B**) Phos-tag gel western blots (αGFP) showing the in vivo phosphorylation state of bZIP63 in the complementation lines after 6 hr light (L) or extended night (EN) in 5 week-old soil-grown plants. Recombinant bZIP63-YFP was used as a nonphosphorylated control. Numbered arrowheads on the right mark the position of each observed bZIP63 band for easy reference with other figures (see *Figure 3—figure supplement 1* for a comparative image of all Phos-tag western blots). (**C**) and (**D**) Dark-induced senescence phenotype of 4.5 week-old soil-grown plants after 9 days in darkness. (**C**) Leaves 3–7 of one representative plant per line. (**D**) Box-and-whiskers plot of the total green leaf area of eight biological replicates. Letters indicate significant differences as determined by ANOVA and pairwise T-testing

*Figure 7. continued on next page*

*Figure 7. Continued*

(p < 0.05). See **Figure 7—figure supplement 2** for untreated plants and green leaf area of individual leaves. (**E**) and (**F**) Metabolite profile. (**E**) Hierarchical clustering of log-2 fold changes of metabolite concentrations compared to wt. Values are means of five biological replicates. (**F**) Principal component analysis (PCA). PC1 is plotted against PC2. The proportion of variance in % is indicated. The red line surrounds *bzip63* and GAY samples, the green line wt, GY, and ox#3 samples. For relative metabolite concentrations and PCA loadings see **Figure 7—source data 1**. (**G**) Relative expression of potential bZIP63 target genes in 5 week-old plants during early extended night as determined by RT-qPCR. Values are means ± SD of four biological replicates and are given as fold change compared to Ws-2 at 0 hr (left) or 4 hr (right). p-values from T-tests between mutants and wt < 0.05, < 0.01, and < 0.001 are indicated by *, ** and ***, respectively. Letters indicate significant differences as determined by ANOVA and pairwise T-testing (p < 0.05). CBB, Coomassie brilliant blue.

The following source data and figure supplements are available for figure 7:

**Source data 1**. Excel table containing the relative metabolite levels of the complementation lines and the PCA loadings.

**Figure supplement 1**. Characterization of the *bzip63* complementation lines.

**Figure supplement 2**. Complementation of the dark-induced senescence phenotype of *bzip63*.

In summary, expression of wt bZIP63 but not of the S/A mutant—which cannot be phosphorylated by AKIN10—in the *bzip63* background led to complementation of the *bzip63* phenotypes. Together, these experiments demonstrate that phosphorylation of bZIP63 at the AKIN10 target sites is essential for the function of bZIP63 in the plant.

## AKIN10-mediated phosphorylation affects bZIP63 dimerization

Our findings show that bZIP63 phosphorylation, at residues distant from the central bZIP domain, strongly regulates its activity. As we did not observe any changes in localization in the bZIP63 mutants and also no change in DNA-binding activity (of the bZIP63 homodimer) towards a C-box motif (GACGTC) as canonical bZIP target site (data not shown), we suspected that the observed effect on transcription would be due to changes in dimerization preferences of bZIP63. Therefore, we tested the effect of AKIN10-mediated phosphorylation on bZIP63 homo- and hetero-dimerization with bZIP1 and bZIP11. Both bZIP1 and bZIP11 are metabolic regulators and mediate transcription of *ASN1* and *ProDH* (**Hanson et al., 2008**; **Dietrich et al., 2011**). In protoplast two-hybrid (P2H) assays, the addition of exogenous AKIN10 resulted in a clear enhancement of dimerization in all cases—homo-dimerization as well as hetero-dimerization with bZIP1 and bZIP11 (**Figure 8A**; **Figure 8—figure supplement 1A**). Note, that the very strong effect observed with bZIP11 in these assays is misleading because bZIP11 is a much stronger activator of transcription as compared to bZIP1 and 63 (see also 'Discussion'). From that we concluded that the phosphorylation of bZIP63 by AKIN10 is required for dimerization and set out to test the effect of S/A mutations of the AKIN10 target sites on bZIP63 homo- and hetero-dimerization with bZIP11 where we had observed the strongest effect before (**Figure 8B**; **Figure 8—figure supplement 1B**). In both cases, the signal was reduced to about 30–40% of the signal obtained from dimerization with wt bZIP63 when S29 or all three serines were mutated to alanine. Mutation of one or two of the C-terminal sites decreased bZIP63 homo-dimerization weakly but had no visible effect on bZIP63-11 dimerization, indicating that these sites play, at most, a minor role in regulation of dimer formation.

To exclude the possibility that the observed effects on dimerization are due to phosphorylation of the hetero-dimerization partner rather than bZIP63 itself, we tested whether AKIN10 is able to phosphorylate any of the S1 group bZIPs. To increase the phosphorylation efficiency of AKIN10 we included SnAK2 in the reactions. In contrast to bZIP63, none of the S1 group bZIPs were phosphorylated by AKIN10 (**Figure 8C**; **Figure 8—figure supplement 2**). Together, these data indicate that AKIN10-mediated phosphorylation of bZIP63—especially at S29—strongly enhances its ability to form homo- as well as hetero-dimers.

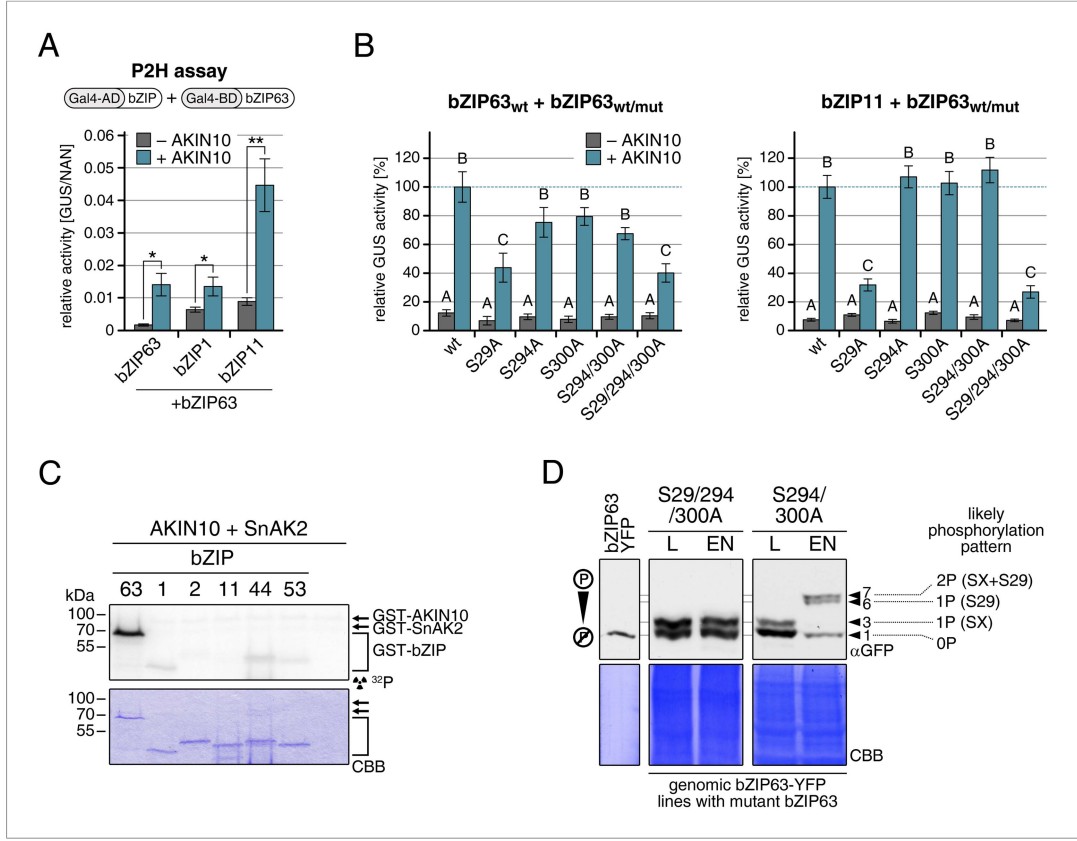

**Figure 8**. AKIN10-mediated phosphorylation of bZIP63 promotes its dimerization. (**A**) and (**B**) Protoplast two-hybrid (P2H) assays. (**A**) Interaction of bZIP63 fused to the Gal4-AD (activation domain) with bZIP63, bZIP1, and bZIP11 fused to the Gal4-BD (binding domain) without and with co-transformation of AKIN10. Bars represent the mean normalized GUS activity ± SD of 3–4 biological replicates. p-values from T-tests < 0.05 and < 0.01 are indicated by * and **, respectively. (**B**) Interaction of AD-bZIP63 (left) or AD-bZIP11 (right) with wt and S/A mutants of BD-bZIP63 without and with co-transformation of AKIN10. Values are given in % of the signal with wt bZIP63 and AKIN10. Letters indicate significant differences as determined by ANOVA and pairwise T-testing (p < 0.05). For western blots for (**A**) and (**B**) see *Figure 8—figure supplement 1*. (**C**) In vitro kinase assay of bZIP63, 1, 2, 11, 44, and 53 with AKIN10 and the SnRK1 upstream kinase SnAK2. For a kinase assay with the bZIPs and SnAK2 alone see *Figure 8—figure supplement 2*. (**D**). Phos-tag gel western blots (αGFP) showing the in vivo phosphorylation state of S29 in bZIP63 after 6 hr light (L) or extended night (EN). 5 week-old soil-grown plants of two genomic *bzip63* complementation lines harboring different S/A mutations were used. In line GAY14 (left) none of the AKIN10 target sites (S29/294/300) can be phosphorylated, while in the S294/300A (right) line S29 can still be phosphorylated. Numbered arrowheads on the right mark the position of each observed bZIP63 band for easy reference with other figures (see *Figure 3—figure supplement 1* for a comparative image of all Phos-tag western blots). The likely phosphorylation state of the bands in the western blot is shown on the right. X stands for one of the non-mutated serines (59, 102, 160, or 261). CBB, Coomassie brilliant blue.

The following figure supplements are available for figure 8:

**Figure supplement 1**. Western blots and controls for the protoplast two-hybrid (P2H) assays.

**Figure supplement 2**. SnAK2 does not phosphorylate bZIP63 or the S1 class bZIPs.

To further substantiate the relevance of S29 phosphorylation in vivo, we compared the phosphorylation patterns of the S29/S294/S300A mutant with a S294/S300A mutant in the light and after extended night treatments in the genomic complementation lines (*Figure 8D*). Compared to the triple mutant, that did not show any change in the phosphorylation pattern in response to starvation, bZIP63 phosphorylation was clearly increased in the S294/S300A double mutant in the

extended night. As S29 is the only AKIN10 target site left in this version, it must indeed be S29 that is phosphorylated by AKIN10 under starvation conditions.

As we saw that AKIN10-dependent phosphorylation promotes dimerization of bZIP63 with all tested partners we wanted to analyze whether phosphorylation has an effect on the dimerization partner preference. Because protoplast two-hybrid assays only allow to analyze the effect of AKIN10 on a single dimer we set up a multicolor BiFC approach (*Waadt et al., 2008*) in *Nicotiana tabacum* leaves with bZIP1 and 11 as alternative interaction partners for bZIP63. For this, we expressed all three bZIPs (1, 11, and 63) from the same plasmid and co-transformed AKIN10 (*Figure 9A*). bZIP11 was fused to $CFP_N$, bZIP1 to $VENUS_N$, and bZIP63 to $CFP_C$. Accordingly, the bZIP63-11 hetero-dimer generates a blue (cyan) signal and the bZIP63-1 hetero-dimer a yellow signal, whereas the bZIP63 homo-dimer cannot be detected. To compare the dimerization preference of phosphorylated and non-phosphorylated bZIP63 we used wt and a S29/294/300A mutant of bZIP63. Quantification of the ratio of the VENUS/CFP signal in more than 100 nuclei clearly showed a shift towards the VENUS emission when wt bZIP63 was compared to the S/A mutant. This demonstrates that phosphorylation by AKIN10 triggers the preferential formation of bZIP63-1 over bZIP63-11 dimers in a competitive in vivo assay.

## Discussion

### bZIP63 is an important metabolic regulator in the starvation response

Here we show that bZIP63 plays an important role in the energy starvation response and metabolic regulation. This is in accordance with the sugar/energy-dependent expression of bZIP63 (*Matiolli et al., 2011*; *Kunz et al., 2014*), as well as with the fact that several of its proposed target genes are involved in the low-energy response and metabolism (*Matiolli et al., 2011*; *Veerabagu et al., 2014*). Furthermore, three members of the S1-group of plant bZIPs—hetero-dimerization partners of bZIP63 (*Ehlert et al., 2006*; *Kang et al., 2010*)—have also been linked to energy starvation response and metabolism. Inducible bZIP11 ox lines exhibit a severe dwarf phenotype (*Hanson et al., 2008*) and a metabolic profile resembling that of carbon starved plants (*Ma et al., 2011*), while overexpression of bZIP1 and bZIP53 results in enhanced dark-induced senescence and reduced levels of proline and branched-chain amino acids (*Dietrich et al., 2011*).

Similar to the bZIP1 ox, bZIP63 ox plants showed increased chlorosis after 9 days of darkness, while the bZIP63 ko displayed a clear stay-green phenotype under these conditions. In contrast, neither the single, nor the double ko of bZIP1 and 53 showed reduced dark-induced senescence (*Dietrich et al., 2011*). This suggests that other bZIPs can take over the function of bZIP1 in starvation-induced leaf yellowing, while bZIP63 plays a more unique role.

Looking at primary metabolism, we found that misregulation of bZIP63 expression has a strong effect, especially on amino acids, which was further enhanced under starvation conditions. In line with the finding that bZIP63 is a positive regulator of *ProDH* and *ASN1* (*Matiolli et al., 2011*; *Veerabagu et al., 2014*, and this work: *Figures 6C, 7G*) and the changes in proline levels in *bZIP63* mutants reported by *Veerabagu et al. (2014)*, we measured the strongest differences in proline and the entire glutamate family as well as in aspartate and asparagine levels in the bZIP63 ko and ox. Notably, there is a strong correlation between the amino acid profiles of the bZIP63 and the bZIP1 ox lines (*Dietrich et al., 2011*). A similar accumulation of amino acids during dark-induced senescence has frequently been observed and was attributed to enhanced protein degradation (reviewed in *Araújo et al., 2011*). Particularly under low carbon conditions or during senescence, alternative energy sources need to be used in plant cells, and the important role of proline and branched-chain amino acids in this process has been highlighted in a number of studies (reviewed in: *Szabados and Savouré, 2010*; *Szal and Podgórska, 2012*). Thus also the altered amino acid levels could contribute to the observed dark-induced senescence phenotype of the bZIP63 mutants.

### bZIP63 function is regulated by SnRK1-dependent phosphorylation

We found that bZIP63 is highly phosphorylated in *Arabidopsis*. By applying different proteolytic digests we were able to identify seven in vivo phosphorylated serine residues, distributed all over the protein. While exogenous sucrose decreased the global phosphorylation level of bZIP63, extended night treatment further increased its phosphorylation, thus supporting the idea that bZIP63 plays a role in energy signaling. Moreover, we found that the SnRK1 kinase AKIN10, which was proposed to

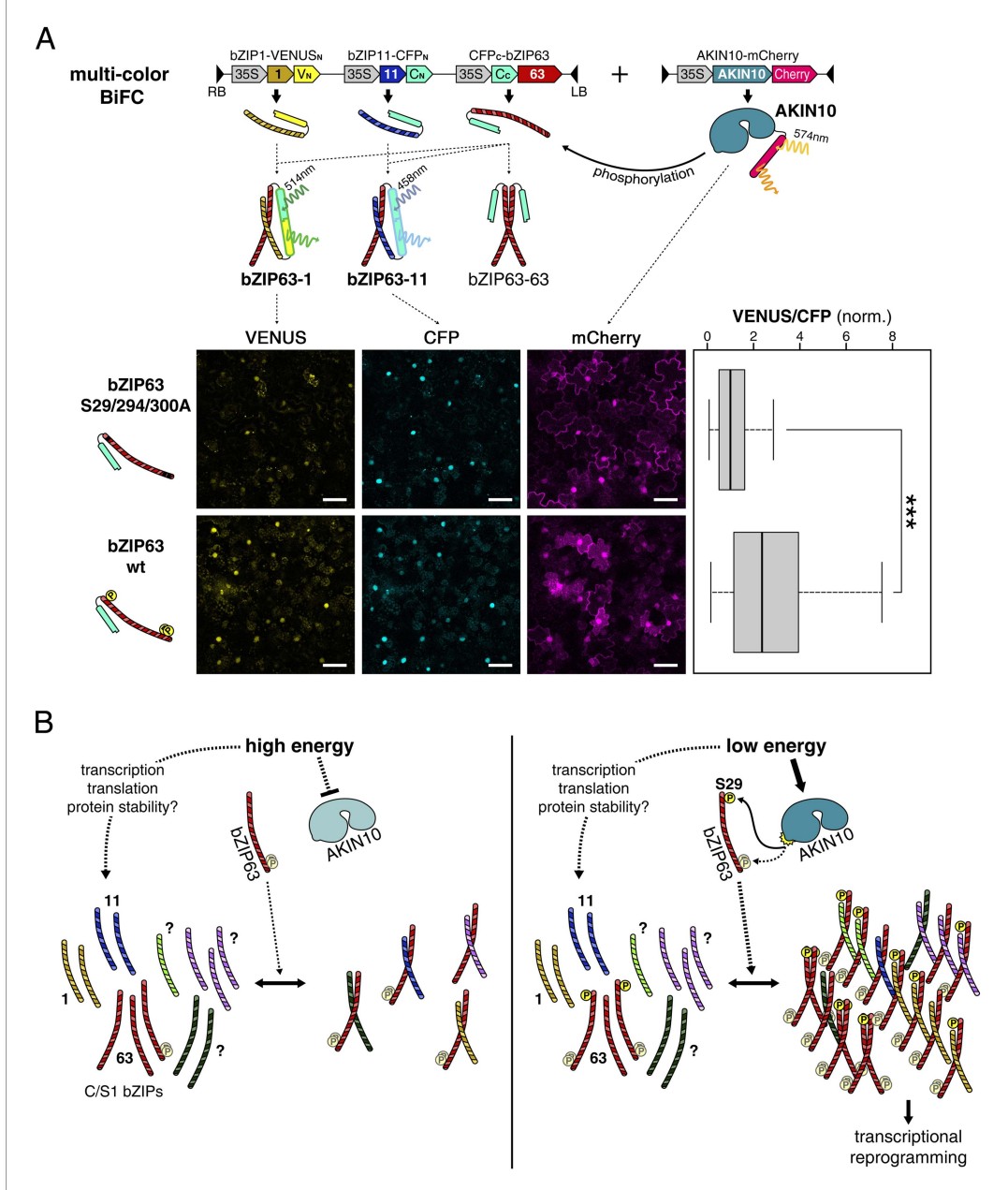

**Figure 9**. Phosphorylation of bZIP63 shifts its dimerization preferences. (**A**) Multi-color bimolecular fluorescence complementation (BiFC) in transiently transformed *N. tabacum* leaves to test the effect of bZIP63 phosphorylation on its dimerization preference. A cassette containing bZIP63 (wt or S29/294/300A), bZIP11, and bZIP1—tagged with the C-terminal moiety of CFP and the N-terminal moieties of CFP and VENUS, respectively—was co-transformed with mCherry-tagged AKIN10. CFP (bZIP63-11) and VENUS (bZIP63-1) fluorescence was detected on a confocal laser scanning microscope and quantified for nuclei showing co-expression of AKIN10. Top: scheme of the multi-color BiFC construct and principle. Bottom center: representative microscopy pictures. Size bar = 50 µm. Bottom right: box-and-whiskers plot of the VENUS/CFP ratio of 115–118 nuclei, normalized to the median of the S29/294/300A bZIP63 construct. T-test p-value was < 0.001, as indicated by ***. (**B**) Model of the regulation of bZIP63 dimerization and activity by AKIN10. Energy deprivation triggers activation of AKIN10, which phosphorylates S29 on bZIP63. This leads to increased formation of specific bZIP63 dimers and altered expression of dimer-specific genes.

be a central regulator of transcription in starvation response (*Baena-González and Sheen, 2008*), is the major kinase responsible for the starvation-induced hyper-phosphorylation of bZIP63. Therefore, bZIP63 presents the first TF acting as direct target of SnRK1 in the transcriptional energy deprivation response. In vitro, AKIN10 phosphorylated three highly conserved and functionally important residues in the N- and C-terminus of bZIP63—S29, S294, and S300, respectively. Reporter activation assays in protoplasts revealed that these sites—especially S29—are essential for AKIN10-dependent induction of *ASN1* and *ProDH* by bZIP63. Remarkably, the phosphorylation patterns of different bZIP63 mutants on Phos-tag gels pointed towards S29 as the main target site for AKIN10 under extended night conditions. The importance of bZIP63 phosphorylation at the SnRK1 target sites for its in planta function was further underlined by complementation of the *bzip63* mutant with genomic bZIP63 constructs. Wt, but not a S29/S294/S300A mutant of bZIP63, could complement the metabolic and senescence phenotypes. Likewise, also the strong delay in extended night-triggered induction of *ASN1*, *DIN10*, and *ProDH* in *bzip63* plants was complemented with wt bZIP63, but not the (triple) S/A mutant. The relatively small difference in *ProDH* expression in plants, as compared to the protoplast assay, is probably due to redundancy within the C/S1 group bZIPs, as it was shown by *Dietrich et al. (2011)*, that only ko of multiple members of the C/S1 group leads to a strong reduction in *ProDH* and *ASN1* expression.

Since the discovery that AKIN10 can activate several members of the C/S1 network (*Baena-González et al., 2007*), it has been speculated (*Baena-González and Sheen, 2008*; *Usadel et al., 2008*), but never shown experimentally, that they are downstream targets of SnRK1. Our data provide compelling evidence that bZIP63, but none of the S1 group bZIPs, is a bona fide in vivo target of SnRK1 in low-energy signaling.

## Phosphorylation of bZIP63 alters its dimerization preferences

Differential dimerization is a well-known mechanism for changing the target recognition site, and thereby the target genes of bZIP TFs (*Tsukada et al., 2011*). For the S1-group bZIP1 is has been shown that dimerization with C-group bZIP10 or bZIP63 affects its in vitro binding to ACGT-based motifs differently (*Kang et al., 2010*). The notion that different C/S1 dimers have different target genes is further supported by a recent transcriptomics study in protoplasts. Overexpression of four C/S1 group bZIPs (bZIP1, bZIP10, bZIP11, and bZIP63), individually or in combination of two, revealed overlapping but distinct gene expression patterns (*Ma, 2012*). This means, that although they regulate a core set of common genes, such as *ASN1* and *ProDH* (*Baena-González et al., 2007*; *Dietrich et al., 2011*; *Ma et al., 2011*), different dimers also have distinct functions, and switching of dimerization partners can have a considerable impact on gene expression.

Our data indicate that dimerization and activity of bZIP63 strongly depend on its phosphorylation status. In general, the dimerization of bZIP63 with bZIP1, bZIP11, and bZIP63, was boosted by AKIN10-mediated phosphorylation. Like in the promoter activation assays, the main influence on dimerization came from S29 phosphorylation, while phosphorylation of S294 and S300 showed a mild effect at most. Incidentally, this enhanced dimerization with bZIP63 would explain why AKIN10 was found to activate the S1 group bZIPs (bZIP1, 2, 11, and 53) in protoplast assays (*Baena-González et al., 2007*), although they do not present direct targets of AKIN10.

At first sight, the phosphorylation triggered boost of dimerization seemed to be strongest for hetero-dimerization with bZIP11. However, it has to be considered that bZIP11 is a very strong activator of transcription as it harbors an activation domain (*Ehlert et al., 2006*) and was recently also shown to recruit the histone acetylation machinery (*Weiste and Droege-Laser, 2014*). The same is not the case for bZIP1 and bZIP63. As the readout of the protoplast two-hybrid interaction assay is transcription-based it is therefore not possible to quantitatively compare the effect of phosphorylation on the formation of different dimers and to draw conclusions about the in vivo dimerization preference. Therefore we used a multi-color BiFC assay with bZIP1 and bZIP11 as alternative partners for bZIP63 to test the effect of bZIP63 phosphorylation by AKIN10 on its dimerization preference. The three bZIPs were co-expressed with AKIN10 in *N. tabacum* leaves and the formation of bZIP63-1 and bZIP63-11 dimers was quantified. Comparison of wt and S/A mutated bZIP63 revealed that, phosphorylation of bZIP63 shifts the dimerization preference from bZIP11 towards bZIP1. This trend fits well to the observation that both TFs are strongly upregulated in the night and extended night, while bZIP11 is downregulated under these conditions (see *Figure 5—figure supplement 3*). Furthermore, increased expression of bZIP63 and bZIP1 under conditions when bZIP63 is phosphorylated might further enhance the observed shift in the dimerization towards bZIP63-1 dimers.

Based on the data presented in this study, we propose a simplified model for the regulation of bZIP63 dimerization by AKIN10 (*Figure 9B*). When bZIP63 is not phosphorylated, its capacity to dimerize with other bZIPs is rather low. Under starvation conditions, bZIP63 is phosphorylated at S29 by AKIN10, favoring the formation of bZIP63 homo- and specific hetero-dimers, particularly of bZIP63-1 dimers. This ultimately results in the induction of a different set of target genes and thereby mediates the transcriptional reprogramming of metabolism. However, it is clear that in the plant additional factors like bZIP expression, stability and interaction with other components add more complexity to the situation.

Surprisingly, to date only a small number of papers have reported an influence of phosphorylation on bZIP dimerization (*Kim et al., 2007*; *Guo et al., 2010*; *Lee et al., 2010*). Moreover, to our knowledge this is the first time that phosphorylation outside the bZIP domain was shown to affect dimerization with different partners in a distinct way. While there are numerous reports on phosphorylation-mediated changes in bZIP activity in animals, plants, and yeast, in many cases the underlying mechanism is still unknown. We therefore believe that this novel mechanism of phosphorylation-triggered switch of bZIP dimerization partners could play a substantial role in the regulation of bZIP TF activity in all higher organisms, and should be further addressed in future studies.

## Materials and methods

### Plant lines

The lines ox#2 and ox#3 are bZIP63 ox lines in the Col-0 background, expressing bZIP63.2 with a C-terminal GFP tag under the control of the 35S promoter. Generation of these plant lines was previously described in *Veerabgu et al. (2014)*. Overexpression was confirmed by RT-qPCR of *bZIP63* mRNA (*Figure 1—figure supplement 1E*).

The bZIP63 ko line (*bzip63*) in the Ws-2 ecotype is a T-DNA insertion line. Pool number CSJ1 (NASC ID: N700001) from the *Arabidopsis* Knockout Facility (AKF) (*Sussman et al., 2000*) was screened for a T-DNA insertion in *bZIP63* and homozygous plants were selected using Kanamycin. Sequencing of the flanking regions revealed that the T-DNA is inserted in the first exon at position 76 (*Figure 1—figure supplement 1A*). The ko was confirmed by PCR and RT-qPCR of *bZIP63* (*Figure 1—figure supplement 1B–D*). The same line was used by *Veerabgu et al. (2014)*.

For the complementation lines (GY9 + GY11 = wt, GAY4 + GAY14 = S29/294/300A, S294/300A), a genomic fragment containing *bZIP63* and 2 kb of upstream sequence was obtained by PCR on Col-0 genomic DNA and ligated into pCRBlunt (Invitrogen, Austria). For the S/A constructs, first S294 and S300 were mutated to alanine by mutagenesis PCR and the plasmid was subsequently used to mutate S29 to alanine (see *Table 1* for primers). Correct sequences were confirmed by restriction digests and sequencing. The genomic fragments were then cloned KpnI/NotI into modified pBIN19, containing a BASTA resistance for plant selection and a C-terminal YFP tag, before transformation into the *Agrobacterium tumefaciens* strain GV3101 (pMP90). Homozygous *bzip63* plants were transformed with the floral dip method and selected for positive transformation events by spraying seedlings with 200 mg/l BASTA solution (Bayer, Germany). Transgene expression in GY and GAY lines was tested using RT-qPCR, western blots with an antibody against GFP, and epifluorescence microscopy (*Figure 7—figure supplement 1A–C*). The S294/300A shown in *Figure 8D* was at the time of submission still heterozygous and in the T2 generation. Therefore, expression of the transgene was checked by epifluorescence microscopy (not shown), but no quantification by RT-qPCR or western blotting was done.

The AKIN10 ko line (*akin10*, GABI_579E09) in the Col-0 ecotype is a T-DNA insertion line from the GABI KAT collection (*Kleinboelting et al., 2012*). For this line, two T-DNA insertions were suggested, one in *AKIN10* and another one in *IMS2* (2-ISOPROPYLMALATE SYNTHASE 2, At5g23020). The insertion in *AKIN10* in front of the last exon was confirmed by PCR, but the second insertion in *IMS2* could not be detected and was thus assumed not to be present (*Figure 5—figure supplement 1A–C*). The ko of *AKIN10* was further confirmed by RT-qPCR and western blotting, respectively. No significant amount of transcript or protein could be detected (*Figure 5—figure supplement 1D,E*). The *akin10* line was further tested for obvious growth and developmental phenotypes by comparing leaf number, fresh and dry weight, water content, and flowering time in soil-grown wt and mutant. Only a slight delay in flowering was observed (*Figure 5—figure supplement 2A,B*). Expression of

**Table 1.** List of primers

| Gene | Forward primer | Reverse primer |
|---|---|---|
| **Primers for cloning** | | |
| genomic *bZIP63* | GGTACCAAAACTATAAATTTCTTGTAGGACAGTG | TTGCGGCCGCCCTGATCCCCAACGCTTCGAATACG |
| *bZIP63.2* | GGGCCCATGGAAAAAGTTTTCTCCGACGAAGAAATCTCC | TTGCGGCCGCCCTGATCCCCAACGCTTCGAATACG |
| *AKIN10.1+3* | GGGCCCATGGATGGATCAGGCACAGGCAGTA | GCGGCCGCAGAGGACTCGGAGCTGAGCAA |
| *AKIN11.1+2* | GGGCCCATGGATCATTCATCAAATAG | GCGGCCGCAGATCACACGAAGCTCTGTAA |
| *SNF4* | GGGCCCATGTTTGGTTCTACATTGGA | GCGGCCGCAAAGACCGAGCAGGAATTGGAA |
| *AKINβ1.1* | GGGCCCATGGGAAATGCGAACGGCAAA | GCGGCCGCACCGTGTGAGCGGTTTGTAG |
| *AKINβ2.1+2* | GGGCCCATGTCTGCTGCTTCTGATGGT | GCGGCCGCACCTCTGCAGGGATTTGTAG |
| *bZIP1* | CGATGGGCCCCATGGCAAACGCAGAGAAG | CGATGCGGCCGCTGTCTTAAAGGACG |
| *bZIP2* | AAAACCATGGCGTCATCTAGCAGCAC | TGCGGCCGCTATACATATTGATATCATTAG |
| *bZIP11* | TAGGGCCCATGGAATCGTCGTCGTCGGGAA | TGGCGGCCGCAATACATTAAAGCATCAGAAG |
| *bZIP44.1* | CAGGGCCCATGAATAATAAAACTG | CTGCGGCCGCAACAGTTGAAAACATC |
| *bZIP53* | AAAACCATGGGGTCGTTGCAAATGCAAAC | TGCGGCCGCTGCAATCAAACATATCAGCAG |
| *AKIN10 VIGS* | GGAATTCACTTTTTCAGCTCAGAAAATTTTG | GGGGTACCCCCTCGAGCCACTCCTGATATTATCTGCTG |
| *AKIN11 VIGS* | CCTCGAGGTTTCTGTATATTTCTTCGCTC | GGGGTACCCAGTACTCTACACCAGATATTATC |
| *BiFC MCS1* | TCTAGAGGGCCCATGGGCCACAGACACAG | CCCGGGGCGGCCGCACATTCTGCGTC |
| *BiFC MCS2* | AAAAGGATCCGGGCCCATGGGCCACAGACACAGCAAGTCCAAATCCTCCG | CCCGGGGCGGCCGCACATTCTGCGTC |
| *multi-color BiFC bZIP1* | TCGGTACCAAGCTTAGCTTGCATGCCTGCAGG | CAGGATCCAGCTGGCGAAAGGGGGGATGTGCTG |
| *multi-color BiFC bZIP11* | CAGGATCCTTAGCTTGCATGCCTGCAGGTCC | CTAGGCCTAAGCTTCGCTATTACGCCAGCTGGCGAAAGG |
| **Mutagenesis** | | |
| *bZIP63S29A* | CGTCGTTGAATCGCTCGGCCGCCGAATGGGCATTCAATC | ACGTCATTCCATTAACCGACCAGTG |
| *bZIP63S59A* | GTGTGGTGTTTCCGTCTCCGCTCCTCCTAATGTTCCTG | CACACGCCGTCGTAGATTCTCCGTCGTC |
| *bZIP63S102A* | GATACTTCTGGTAGAGCTGACAATGGTGGAGC | GTATCCTGAGGTTTGATGAAAGTTC |
| *bZIP63S160A* | CTGATTCTCTATTAGCTAGCATCCTTTTAACG | ATCAGCTAGACGGTCCAGAAGAAG |
| *bZIP63S261A* | CAGAGACATCAAATGCTCCAGACACTACAAG | CTTGTAGTGTCTGGAGCATTTGATGTCTCTG |
| *bZIP63S294A* | GAACAGAACAGCTGCCATGCGTAGAGTTGAG | TGTTCATCTTGCACCCTATCAAGGC |
| *bZIP63S294/300A* | GAACAGAACAGCTGCCATGCGTAGAGTTGAGGCCTTGGAACATCTGCAG | TGTTCATCTTGCACCCTATCAAGGC |

*Table 1. Continued on next page*

*Table 1. Continued*

| Gene | Forward primer | Reverse primer |
|---|---|---|
| AKIN10K48M | GGTTGCTATCATGATCCTCAATCGTCG | GCAACCTTATGTCCTGTCAATGC |
| **qPCR** | | |
| bZIP63 | GAAGAAATCTCCGGTAACCATCAC | GATTCTCCGTCGTCTGCAGC |
| bZIP1 | AACGCGGGTCTTAGATCGGAGAAG | TCAGCGTTAAACTCGTCGTAGCAA |
| bZIP11 | TCGTCAGGATCGGAGGAGAGT | GATCGTCTAGGAGCTTTTGTTTCTTC |
| CAB2 | TCAATCTTTTGAATTCGAGTGAGA | TCCACCACAAACACAAACCTAC |
| SEN1 | CAGAGTCGGATCAGGAATGG | ATTATGATTTCATCGTGTTTCC |
| SAG103 | AGCTCGAGTGCTGGGATG | CGGATTCACAGATCCTTCCT |
| YLS3 | GACATCACTAAGTGCCCTGCT | ACTGTTTCGTTCAGACCTTTAGC |
| AKIN10 | ACTGGATTTGCAGAGAGTACAAGGTCC | TCAGAGGACTCGGAGCTGAGCA |
| AKIN11 | GCTCGTAACTTTTTCCAGCAGA | TTCAGGTCTCTATGGACAACCA |
| ASN1 | GTCGCAAGATCAAGGCTC | TGAAGTCTTTGTCAAGGAAAGG |
| DIN10 | GCTTGTATTGCCTGATGGA | ATCTTTAGCAAGCTGACACC |
| ProDH | CGCCAGTCCACGACACAATTCA | CGAATCAGCGTTATGTGTTGCG |
| BCAT2 | AAGGTTATCAGGTAGTAGAGAAGG | TTCCTGATATGTGATAGTGCC |
| AA-TP family protein | GTTCTTGGGATCAACTTCTACAG | AACCATTTACATTCCGTAGGAC |
| TBP2 | TGCAAGAAAGTATGCTCGG | ACATGAGCCTACAATGTTCTG |
| MHF15 | GTTTCCTGAGCTTCTCCAC | TGGTCGCTTCATCTTGAG |
| **Characterization of plant lines (PCR)** | | |
| bZIP63 | TTGCGGCCGCCCTGATCCCCAACGCTTCGAATACG | AACCATGGATAATCACACAGCTAAAGACATTGG |
| bzip63 RB | TGGCGAATGAGACCTCAATTGCGAGCTTT | |
| bzip63 LB | CATTTTATAATAACGCTGCGGACATCTAC | |
| AKIN10 | CCAGCATAATAGAGAACGAAGC | GTCCGGTTTAGTATTCAGAGG |
| akin10 LB | ATATTGACCATCATACTCATTGC | |

selected AKIN10 target genes after 6 hr of light or extended night was tested by RT-qPCR. Some genes showed a slight, but not dramatic reduction in expression (*Figure 5—figure supplement 2C*).

For the line expressing bZIP63-GFP in *akin10*, *bZIP63.2* was amplified from Col-0 cDNA (see *Table 1* for primers) and cloned ApaI/NotI into modified pBIN19 containing the UBI10 promoter, a BASTA resistance for plant selection, and a C-terminal YFP tag. Homozygous *akin10* plants were transformed and selected like the GY and GAY lines.

## Plant growth

*Arabidopsis* seeds were surface sterilized with chlorine gas before sowing on soil or growth medium and then vernalized at 4°C for 2 days. Plants were grown in a growth chamber in a 12 hr light/12 hr dark regime with day temperatures between 20 and 22°C and night temperatures between 18 and 20°C and a light intensity of 60–150 µmol/m²s unless specified otherwise. Plants grown under short or long day conditions were cultivated with 8 hr or 16 hr of light per day, respectively. The soil mixture consisted of 4 parts Huminsubstrat N3 (Neuhaus, Germany), 1 part perlite (Gramoflor, Germany), and the fertilizer Osmocote (Substral/Scotts, Germany) according to manufacturer's instructions. For hydroponic cultures, plants were grown with their roots in light-tight box filled with liquid ½ Hoagland

medium (2 mM Ca(NO$_3$)$_2$, 0.25 mM K$_3$PO$_4$, 3 mM KNO$_3$, 1 mM MgSO$_2$, 45 µM NaFeIII EDTA, 5 µM H$_3$BO$_3$, 1 µM MnCl$_2$, 0.15 µM ZnSO$_4$, 0.1 µM CuSO$_4$, 7 nM MoO$_3$, 4.5 nM Co(NO$_3$)). Seedling cultures in liquid medium were grown in ½ Gamborg medium (Duchefa, Harlem, The Netherlands) with or without 0.5% sucrose and the medium was exchanged after 7 days. Seedlings on plates were grown on ½ MS (Duchefa, Harlem, The Netherlands) with 0.7 g/l plant agar (Duchefa, Harlem, The Netherlands) and a pH of 5.8. The root cell suspension culture was grown in MS medium containing 30 g/l sucrose and 2.5 µM 2,4D at 22°C in the dark under constant shaking. The medium was exchanged every 7 days by transferring ⅓ to ½ of the culture to a fresh flask and addition of fresh medium.

## Dark-induced senescence

4.5 week-old soil-grown plants were incubated in the dark—in a box with tubes allowing for gas exchange—for 9 days. Before and after incubation the true leaves of 4–8 representative plants were harvested, stuck on white paper with double sided tape, scanned with a flatbed scanner without color correction at a resolution of 600 dpi, and saved as TIFF files. Images were then imported in ImageJ (FIJI) (*Schindelin et al., 2012*) and the total and green leaf area of each leaf were quantified using the built-in threshold and color threshold function, respectively. The green leaf area in % was then calculated by dividing the green area of a leaf or the whole plant by the respective total area.

See *Figure 1—source data 1* for the ImageJ macro for semi-automatic image processing. A scheme of this method can be found in *Figure 1—figure supplement 2A*. Alternatively, chlorophyll content was determined according to *Porra et al. (1989)*. Total rosettes or seedlings were weighed and frozen in liquid nitrogen. Rosettes were crushed crudely with a spatula to increase extraction efficiency. Chlorophyll was extracted by adding 1–5 ml of a mixture of 80% Acetone and 20% 12.5 mM Hepes KOH pH 8.2 and incubation in the dark for at least 12 hr. Absorbance at 646.6 nm, 663.6 nm, and 750 nm was measured in a quartz cuvette and the chlorophyll content in µg chlorophyll/mg fresh weight was determined as follows: (12.5*(A663.6 − A750) − 2.55*(A646.6 − A750) + 20.31*(A646.6 − A750) − 4.91 (A663.5 − A750))/FW. For sugar rescue assays seeds were germinated on ½ MS agar plates containing 0.5% sucrose and grown in a 12 hr light/12 hr dark cycle for 12 days before transfer to ½ MS agar plates containing 0% or 2% glucose. After 6 additional days seedlings were either harvested or put in a dark box for 7 days. The chlorophyll content in µg/mg FW of individual seedlings was determined as described above.

## Metabolic profiling

Metabolites were extracted from leaves of 5 week-old plants, derivatized and measured as described in *Naegele et al. (2014)* with minor variations. Approximately 80 mg frozen and ground plant material were extracted with 1 ml –20°C cold MeOH:chloroform:H$_2$O (2.5:1:0.5) by mixing and incubation on ice. The supernatant after centrifugation was mixed with 400–500 µl H$_2$O and centrifuged. For the experiment shown in *Figure 2* (2) the polar phase was split into 2 aliquots, spiked with 1 µg C13 labeled Sorbitol (Campro Scientific, Berlin, Germany) and dried. For the experiment shown in *Figure 7* (7) the polar phase was not split and 2 µg of Sorbitol were added. For derivatization, metabolites were first dissolved in 10 µl (2) or 30 µl (7) pyridine containing 40 mg/ml methoxyamine hydrochloride by 90 min incubation at 30°C. Then, 40 µl (2) or 120 µl (7) of N-methyl-N-trimethylsilyltrifluoroacetamid (MSTFA; Macherey–Nagel, Düren, Germany), spiked with 60 µl/ml of an Alkane Standard Mixture C$_{10}$–C$_{40}$ (Fluka, Vienna, Austria), were added and the samples were incubated for 30 min at 37°C. GC–MS measurements were carried out on an Agilent 6890 gas chromatograph coupled to a LECO Pegasus 4D GCxGC-TOF mass spectrometer (LECO, USA). Injection volume was 1 µl. In the GC step, the initial oven temperature was 70°C, which was held for 1 min, followed by a 9°C/min temperature increase until the final temperature of 350°C was reached, which was held again for 8 min. Metabolites were measured in splitless mode (2 and 7) and alternatively also in split mode with a split ratio of 5 (7). In the MS step the data acquisition rate was set to 20 spectra/s, the detector voltage to 1550 V and the mass range to 40–600 m/z. Raw data were processed with the LECO Chroma-TOF software (LECO, USA). Peak areas were normalized to the area of the internal standard and to the fresh weight before statistical analysis. Outliers, as determined by Grubb's test, were removed from the dataset. For Hierarchical clustering, log-2 transformed fold change values were imported into MeV (MultiExperimentViewer, version 4.9.0, *Saeed et al., 2003*) and clustering was done using the standard settings with gene tree optimization, Pearson correlation, and average linkage clustering. For the PCA plot, normalized data were imported into R (RStudio, version 0.98.507), missing values

were replaced with the k nearest neighbor (knn) method using the 'impute.knn()' function and data were Z-transformed. PCA analysis was done using the 'prcomp()' function and scores for PC1 and PC2 were plotted against each other.

## Electrophoresis and western blotting

For 2D gels, proteins were extracted from 5–7 week-old soil-grown ox#3 plants which were grown in short day. 4 ml of 1× lambda phosphatase (λPP) buffer (NEB, Frankfurt am Main, Germany), including cOmplete protease inhibitor (Roche, Vienna, Austria), were added to 2 ml frozen and ground plant material, followed by vortexing and centrifugation. 0, 30 or 50 μg of λPP were added to 1.5 ml of the supernatant, followed by 15 min incubation at 30°C. Proteins were extracted with phenol (Carl Roth, Karlsruhe, Germany), precipitated with ammonium acetate and resuspended in 1× rehydration stock solution (7 M urea, 2 M thiourea, 2% (wt/vol) CHAPS, 2% IPG buffer (GE Healthcare, Vienna, Austria), 2.8 mg/ml DTT, Bromphenol blue). Protein extracts were then applied to 7 cm Immobiline DryStrips (GE Healthcare, Vienna, Austria) over night and separated by IEF on an IPGphor (GE Healthcare, Vienna, Austria) according to the manufacturer's instructions. For second dimension separation, the strip was incubated in SDS equilibration buffer (6 M urea, 75 mM TrisCl, 29.3% glycerol, 2% SDS, Bromphenol blue, pH 8.8) with 10 mg/ml DTT and then without DTT for 15 min each, followed by standard SDS PAGE and western blotting.

Phos-tag gel electrophoresis was done according to the manufacturer's instructions. Proteins were separated on SDS PAGE gels containing 8% SDS, 25 μM Phos-tag (WAKO, Neuss, Germany) and 50 μM $MnCl_2$ with an amperage of 15 mA/gel for 1.25 hr. Before semi dry blotting, the gels were incubated in transfer buffer containing 1 mM EDTA, followed by washing with transfer buffer without EDTA. Recombinantly expressed bZIP63-YFP was used as a size marker for the nonphosphorylated fusion protein. For the expression, bZIP63 was amplified from cDNA (see *Table 1* for primers) and cloned NcoI/NotI into pTWIN (NEB, Frankfurt am Main, Germany) containing YFP, to create a c-terminal YFP fusion. The protein was expressed in *Escherichia coli* (ER2566 strain) and purified according to the manufacturer's instructions. Total plant proteins were extracted with phenol. For light/extended night comparison, rosettes of 5 week old plants were collected after 6 hr of light or extended night. For ± sucrose comparison, seedlings were first germinated and grown for 1 week in liquid ½ Gamborg medium containing 0.5% sucrose, followed by 1 week in medium without sucrose. For treatment, at the onset of light 1% suc was added to half of the cultures and all cultures were kept in the dark for 6 additional hours. For Phos-tag gels from kinase assays, the reactions were mixed with 2× Laemmli sample buffer and loaded on the gel.

Western blotting was done by semi dry transfer onto a PVDF membrane, antibody incubation, and detection with an ECL kit following standard procedures. The following primary antibodies were used: Anti-GFP (Roche, Vienna, Austria; or ChromoTek, Munich, Germany), Anti-AKIN10 (Agrisera, Sweden), Anti-AMPKalpha-pT172 (Cell Signaling, Leiden, The Netherlands), Anti-Flag (Sigma–Aldrich, Vienna, Austria), Anti-HA High Affinity (Roche, Vienna, Austria), Anti-HA (Santacruz, Heidelberg, Germany), Anti-GAL4 DNA BD (Sigma–Aldrich, Vienna, Austria), polyclonal peptide antibodies against bZIP63: peptide antibodies against an N-terminal (EKVFSDEEISGNHHWSVNGM) and a C-terminal (SLEHLQKRIRSVGDQ) peptide were raised in rabbit and affinity purified (Davis bio-technology, Regensburg, Germany). The bZIP63 antibodies were tested on protein extracts from wt, bZIP63 ko and ox plants. While the antibodies recognized recombinant protein and bZIP63 in the ox extracts well, they failed to detect endogenous levels of bZIP63, possibly due to the low abundance of the protein (not shown). The following HRP-coupled secondary antibodies were used: Anti-mouse IgG, Anti-rabbit IgG, Anti-rat IgG (GE Healthcare, Vienna, Austria).

## Immunoprecipitation of bZIP63-GFP

To identify in vivo phosphorylation sites on bZIP63, bZIP63-GFP was immunoprecipitated from ox#3 seedlings grown on ½ MS agar plates or leaves of mature soil-grown ox#3 plants, harvested at different time points in the light cycle and in extended night (see *Figure 3—figure supplement 2A* for growth and harvesting conditions). In one experiment, leaves were infiltrated with $H_2O$ containing 100 μM of the proteasome inhibitor MG-132 (Calbiochem/Merck Millipore, Vienna, Austria) 6 hr before harvesting. Protein extracts were prepared by mixing frozen and ground plant material with an equal volume of cold extraction buffer (25 mM TrisCl, 10 mM $MgCl_2$, 15 mM EDTA, 150 mM NaCl,

1 mM DTT, 1 mM NaF, 0.5 mM $Na_3VO_4$, 15 mM β-glycerophosphate, 0.1% Tween20, Complete protease inhibitor, pH 7.5), followed by centrifugation. The supernatant was then incubated with protein A Sepharose CL-4B (GE Healthcare, Vienna, Austria), which had been pre-incubated for 2.5 hr in extraction buffer with Anti-GFP antibody (Roche, Vienna, Austria), or with GFP-Trap_A beads (ChromoTek, Munich, Germany) for 1 hr at 4°C. The beads were washed 2–3 times with extraction buffer and alternatively twice with wash buffer (50 mM TrisCl, 250 mM NaCl, 0.1% NP–40, 0.05% sodium deoxycholate, pH 7.5). Finally, the beads were resuspended in 1× Laemmli sample buffer, boiled for 5 min at 95°C, and centrifuged. The supernatant was separated by SDS PAGE and bands were excised for LC-MS/MS analysis.

## Identification of proteins and in vivo phosphorylation sites by LC-MS/MS

For the identification of phosphorylation sites, bZIP63-GFP was immunoprecipitated from leaves of ox#3 plants (see 'Immunoprecipitation of bZIP63-GFP'). For the identification of kinases, root protein extracts were affinity purified with recombinant GST-bZIP63 (see *Figure 4B* for a scheme and the methods section 'Kinase assays' for detailed description of the affinity purification).

Proteins were first separated by SDS PAGE. Bands of interest were then excised from Coomassie-stained gels and gel sections were chopped, washed with 50 mM ammonium bicarbonate (ABC, pH 8.5), and dried with acetonitrile (ACN). Disulfide bonds were reduced by incubating in 200 µl of 10 mM DTT for 30 min at 56°C. DTT was washed off and cysteines were alkylated by incubation with 100 µl of 54 mM iodoacetamide for 20 min at RT in the dark. Gel pieces were dried with ACN, then swollen in 10 ng/µl trypsin (recombinant, proteomics grade, Roche, Vienna, Austria) in 50 mM ABC and incubated over night at 37°C. For higher sequence coverage of bZIP63 alternative proteases were used: LysC (MS grade, WAKO, Neuss, Germany) at 37°C overnight, subtilisin (Sigma–Aldrich, Vienna, Austria) at 37°C for 0.5–2 hr, chymotrypsin (sequencing grade, Roche, Vienna, Austria) at 25°C for 4 hr. Digestion was stopped by adding formic acid to a final concentration of approximately 1% and peptides were extracted by sonication. Peptides were separated on an UltiMate 3000 HPLC system or on a U3000 nano HPLC (both Dionex, Thermo Fisher Scientific). Digests were loaded on a trapping column (PepMap C18, 5 µm particle size, 300 µm i.d. × 5 mm, Thermo Fisher Scientific), equilibrated with 0.1% trifluoroacetic acid (TFA), and separated on an analytical column (PepMap C18, 3 µm, 75 µm i.d. × 150 mm, Thermo Fisher Scientific) by applying a 60 min linear gradient from 2.5% up to 40% ACN with 0.1% formic acid, followed by a washing step with 80% ACN and 10% trifluoroethanol (TFE) on the U3000 HPLC. The UltiMate 3000 HPLC was directly coupled to a linear ion trap (LTQ, Thermo Fisher Scientific), which was operated in a data-dependent MS3 method for the phosphorylation analysis. One full scan (m/z: 450–1600) was followed by maximal 4 MS/MS scans. If in the MS/MS scan a fragment corresponding to a neutral loss from the precursor of 98, 49, or 32 Th was observed among the top 8 peaks, a MS3 scan was triggered. Fragmentation energy was set at 35%, Q-value at 0.25, and the activation time at 30 ms. High resolution measurements were acquired on an LTQ-Orbitrap Velos mass spectrometer (Thermo Fisher Scientific), equipped with a nanoelectrospray ionization source (Proxeon, Thermo Fisher Scientific). The electrospray voltage was set to 1500 V. The mass spectrometer was operated in the data-dependent mode: 1 full scan (m/z: 350–1800, resolution 60,000) with lock mass enabled was followed by maximal 12 MS/MS scans. The lock mass was set at the signal of polydimethylcyclosiloxane at m/z 445.120025. Monoisotopic precursor selection was on, precursors with charge state 1 were excluded from fragmentation. The collision energy was set at 35%, Q-value at 0.25, and the activation time at 10 ms. Fragmented ions were set onto an exclusion list for 60 s. When ETD (electron transfer dissociation) was applied, the top 6 peaks from the full scan where fragmented with CID (collision-induced dissociation) and subsequently with ETD. For ETD, the energy parameters were as for the CID except the activation time was set to 80 or 120 ms.

Data interpretation: raw spectra for the kinase identification were interpreted by Mascot 2.2.04 (Matrix Science) using Mascot Daemon 2.2.2. Spectra were searched against the *Arabidopsis thaliana* entries in the nr-database with the following parameters: the peptide tolerance was set to 2 Da, MS/MS tolerance was set to 0.8 Da, carbamidomethylcysteine was set as a static modification, oxidation of Met as variable modification. Trypsin was selected as the protease allowing two missed cleavages. Mascot score cut-off was set to 30, except for the low abundance sample 1, where the cut-off was set to 20. For the phosphorylation analysis of purified bZIP63, either Mascot or Sequest (Proteome Discoverer 1.2; Thermo Scientific) were used. The search was extended to the phosphorylation of Ser,

Thr, and Tyr. High resolution data were searched with 3 ppm precursor mass tolerance. Proteolytic specificity was defined according to the digest. Results were manually validated including comparison of the fragmentation pattern and the relative retention of the nonphosphorylated counterpart. Site localization was checked by manual inspection at the spectrum level in the first place and was confirmed by the site-localization algorithm PhosphoRS (*Taus et al., 2011*).

## Kinase assays

In vitro kinase assays were performed with GST-tagged recombinant proteins. The cDNA of *bZIP63.2*, *bZIP1*, *bZIP2*, *bZIP11*, *bZIP44.1*, *bZIP55*, *AKIN10.1/3*, and *AKIN11.1/2* was obtained by PCR (see *Table 1* for primers) and cloned ApaI/NotI or NcoI/NotI into pGEX-4T. *SnAK2* was in pDEST15 (*Crozet et al., 2010*). An inactive version (K/M) of *AKIN10* and non-phosphorylatable (S/A) versions of *bZIP63* were created by mutagenesis PCR using the primers listed in *Table 1*. The proteins were expressed in *E. coli* (ER2566 or BL21 strain), purified using Glutathione Sepharose 4B (GE Healthcare, Vienna, Austria) according to the manufacturer's instructions, and stored at −80°C in GST elution buffer containing 10–25% glycerol.

Kinase assays were performed by incubating the kinase and substrate for 20–30 min in kinase reaction buffer (20 mM Hepes, 20 mM MgCl$_2$, 100 µM EGTA, 1 mM DTT, 50 µM ATP, pH 7.5) at room temperature. For radioactive assays, 1 µCi γ-$^{32}$P-labeled ATP (NEN/PerkinElmer, Waltham, MA, USA) was added in each reaction. The reactions were then separated by SDS PAGE and exposure on a Storage Phosphor Screen (GE Healthcare, Vienna, Austria) or Phos-tag gel electrophoresis and Western blotting, respectively.

For in-gel kinase assays, bZIP63 with an N-terminal 6xHis tag was used as a substrate. To construct the expression vector, bZIP63.2 was amplified from cDNA (see *Table 1*) and first cloned ApaI/NotI into pRSETa-QM (Invitrogen, Germany). From there the expression cassette was excised by SalI digest, followed by a fill in with DNA Polymerase I (Klenow fragment, NEB, Frankfurt am Main, Germany) and XbaI digest. The cassette was ligated with pTXB3 (NEB, Frankfurt am Main, Germany), which was first digested with BamHI, followed by a fill in and XbaI digest. The protein was expressed in *E. coli* (ER2566 strain) and purified over a HiTrap column (GE Healthcare, Vienna, Austria) according to the manufacturer's instructions. Plant protein extracts for the in-gel kinase assays were made from different plant material. For the identification of bZIP63 kinases, proteins were extracted either from roots of 8 week-old plants that were grown in hydroponic culture in short day and collected in the light phase (shown in *Figure 4A,C*), from root cell suspension culture, or from seedlings grown on ½ MS agar plates, which were harvested in the dark phase. 4 independent experiments were conducted. For specifications on the plant material used in each of the experiments and the kinases identified please refer to *Figure 4—source data 2*. For the comparison of bZIP63 phosphorylation with wt and *akin10* plant extracts, roots and leaves of 2 week-old seedlings grown in liquid culture in a 12 hr light/12 hr dark cycle and collected after 4 hr of extended night were used. Extraction was done by mixing the frozen and ground plant material with an equal volume of cold protein extraction buffer (25 mM TrisCl, 15 mM EGTA, 10 mM MgCl$_2$, 75 mM NaCl, 1 mM NaF, 0.5 mM NaVO$_3$, 15 mM beta-glycerophosphate, 0.1% Tween20, 1 mM DTT, Complete protease inhibitor, pH 7.5), followed by centrifugation. Protein amounts were determined by Bradford assay. For affinity purification of bZIP63-binding proteins, GST-tagged bZIP63 was expressed in *E. coli* and the cell lysate of up to 1l culture in GST binding buffer (50 mM TrisCl, 20 mM MgSO$_4$, 2 mM DTT, 5 mM EDTA, 0.5% Tween20, pH 8) was loaded on an equilibrated GSTrap FF column (GE Healthcare, Vienna, Austria). The column was then washed with 5 ml cold GST binding buffer and protein extraction buffer, respectively, and 2–5 ml of total root protein in cold protein extraction buffer were loaded. Subsequently, proteins were eluted from the column by repeated washing with 5 ml cold protein extraction buffer with increasing salt concentrations, and concentrated with Amicon Ultra Centrifugal Filter Units (Millipore). 8–20 µg total protein extracts and up to 40 µg affinity purified proteins were loaded on standard SDS PAGE gels containing 1 mg/ml substrate (6× His-bZIP63) in the separating gel and run at 4°C with 20 mA per gel. The gel was then incubated 3 times for 20 min, respectively, at room temperature in each of the following buffers: wash buffer I (50 mM TrisCl, 20% isopropanol, pH 8), wash buffer II (50 mM TrisCl, 1 mM DTT, pH 8), and denaturation buffer (50 mM TrisCl, 1 mM DTT, 6 M guanidinium HCl, pH 8). Subsequently, the gel was incubated in renaturation buffer (50 mM TrisCl, 0.05% Tween 20, pH 8) at 4°C for 12–18 hr. In this period, the buffer was exchanged 10 times after at least 30 min. After

renaturation, the gel was incubated 2 times for 30 min, respectively, in kinase buffer (20 mM HEPES, 20 mM MgCl$_2$, 50 µM CaCl$_2$ or 500 µM EGTA, 1 mM DTT, 0.05% Tween 20, pH 7.5), followed by 30 min incubation in kinase reaction solution (kinase buffer containing 50 µM ATP and 100 µCi γ-$^{32}$P-labeled ATP). Finally, the gel was washed 2 times for 15 min, respectively, in 5% TCA and several times in 5% TCA containing 1% sodium pyrophosphate, until the wash solution was only weakly radioactive. The dried gels were exposed on a Storage Phosphor Screen and the signal was recorded on a Typhoon 8600 (GE Healthcare, Vienna, Austria).

## Y2H assay

*AKIN10.1/3*, *AKIN11.1/2*, *AKINβ1*, *AKINβ2*, *SNF4*, and *bZIP63.2* were amplified from cDNA (see *Table 1* for primers) and cloned ApaI/NotI or NcoI/NotI into pBTM117 and *bZIP63.2* was cloned into pACTIIJ to generate N-terminal fusions with the LexA-BD and the GAL4-AD, respectively. The yeast strain L40 (MATα *hisΔ200 trp1-900 leu2-3.112 ade*2 LYS2::(*lexA op*)$_4$HIS3 URA3::(*lexA op*)$_8$lacZ Gal4 gal80) was transformed with empty pACTIIJ or *bZIP63.2*-containing pACTIIJ in combination with different pBTM117 vectors. Freshly grown L40 was mixed gently with 1 ml transformation mix (800 µl 50% PEG 3600, 100 µl 2 M LiAc, 100 µl 1 M DTT, 10 µl bacterial RNA [10 µg/µl]), to get a cloudy suspension. 2.5 µg of each plasmid were added to 125 µl transformation mix, followed by 20 min incubation at 30°C and 44°C, respectively, addition of 1 ml H2O, and 1 min centrifugation at 3500×*g*. The cells were resuspended in a small volume and plated on SD medium without Leu and Trp to select for successful transformation. Single colonies were inoculated in SD medium without Leu and Trp and proteins were extracted from 2 ml of an over-night culture. The yeast cells were resuspended in 200 µl enzyme lysis buffer (25 mM TrisCl, 20 mM NaCl, 8 mM MgCl$_2$, 5 mM DTT, 0.1% NP-40, pH 7.5), 200 µl glass beads (0.4–0.6 mm diameter) were added, the cells were frozen in N$_2$, thawed, and broken by vigorous shaking on a Vibrax at 4°C for 20 min. The supernatant after 10 min centrifugation was transferred to a fresh tube and kept on ice. The protein concentration of the extract was determined by Bradford assay. The GUS activity was determined by mixing 50 µl of the extract with 650 µl Z-buffer (60 mM Na$_2$HPO$_4$, 40 mM NaH$_2$PO$_4$, 10 mM KCl, 10 mM MgSO$_4$, 0.25% beta mercaptoethanol) and 100 µl ONPG (4 mg/ml), and incubating for up to 10 min at room temperature. The reaction was stopped by adding 400 µl 1 M Na$_2$CO$_3$ and the extinction at 420 nm was measured on a photometer. The GUS activity in U/mg protein was calculated as follows: (A420 × 24 × 1000)/(45 × incubation time [min] × protein concentration [mg/ml]).

## BiFC and multi-color BiFC

For BiFC and multi-color BiFC interaction studies in *N. tabacum*, *bZIP63.2*, *AKIN10.1/3*, *AKIN11.1/2* and *SNF4* were cloned into modified pBIN19 vectors, containing either the N- or C-terminal moieties of the split CFP system as N-terminal fusions as described in *Waadt et al. (2008)* (see *Figure 5D* for scheme). To generate BiFC plasmids for ApaI/NotI and NcoI/NotI cloning, CPK3, harboring XbaI, NcoI, and ApaI (for plasmids with MCS [multi cloning site] 1) or ApaI and NcoI (for plasmids with MCS 2) in the terminus and NotI and SmaI in the C-terminus was generated by PCR from cDNA (see *Table 1* for primers). It was then cloned XbaI/SmaI or ApaI/SmaI into the different BiFC casettes in pUC19 from *Waadt et al. (2008)*. The cassettes were then cloned into pBIN19 with HindIII/EcoRI and CPK3 was exchanged for other genes using ApaI and NotI. *A. tumefaciens* (AGL1 strain) was transformed with the resulting plasmids by electroporation and further used for transient transformation of tobacco leaf epidermis. For transformation, 5 ml agrobacterium overnight cultures were filled to 50 ml with fresh LB medium containing 50 µg/ml Kanamycin and 10 µg/ml Gentamicin and grown for 4 hr at 30°C. The cells were pelleted by centrifugation at 3500×*g*, resuspended in LB containing 150 µM acetosyringone, and grown for another 2 hr. The cultures were then pelleted again and resuspended in 5% sucrose solution to reach a final OD600 of 2. For co-infiltration, equal volumes of agrobacteria suspensions, containing the respective constructs, were mixed and infiltrated into the leaves of 5 week-old plants. 48 hr after infiltration equally sized leaf sections were analyzed for their CFP fluorescence signal with an LSM510 confocal laser scanning microscope (Zeiss) and the corresponding ZEN software (Zeiss). The same settings were used for each construct to allow comparison of the signal intensities. To verify that the fusion proteins were expressed and to determine the relative amount of each interaction partner, proteins were extracted from the leaf sections used for

microscopy and subjected to Western blot analysis with antibodies against the Flag (N-terminal CFP moiety) and HA (C-terminal CFP moiety) epitopes.

Multi-color BiFC was done as described in *Waadt et al. (2008)* with some modifications. bZIP11 fused to the N-terminal moiety of CFP (CFP-N), bZIP1 fused to the N-terminal moiety of VENUS (VENUS-N), and bZIP63 (wt or S29/294/300A) fused to the C-terminal moiety of CFP (CFP-C) were expressed from one plasmid to have co-expression at equal amounts in all transformed cells (for a scheme of the construct see *Figure 9A*). To generate the construct, bZIP1 and bZIP11 were first amplified from cDNA by PCR (see *Table 1* for primers) and cloned ApaI/NotI into pBIN19 containing a c-terminal fusion of CFP-C and CFP-N, respectively (plasmids described above). From there, 35S::bZIP1 was cloned HindIII/NotI into pUC19 containing CPK3 with a c-terminal VENUS-N fusion (described above) to exchange the shorter 35S primer in the VENUS construct. Then, the two cassettes including the 35S promoter, bZIP1 or 11 and the VENUS-N or CFP-N tag were amplified by PCR introducing KpnI/HindII in the front and BamHI in the back of the bZIP1 cassette and BamHI in the front and HindIII in the back of the bZIP11 cassette (see *Table 1* for primers). The PCR products were ligated into pCRBlunt (Invitrogen, Germany) and the bZIP1 cassette was then cloned KpnI/BamHI in front of the bZIP11 cassette. Finally, the combined cassette with bZIP1 and 11 was cloned HindII into pBIN19 containing bZIP63 wt or S29/294/300A (mutated by PCR) with an n-terminal CFP-C fusion. AKIN10 was amplified from cDNA by PCR (see *Table 1* for primers) and cloned ApaI/NotI into pBIN19 containing a c-terminal mCherry tag. *A. tumefaciens* (AGL1 strain) was transformed with the resulting plasmids by electroporation and further used for transient transformation of tobacco (*N. tabacum*) leaf *epidermis* as described above. 48 hr after infiltration equally sized leaf sections were analyzed for their VENUS, CFP, and mCherry fluorescence signal with a TCS SP5 DM-6000 confocal laser scanning microscope (Leica) and the corresponding Leica software. CFP, VENUS, and mCherry were excited with 458 nm (Ar laser), 514 nm (Ar laser), and 574 nm (white light laser), respectively. The fluorescence was detected at 461–495 nm, 520–550 nm, and 600–615 nm, respectively. To avoid bleed-through of the CFP signal into the VENUS channel CFP and VENUS were excited and detected separately. Pictures were taken with a 20× air objective and the same settings for both combinations (wt bZIP63 and S29/294/300A). The pinhole was set to 1.5. To determine the relative VENUS/CFP signal ratio, images were processed in the LasX software from Leica. Nuclei showing a CFP/VENUS as well as mCherry signal were selected and the signal intensity was determined with the histogram tool. For each nucleus a ratio of the background corrected pixel sum from VENUS and CFP was built. All values for one leaf were divided by the median of the ratio from the bZIP63 S29/294/300A construct to allow comparison between experiments. In total, six leaves from two infiltrations were analyzed with similar results.

## Virus-induced gene silencing (VIGS)

2 week-old GY9 plants were infiltrated (*akin10/11*) or not (control) with an *AKIN10-AKIN11* silencing construct according to the method described by *Burch-Smith et al. (2006)*. For the *AKIN10-AKIN11* construct two gene-specific fragments corresponding to *AKIN10* and *AKIN11* were cloned in tandem into the TRV-based vector pYL156a. First, a 480 bp *AKIN10* fragment was amplified from cDNA using the *AKIN10 VIGS* primers (see *Table 1* for primers) and cloned into the pYL156/pTRV2 vector using EcoRI/KpnI. A 503-bp *AKIN11* fragment (*Baena-González et al., 2007*) was thereafter amplified from cDNA using the *AKIN11 VIGS* primers (see primer list) and cloned into the pYL156/pTRV2 vector harboring the KIN10 fragment using XhoI/KpnI. 2 weeks after infiltration the *AKIN10-AKIN11* knock-down plants displayed visible growth defects and anthocyanin accumulation. Silencing of AKIN10 and AKIN11 in these plants was confirmed by western blotting with an antibody against phosphorylated AMPK alpha (Anti-AMPKalpha-pT172) before use in Phos-tag gels (see *Figure 5—figure supplement 7*). The VIGS experiments were repeated 4 times and 3 biological replicates were analyzed in each experiment with consistent results.

## Protoplast transformation for P2H and promoter activation assays

Protoplast were obtained from 3 week-old soil-grown wt *Arabidopsis* plants and transformed according to the guide method (*Yoo et al., 2007*) with small modifications. Leaves were harvested 1 hr after onset of the light phase, cut into tiny stripes, and digested for 30 min under vacuum and 3 hr at atmospheric pressure, respectively, with enzyme solution (1.25% [wt/vol] Cellulase R-10, 0.3% Macerozyme R-10, 400 mM mannitol, 20 mM KCl, 10 mM CaCl$_2$, 20 mM MES, pH 5.7). The protoplast

suspension was filtered on a metal net to remove leaf debris and washed twice with 10 ml of W5 solution (2 mM MES, 154 mM NaCl, 125 mM CaCl$_2$, 5 mM KCl, pH 5.7). Protoplasts were resuspended in 10 ml of W5, incubated on ice for at least 1 hr and subsequently resuspended to a final concentration of 1 × 10⁵ cells/ml in MMg buffer (4 mM MES, 400 mM mannitol, 15 mM MgCl$_2$, pH 5.7). Protoplasts were then co-transformed with 10 µg each of up to three effector plasmids, 7 µg of a reporter plasmid, and 3 µg of a *35S::NAN* transfection control reporter plasmid (*Kirby and Kavanagh, 2002*) for normalization. For P2H assays, the effector plasmids were pHBTL containing *bZIP63.2*, *bZIP1*, or *bZIP11* with an N-terminal GAL4-AD or GAL4-BD fusion (described in detail in *Ehlert et al., 2006*) and alternatively pHBTL containing *AKIN10.1/3*. The reporter plasmid contained *GAL-UAS₄::GUS* (beta galactosidase) (*Ehlert et al., 2006*). For promoter activation assays, effector plasmids were pHBTL containing HA-tagged *bZIP63.2* or *AKIN10.1/3*. pBT10 containing *proASN1:: GUS* or *proProDH::GUS* (*Dietrich et al., 2011*) was used as a reporter plasmid. For transformation, 200 µl of the protoplast suspension were gently mixed with the DNA and 220 µl of PEG (40% PEG 4000, 200 mM mannitol, 100 mM CaCl$_2$) were added, followed by gentle mixing, and 10 min incubation at room temperature. The protoplasts were then washed by addition of 800 µl W5 and 1 min centrifugation at 300×*g*. The supernatant was removed and the protoplasts were incubated for 16 hr in 200 µl of WI solution (4 mM MES, 500 mM mannitol, 20 mM KCl, pH 5.7) in the growth chamber in order to not affect their diurnal circle. GUS and NAN enzyme assays were performed according to *Kirby and Kavanagh (2002)* and the relative activity of GUS and calculated as GUS/NAN. The expression of the effector constructs was confirmed by Western blot analysis.

## Sequence alignment of bZIP63 homologues

Homologues of bZIP63 in 8 plant species were identified by blasting the protein sequence of bZIP63.2 on the Phytozome webpage (http://phytozome.net). The protein sequences were aligned with ClustalΩ (http://www.ebi.ac.uk/Tools/msa/clustalo/) using the default settings and the Clustal output file was imported into GeneDoc (http://www.psc.edu/biomed/genedoc) for visualization and minor adjustments of the alignment. The identity matrix for the alignment was calculated using the PID3 method in the SIAS webmask (http://imed.med.ucm.es/Tools/sias.html).

Gene identifiers of the *bZIP63* homologues (numbers are the Phytozome10 references): *Medicago truncatula* (Medtr7g115120), *Glycine max* (Glyma.10G162100), *Populus trichocarpa* (Potri.013G040700), *Vitis vinifera* (GSVIVG0102179000), *Zea mays* (GRMZM2G007063), *Oryza sativa* (LOC_Os03g58250), *Selaginella moellendorffii* (270282), *Physcomitrella patens* (Phpat.009G02690).

## RT-qPCR

RNA was extracted with phenol from total rosettes of 4.5 week-old soil-grown plants at the indicated time points. 500 µl RNA extraction buffer (1% SDS, 10 mM EDTA, 200 mM NaAc, pH 5.2) and 500 µl acidic phenol (pH 4, Carl Roth, Karlsruhe, Germany) were added to 100 mg frozen and ground plant material, followed by 1 min vortexing and 10 min centrifugation. The aqueous phase was extracted twice with an equal volume of PCI (Phenol:Chloroform:Isoamylalcohol (25:24:21), Carl Roth, Karlsruhe, Germany) by vortexing for 30 s and 2 min centrifugation, and twice with chloroform. Then, the RNA was precipitated by adding ⅓ volume 10 M LiCl and incubating at 4°C for at least 2 hr, followed by 15 min centrifugation at 4°C. The pellet was washed once with 2.5 M LiCl and then with 70% EtOH, dried and resuspended in 50 µl RNAse free H$_2$O. 6.5 µg RNA were treated with 2u of RQ1 DNAse (Promega, Mannheim, Germany), according to the manufacturer's instructions and precipitated again for at least 2 hr with ⅓ volume of 10 M LiCl to remove the DNAse from the solution. The solution was then centrifuged for 15 min, washed once with 70% EtOH, dried and resuspended in RNAse free H$_2$O. 1.5 µg RNA were reverse transcribed using M-MLV RT [H-] Point Mutant (Promega, Mannheim, Germany) according to the manufacturer's instructions and diluted 1:4 with RNAse free H2O. 2 µl of cDNA (15 ng/µl) or H$_2$O (for the no template control) were added to 18 µl of a PCR reaction mix containing 0.25u DreamTaq polymerase (Thermo Scientific, Vienna, Austria), 1× Dream Taq buffer, additional 3 mM MgCl$_2$, 100 µM dNTPs, 400 nM of each primer, and 0.4× SYBR Green I nucleic acid gel stain (Sigma–Aldrich, Vienna, Austria). The qPCR was performed on a Mastercycler ep realplex2 (Eppendorf) in white 96-well twin.tec real-time PCR plates (Eppendorf, Vienna, Austria) with the following program: 1 cycle with 3 min at 95°C, followed by 40 cycles with 20 s at 95°C, 20 s at 59/60°C, and 12 s at 72°C. 2–3 technical replicates were used. A 20 min melting curve was added in the end. For

data evaluation, raw data were imported into LinRegPCR (version 2014.02, Ramakers et al., 2003), where the PCR efficiency was checked and the N0 (staring concentration of cDNA) was calculated with a common baseline setting for each primer pair. Samples excluded by LinReqPCR were not used. In case only two technical replicates were used, samples for which the dCq between the two technical replicates was bigger than 0.5 were also excluded from the calculations. The mean N0 of the technical replicates was calculated for each sample and primer pair. The resulting mean N0 of the test genes was then normalized by dividing by the mean N0 of the reference genes. For qPCR of *AKIN10*, *AKIN11* (*Figure 5—figure supplement 1D*); *bZIP63*, *bZIP1*, *bZIP11* (*Figure 5—figure supplement 3*); *ASN1*, *DIN10*, and *ProDH* (*Figure 7G*) TBP2 (Tata binding protein 2) was used as a reference gene. For pPCR of *bZIP63*, (*Figure 1—figure supplement 1D,E*, *Figure 7—figure supplement 1A*); *CAB2*, *SEN1*, *SAG103*, and *YLS3* (*Figure 1—figure supplement 3B*); *ASN1*, *ProDH*, *BCAT2*, *AA-TP family protein*, and *DIN10* (*Figure 5—figure supplement 2C*) the geometric mean of the values for *TBP2* and *MHF15* (Thioredoxin superfamily protein) were used. The normalized N0 values were then used to calculate the mean and SD. Outliers were determined with the Grubb's test and removed.

## Statistical tests

Statistical tests were performed in excel or R. Equality of variances was tested with Levene's test. This was followed by unpaired t-tests or multivariate statistics. In case of equal variance ANOVA followed by Bonferroni or TukeyHSD corrected pairwise comparison was chosen. In case of unequal variance Welch corrected ANOVA was applied, followed by Games Howell test of all samples or pairwise comparison of samples with equal variance.

## Gene identifiers of *Arabidopsis thaliana* genes

*bZIP63* (At5g28770), *bZIP1* (At5g49450), *bZIP11* (At4g34590), *bZIP2* (At2g18160), *bZIP44* (At1g75390), *bZIP53* (At3g62420), *AKIN10/SnRK1.1* (At3g01090), *AKIN11/SnRK1.2* (At3g29160), *AKINβ1* (At5g21170), *AKINβ2* (At4g16360), *SNF4/AKINβγ* (At1g09020), *SnAK2/GRIK1* (At3g45240), *ASN1/DIN6* (At3g47340), *DIN10* (At5g20250), *ProDH/ERD5* (At3g30775), *BCAT2* (AT1G10070), *AA-TP* (amino acid transporter) *family protein* (AT2G40420), *TBP2* (At1g55520), *MHF15* (At5g06430), *CAB2* (At1g29920), *SEN1/DIN1* (At4g35770), *SAG103* (At1g10140), *YLS3* (At2g44290).

## Acknowledgements

We thank Lena Fragner (University of Vienna) for help with the metabolomics measurements and Klaus Harter (University of Tübingen) and Sjef Smeekens (Utrecht University) for critical comments on the manuscript. We further thank the Core Facility Cell Imaging and Ultrastructure Research (CIUS, University of Vienna) for providing access and support to the Leica confocal laser scanning microscope. This work was funded by the FP7 Marie Curie ITN MERIT (GA 264474) (LP, AS, TN), the Austrian Science Fund (FWF) projects AP19825 (AM, BW, MT) and AP23435 (AM, MT), the Deutsche Forschungsgemeinschaft (DFG) project HA2146/8-2 (CC), and the Fundação para a Ciência e a Tecnologia projects PTDC/BIA-PLA/3937/2012 and UID/Multi/04551/2013 (EBG).

## Additional information

### Funding

| Funder | Grant reference | Author |
|---|---|---|
| Austrian Science Fund (FWF) | P23435 | Andrea Mair, Bernhard Wurzinger, Markus Teige |
| European Commission | ITN MERIT GA 264474 | Lorenzo Pedrotti, Andrea Simeunovic, Thomas Nägele |
| Deutsche Forschungsgemeinschaft | HA2146/8-2 | Tobias Kirchler |
| Fundação para a Ciência e a Tecnologia (Portuguese Science and Technology Foundation) | PTDC/BIA-PLA/3937/2012 | Concetta Valerio |

| Funder | Grant reference | Author |
|---|---|---|
| Fundação para a Ciência e a Tecnologia (Portuguese Science and Technology Foundation) | UID/Multi/04551⁄2013 | Elena Baena-González |

The funders had no role in study design, data collection and interpretation, or the decision to submit the work for publication.

### Author contributions

AM, Conception and design, Acquisition of data, Analysis and interpretation of data, Drafting or revising the article; LP, BW, DA, AS, CW, CV, KD, TK, TN, Acquisition of data, Analysis and interpretation of data; JVC, JH, Analysis and interpretation of data, Contributed unpublished essential data or reagents; EB-G, Drafting or revising the article, Contributed unpublished essential data or reagents; CC, Analysis and interpretation of data, Drafting or revising the article, Contributed unpublished essential data or reagents; WW, Analysis and interpretation of data, Drafting or revising the article; WD-L, MT, Conception and design, Analysis and interpretation of data, Drafting or revising the article

### Author ORCIDs

Jesús Vicente Carbajosa, http://orcid.org/0000-0002-6332-1712
Markus Teige, http://orcid.org/0000-0001-7204-1379

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
