## [Decision Letter]

[Editors’ note: this article was originally rejected after discussions between the reviewers, but the authors were invited to resubmit after an appeal against the decision.]

Thank you for choosing to send your work entitled “SnRK1-triggered switch of bZIP63 dimerization mediates the low-energy response in plants” for consideration at *eLife*. Your full submission has been evaluated by Ian Baldwin (Senior Editor), a Reviewing Editor and three peer reviewers, and the decision was reached after discussions between the reviewers. Based on our discussions and the individual reviews below, we regret to inform you that your work will not be considered further for publication in *eLife*.

While the reviews acknowledge the overall quality of your work, questions remain about the validity of one of the major tools used in your study (the *akin10* mutant). As the use of this mutant is crucial for major conclusions of your work, comprehensive characterization of this mutant is required before further consideration of your submission. We would welcome re-reviewing this manuscript if the major concerns listed in the reviews below were convincingly addressed.

Reviewer #1:

Plant SnRK1 kinases are structural and functional homologs of yeast Snf1 and animal AMP-activated kinases that play a central role in sensing energy depletion and maintaining cellular energy homeostasis. In this paper, a multinational consortium set out to demonstrate that *Arabidopsis* SnRK1, carrying the AKIN10 catalytic subunit, phosphorylates in vivo the C-group bZIP63 transcription factor. bZIP63 has been identified to control transcription of some SnRK1-regulated target genes, involving ASPARGINE SYNTHASE 1 (*ASN1*/*DIN6*), PROLINE DEHYDROGENASE (*PDH1*/*PRODH*), RAFFINOSE SYNTHASE (*DIN10*) and SENESCENCE 1 (*SEN1*). To justify the statements in the Abstract and Introduction that bZIP63 is a key regulator of starvation response, the paper starts with limited characterization of a *bzip63* T-DNA insertion mutant, and two bZIP63-ox overexpression lines.

The authors use a phenotypic assay by placing plants for 9 days into the dark and monitoring rosette leaf senescence/bleaching by ImageJ scanning of green versus bleached/yellow leaf areas. Whereas in wild type (wt) the first two leaves (numbered 3 and 4 in Figure 1) show bleaching, they remain greenish in the *bzip63* mutant (defined as stay-green phenotype), and 3-5 more leaves undergo bleaching in the bZIP63-ox plants, suggesting enhanced senescence response.

When grown in the light, the *bzip63* and bZIP63-ox plants do not differ from wt.

Next, the authors compare metabolites entering or derived from glycolysis and citrate cycle in light-grown (L) and dark exposed (elongated night, EN) plants and find that the levels of “most” free amino acids is increased in the *bzip63* mutant and decreased in bZIP63-ox plants, whereas sugar levels are stated to show an opposite trend of changes. Based on these data the authors conclude that bZIP63 is a key regulator of starvation responses (see Abstract and Introduction).

When inspecting the metabolite profiling data, it is clear that the discussion of the results is rather superficial. For example, raffinose levels are decreased, whereas trehalose levels are increased in both mutant and ox lines, similarly to glucose in the dark. In general, we do not learn much from the metabolomics data. The actual conclusion is that “misregulation of bZIP63” (in the subsection “bZIP63 controls dark induced senescence and primary metabolism”) has an effect on metabolism.

On the other hand, we do not find any additional data (such as quantitative measurement of chlorophyll levels, degradation products, or other senescence markers, such as enzymes or transcripts), which would provide a further insight documenting how alteration of bZIP63 levels modulate the starvation-induced and/or senescence responses. Whether 9 days of dark treatment can be considered as starvation or general stress response remains a matter of discussion.

By performing 2D and Phos-tag gel westerns (Figure 2), the authors detect multiple phosphorylated (lambda phosphatase sensitive) isoforms of bZIP63-GFP immunoprecipitated from overexpressing plants. On the Phos-tag gel, they show a rather weak band termed “supershift” of bZIP63, which appears in dark-treated plants, and detect somewhat weaker bZIP63-GFP bands with altered pattern plants grown with 1% sucrose. They use the latter result, without explicit explanation or demonstration, to suggest that the observed phosphorylation changes reflect inhibition of SnRK1 by 1% sucrose.

By mass spectrometry analysis, seven phosphorylated residues were then identified in the immonoprecipitated bZIP63-GFP protein. Subsequent in-gel kinase tests indicate that bZIP63 is phosphorylated by multiple protein kinases showing apparent molecular masses of 50-60 and 35-38 kDa. In Figure 2, it should be clearly indicated that the plant extract used for in-gel kinase assays derived from hydroponic root cultures (see “Kinase assays”, third paragraph), along with the conditions used for root culture in the case of the experiment in Figure 2.

To isolate the kinases detected in the in-gel assay, GST-bZIP63 was immobilized on GST-Sepharose and plant protein extracts made from root cultures or root-derived cell suspension (as the latter is maintained in the presence of sucrose, please specify which of these sample is shown in Figure 2) were subjected to affinity binding followed by NaCl elution. The eluted proteins resolved between 30 and 60 kDa were isolated from in-gel kinase PAGE (if I understood well), and identified by mass spectrometry. From 28 candidates only 7 kinases (including the SnRK1 catalytic AKIN10, AKIN11 and activating subunit SNF4, as well as casein kinase subunits CKA1, CKA2, CKB1 and CKL2) were identified by 2-9 peptides. It should be noted in the Materials and methods how the excess of His6-bZIP63 was filtrated out during the mass spectrometry analysis.

In addition, it would be worth knowing whether the GST-alone matrix had retained any of these kinases.

Furthermore, αAKIN10 western showing an enrichment of AKIN10 on the GST-bZIP63 matrix versus GST would be more convincing.

From here on, the authors concentrate on the analysis of bZIP63 phosphorylation by the SnRK1 catalytic subunits AKIN10, which according to Baena-Gonzalez et al. and Jossier et al. (Plant J 59: 316, 2009), is at least 10-20 times more active/abundant compared to its homolog AKIN11 in *Arabidopsis*. Figure 3 illustrates that GST-AKIN11 purified from *E. coli* also phosphorylates bZIP63 less efficiently compared to GST-AKIN10 in vitro. It is strange that no autophosphorylation of AKIN10 was ever detected by the authors either in normal or in-gel kinase assays. Any explanation?

As a key argument, the authors use the *akin10* GABI_579E09 T-DNA insertion mutant to demonstrate that AKIN10 is indeed responsible for in vivo phosphorylation of bZIP63. Unfortunately, according to the reviewer's knowledge, this mutant has not yet been characterized by others, so, to a certain extent, this must be performed by the authors in this paper.

In Figure 4, they show that one of the phosphorylated bZIP63 bands around 55-57 kDa is missing in their in-gel kinase assay with *akin10* mutant extracts, and they fail to detect AKIN10 by western with the Agrisera αAKIN10 antibody in roots, but observe a lower Mw weakly cross-reacting protein band in leaf extracts. According to these results, the GABI_579E09 insertion mutant appears to produce no immunologically detectable AKIN10. Yet, the authors could successfully cultivate the mutant, which means that it must be fertile. If AKIN10 is a central regulator of starvation responses, and shows an order of magnitude higher activity compared to AKIN11 in *Arabidopsis*, based on the published striking AKIN10 RNAi senescence phenotype by Bhaena-Gonzalez et al., one would expect a dramatic growth phenotype. Consequently, it is absolutely necessary to show the phenotype of the *akin10* mutant both in the light and dark that we can compare that to that of the *bzip63* mutant.

Furthermore, the analysis of transcription of bZIP63 target genes should provide further clear evidence that post-translational regulation of bZIP63 is indeed impaired in the GABI_579E09 T-DNA insertion mutant. This is because Figure 4 does not really show a very convincing change in the Phos-tag pattern of phosphorylated bZIP63 despite a lower level of “supershift” indicated by asterisks (see Figure 4 and Figure 4–figure supplement 3).

Figure 4 show yeast 2-hybrid and Agrobacterium infiltrated *N. tabacum* protein-protein interaction assays indicating that that unlike AKIN11, the AKIN10 catalytic and SNF4 activating subunits of SnRK1 interact with bZIP63. Given the high level of amino acid homology between AKIN10 and AKIN11, it is surprising that bZIP63 fails to bind AKIN11 and provide a possible hint about the location of possible interaction sites. Did you map these?

On the other hand, it is remarkable that the kinase-substrate interaction is so strong that this can be clearly detected by both assays. However, as these are heterologous assays based on overexpression of proteins, it is absolutely necessary to present co-immunoprecipitation data using the later described bZIP63 GENE construct (and not 35S ox lines) and αAKIN10 western from wt and *akin10* mutant plants.

According to Figure 4—figure supplement 1, the putative upstream activating kinase SnAK2 enhances the activity of AKIN10 (and may be that of AKIN11). SnAK2 is supposed to phosphorylate the T-loop of both AKIN10 and AKIN11. Yet, we do not see enhanced phosphorylation of GST-AKIN10/11 in this figure. Any explanation?

As GST-AKIN10/11, GST-bZIP63 and GST-SnAK2 have closely similar masses, it would be necessary to resolve them in a lower % of SDS-PAGE.

In vitro phosphorylation of bZIP63 by AKIN10 and AKIN11 looks convincing in Figure 4—figure supplement 1, so this approach should have provided a simple means to determine both AKIN10 and AKIN11 phosphorylation sites comparatively in bZIP63 by mass spectrometry. Yet, in Figure 5, a complicated intuitive combination of alanine replacements of in vivo observed seven serine phosphorylation sites is used for demonstration that GST-AKIN10 probably phosphorylates only S29, S294 and S300 residues of bZIP63.

The referred “consensus SnRK1 sites” in Figure 5 require a corresponding reference either in the text or figure legend.

The promoter activation assays in protoplast support the conclusion that S29 is the critical AKIN10 phosphorylation site, but in combination with the S294A and S300A amino acid exchanges, AKIN10-S29A shows lower transactivation of *ASN1*/*DIN6-GUS* and *ProDH-GUS* reporters. Whereas sequential phosphorylation might indeed trigger threshold-dependent conformational change of TFs, it is premature to conclude anything about possible order of phosphorylation sites (in the subsection “AKIN10 phosphorylates three conserved and functionally important serine residues in bZIP63”).

In the subsection “The AKIN10 target sites play an important role for bZIP63 function in planta” it is stated that: “The phosphorylation status of the wt construct after 6h of light and extended night and in the presence or absence of sugar was similar to the one observed in ox#3 plants (Figure 6; Figure 6—figure supplement 1).” Concerning the reproducibility and similarity of Phos-tag gels, would it be possible to run on the same gel the reactions shown in Figure 2 and/or Figure 4 with Figure 6, as well as Figure 4–figure supplement 3 and Figure 6—figure supplement 1, to have an idea about the identity/mobility of phosphorylated bands in the respective reactions? Probably, it would also be beneficial to compare the Phos-tag assays on the same gel with the reactions shown in Figure 5—figure supplement 2. This would be necessary because the Phos-tag gel depicted in Figure 6 looks dramatically different for the GY11 line in L and EN compared to the 35S:bZIP63-GFP line in Figure 2 (i.e., despite the authors' statement).

In the same subsection, I do not understand the logic of the following sentence: “Notably, hyper-phosphorylation of bZIP63 in the extended night was even stronger in the GY lines than in ox#3, probably due to lower bZIP63 expression.” Do the authors suggest that AKIN10 activity is limiting in terms of bZIP63 phosphorylation even if AKIN10 was supposed to be activated by extended night?

With the limitations mentioned above concerning the necessary parameters of *bzip63* mutant characterization, the lack of complementation by the triple S/A *bZIP63* mutant gene is documented. As the mutant gene constructs were generated by PCR of 2+xkb genomic DNA fragments, the Materials and methods should mention that all constructs were verified by sequencing.

In Figure 6, it is rather surprising that the GY11 wt complementing line shows significantly higher induction of *ASN1* and *DIN10*. In the fold-induction curve and column diagram to the right, the standard error of Ws-2 is huge at 4h of EN, thus no significant difference between wt and *bzip63* mutant can be claimed. In this figure part, either the curves shown to the left should include all four samples (Ws-2, *bzip63*, GY11 and GAY14) or all samples at the four time points should be presented as column diagrams. Given the data presented in Figure 6—figure supplement 1, I wonder why did the authors use the low expressing GY11 line for comparison with GAY14. From Figure 6—figure supplement 1 it is also apparent that the GAY4 line shows no expression of triple S/A mutant bZIP63-YFP, so only one *bzip63* mutant transformed with the triple phosphor-site mutant of bZIP63 was analysed.

Ehlert et al. reported that, when overexpressed in plant protoplasts, bZIP63 can heterodimerize with members of the S-group, and in those assays bZIP63 homodimers were barely detectable. [22] reported later that bZIP63 also interacts with the *Arabidopsis* response regulator ARR18 involved in cytokinin signaling, whereas alone (i.e. by interacting with S-group partners in heterodimers), bZIP63 functions as negative regulator of osmotic stress responses and seed germination.

Several heterodimerization partners of bZIP63 are well characterized in terms of their transcription regulatory targets. Therefore, it is apparent that their regulatory functions might be modified by bZIP63, where heterodimerization capability of bZIP63 could be controlled post-translationally, e.g. by phosphorylation through SnRK1 and potentially other protein kinases. This idea should be introduced clearly, with some necessary references to the functions of suspected bZIP63 dimerization partners at the beginning of the manuscript and discussed in this term at the end. For example, the observed effect of bzip63 mutation on the levels of free amino acids could be related to the function of its heterodimerization partner bZIP1. So, probably citing a recent paper from Para et al. (Proc Natl Acad Sci USA. 111: 10371, 2014) may be helpful for discussion of metabolite changes observed in the *bzip63* mutant.

Evidently, the *bzip63* mutation could influence the function of bZIP1, i.e. if we assumed that the bZIP1 functions as heterodimer with bZIP63. It is therefore interesting that overexpression of AKIN10 in protoplasts has only a slight effect on heterodimerization of overexpressed Gal-AD-bZIP1 and Gal-BD-bZIP63. While this artificial protoplast dimerization assay in Figure 7 indicates over 60-fold enhancement of Gal-AD-bZIP11+Gal-BD-bZIP63 dimerization, at the level of GAL-UAS4:GUS reporter, about 15-fold increase is observed, which is reduced to about 2-3-fold by triple mutation of the identified S29/S294/S300 bZIP63 phoshorylation sites. In case of Figure 7 could not fully understand why the authors used two different (i.e. GAL-UAS4:GUS and GAL-UAS4:LUC) reporters in the dimerization and reporter activation assays. This correlative evidence between the dimerization and reporter activation assay should definitely be strengthened by performing similar dimerization assays with bZIP63 carrying the S29A/S294/S300 phospho-site replacements. This appears to be obligatory.

Although the Discussion takes care to announce that “to our knowledge this is the first time that phosphorylation outside the bZIP domain was shown to affect dimerization with different partners in a distinct way”, it is rather flat. Somewhere, it should mention that the current study did not answer the question of whether the observed phosphorylation sites would modify the cellular location (i.e. nuclear import) and stability of bZIP63, in addition to its heterodimerization, which should be anyhow confirmed by co-IP studies using native gene constructs.

Unfortunately, the majority of the Discussion includes high-flying general statements, such as, “Here we show that bZIP63 plays an important role in the energy starvation response and metabolic regulation…”. If this is so, the reader will ask whether transcription of bZIP63 and its heterodimerization partners is controlled by AKIN10 and is altered in the *akin10* mutant (i.e. a necessary but missing control).

Can the enhanced senescence caused by bZIP63-ox compensated by glucose or sucrose supply? Is there any evidence for enhanced ROS production in the bZIP63-ox lines or change or red-ox regulation in the *bzip63* mutant?

“We measured strong differences in proline and asparagine levels in the bZIP63 ko and ox […] like altered carbon or nitrogen assimilation or protein turnover” (in the subsection “bZIP63 is an important metabolic regulator in the starvation response”). So, how would bZIP63 control these mechanisms? Through controlling genes involved in N or C signaling or uptake?

You say: “We found that the SnRK1 kinase AKIN10, which was proposed to be a central regulator of transcription in starvation response (3), is the major kinase responsible for the starvation-induced hyper-phosphorylation of bZIP63. Therefore, bZIP63 presents the first TF acting as direct target of SnRK1 in the transcriptional energy deprivation response” (in subsection “bZIP63 function is regulated by SnRK1-dependent phosphorylation”). Yes, but the hyper-phosphorylation of bZIP63 under extended night (called “starvation”, though it was 9 days of dark treatment) caused the appearance of only a weak band marked by asterisks on the images of Phos-tag gels, which represents the only evidence supporting this statement.

In the same subsection, you state: “Reporter activation assays in protoplasts revealed that these (i.e., AKIN10 phosophorylation) sites are essential for AKIN10-dependent induction of *ASN1* and *ProDH* by bZIP63”. This may be, but AKIN10 alone also activates these reporter genes, probably through other factors, which according to this paper are probably not the S-group bZIPs (i.e. as they appear not to be phosphorylated by AKIN10). It would be fair to mention this somewhere.

Still in the same subsection, the authors say: “The relatively small difference in *ProDH* expression in plants, as compared to the protoplast assay, is probably due to redundancy within the C/S1 group bZIPs…”. Does this sentence predict that other C-group bZIPs could be SnRK1 substrates? Would it be not necessary to discuss potential conservation of observed AKIN10 sites in other bZIPs?

In the same part of the text, the authors state: “Our data provide compelling evidence that bZIP63, but none of the S1 group bZIPs, is a bona fide in vivo target of SnRK1 in low energy signaling.” This should clearly be demonstrated by identical Phos-tag gel patterns of bZIP63 isolated from sucrose-treated or *akin10* mutant plants and that of the bZIP63 S29A/S294A/S300A protein obtained from *bzip63* plus/minus sugar or the *akin10* mutant.

Reviewer #2:

In this manuscript, Mair et al. report the function of bZIP63 as a key regulator in plant starvation response. Furthermore, using a combination of biochemical and omics approaches, the authors demonstrate that bZIP63 is regulated by protein phosphorylation and the kinase involved is AKIN10, an AMPK/Snf1/SnRK1 member. Phosphorylation of bZIP63 by AKIN10 changed its dimerization preference, thereby affecting target gene expression and ultimately plant primary metabolism.

This report identifies a missing link between energy sensing and transcriptional reprogramming of metabolism in plants under starvation conditions. The topic is of interest to a wide audience. The data are of high quality, clearly laid out, and mostly convincing. I only have one suggestion that should clarify the Phos-tag results and further enhance the quality of the paper:

Phos-tag analysis of in vitro phosphorylated bZIP63 samples allows the attribution of each shifted band to the phosphorylation of a specific Ser residue (Figure 5—figure supplement 2). In contrast, the same analyses of in vivo samples showed variable banding patterns that are not easy to reconcile (Figure 2, Figure 4, Figure 4–figure supplement 3, and Figure 6). This could be a result of phosphorylation of bZIP63 by other kinase(s) on S59, S102, S160, and S261, four sites that are not phosphorylatable by AKIN10. I would suggest the authors to determine whether the mutation of these four Ser residues to Ala will affect the function of bZIP63. If it does, the data would reveal that bZIP63 is regulated by multiple kinase pathways to carry out its function in the energy transition response. If it does not, the data would demonstrate that AKIN10-phosphorylation on S29, S294 and S300 is the major phosphorylation event that controls the activity of bZIP63. By performing Phos-tag analysis of bZIP63-S59A/S102A/S160A/S261A, the authors can provide more convincing results about the in vivo phosphorylation of bZIP63 by AKIN10 kinase.

Reviewer #3:

The authors demonstrate low energy stress-dependent SnRK-catalyzed direct phosphorylation of bZIP63 transcription factor as prerequisite for differential bZIP homo/hetero dimerization. Their comprehensive clear data provide for the first time mechanistic insight in how transcriptional reprogramming of gene subsets in a given pathway is driven by a conditional differential hetero/homodimerization of key transcriptional regulators.

The manuscript is well written, clear in its argumentation, and statements are well backed by an almost overly thorough analysis and a large amount of data of high quality. I would like to comment on two aspects:

1) Because SnRK and bZIP are well known proteins in plant biology, and because phosphorylation of bZIPs has been reported before, the true novelty in the manuscript is in my opinion the phosphorylation-dependent differential dimerization leading potentially to a change in transcriptional specificity plus activity toward distinct target genes (summarized in Figure 7 – model). Data in Figure 7 rely on a protoplast two-hybrid assay, allowing a transcriptional readout (equal protein amount in transfected cells has not been included here). To more prominently support the key message of the manuscript will have to address:

i) Differential dimerization on protein level: conducting for example transient co-localization studies based on (multi-color) fluorescent proteins/BiFC (60), or using biochemical means to compare phosphorylation-dependent bZPI63/bZIP63 with bZIP63/bZIP11 formation, and in case even include bZIP S29D (or other) phospho-mimic constructs.

ii) Show differential target gene specificity: Figure 7 contains as marker for transcriptional activity a gene which is target to all three bZIPs (1, 11, 63). Even more convincing will be phosphorylation/heterodimerization-dependent activity toward a distinct bZIP11- or bZIP63-controlled gene, respectively.

2) The primary identification of “in vivo” phosphorylation sites is conducted on bZIP-GFP overexpressing lines on material harvested in both light and prolonged dark. Therefore, basic constitutive bZIP phosphorylation (and the involvement of other kinases than AKIN10) cannot be excluded. Ideally, one may ask for the degree of conditional in vivo phosphorylation (in particular at S29 of endogenous bZIP) performing MRM/SRM phosphoproteomics.

---

## [Author Response]

[Editors’ note: the author responses to the first round of peer review follow.]

We have carefully read the reviewer’s comments and found that although some points that are mentioned are valid and we agree that these have to be addressed, we also saw that some parts of our manuscript were not written clearly enough and have been misunderstood. Therefore we would like to respond here directly to two issues that we think are the major issues for all three reviewers and ask furthermore for concrete feedback regarding the resubmission of a revised version of this work. To this end, we outline the experiments and further clarifications we would propose for a revision.

1) The *akin 10* mutant:

We understand the necessity of a comprehensive characterization of the *akin10* mutant, which we used for our study, and we apologize for not including that in our manuscript. This mutant has been used – and confirmed – by three groups in our consortium independently. Figure 10 shows clearly that this mutant (GABI KAT *GABI_579E09*) is indeed a valid *akin10* null mutant. The T‐DNA insertion was indicated at the very 3’ end of the *AKIN10* gene directly in front of the last exon (Figure 10). We could indeed confirm the insertion at this position using specific primer combinations (Figure 10). The Gabi homepage indicates that there might be a second insertion in another gene (IMS2, At5g23020; Figure 10). This was tested by PCR using gene‐specific primers against the flanking region of the second insertion but no insertion could be detected (Figure 10).

We checked the expression level of AKIN10 protein by Western blotting using two antibodies: firstly the AKIN10 specific antibody from Agrisera (AS 10919), and secondly an antibody against the phosphorylated T‐loop threonine (T172) of AMPKα, the mammalian AKIN10 ortholog (Cell Signaling #2531), that specifically recognizes the phosphorylated T‐loop threonine in AKIN10 and AKIN11 (T175 and T176, respectively; Baena‐González et al., 2007). The position of the peptide recognized by each antibody is indicated in Figure 10. Notably both antibodies bind to regions far before the T‐DNA insertion in the AKIN10 sequence. As shown in Figure 10, the AKIN10 antibody recognizes a single protein at the expected MW (59‐61 kDa) in wt extracts, but not in the *akin10* mutant. The anti Thr 172 antibody detects a double band of 61 and 58 kDa in the wt, and only the lower band (58 kDa = AKIN11) in the *akin10* line. These data were completely independently obtained in the Baena‐ Gonzalez lab as well as in our lab.

As reviewer 1 raised also concerns regarding a potential growth phenotype of the *akin10* mutant, we include here also a phenotypical growth analysis of the *akin10* mutant in Figure 10. Figure 10 shows that the *akin10* mutant displays normal growth with respect to leaf number, fresh and dry weight as well as water content. As can be seen in Figure 10 the *akin10* mutant displays a little delay in flowering as compared to the wt, but it is flowering and produces viable seeds without problem. This is also in perfect agreement with the data on the *AKIN10* RNAi line reported by [3] (Nature 448 (7156):938‐42). The two different *AKIN10* RNAi lines described in that paper showed also normal growth. It is the *akin10/11 double mutant* that is apparently not viable and when an *akin10/11* knockdown is generated through VIGS, the plants show premature senescence that precludes flowering. It is also shown there that *AKIN10 overexpression* alters inflorescence development.

Author response image 1.Characterization of the *akin10* mutant line GABI_579E09.**(A‐D)** Mapping of the T‐DNA insertion site. **(A)** The top scheme represents the genomic locus of *AKIN10* (AT3G01090). UTRs are grey, exons are green boxes. The first exon, which is only present in splicing form 2 (sf2), is shown in light green. By sequencing of the flanking regions the T‐DNA insertion was mapped to position 2583‐2593 (11pb deletion), just before the last exon. Primers used for the genotyping PCR shown in **(B)** are indicated. The scheme below represents the AKIN10 protein with the kinase domain (KD) in yellow and the kinase associated (KA) domain in blue. The position of K48 (ATP binding; mutated to M in the inactive version) and T175 (target for SnAK2 in the T‐loop), as well as the approximate binding sites of the antibodies against AKIN10 (Agrisera; AS 10919) and AMPKα‐pT172 (Cell Signaling; #2531) are indicated. **(B)** Genotyping PCR with primers binding in the AKIN 10 sequence (fwA and rvA) and the T‐DNA left border (LB). **(C)** The scheme represents the genomic locus and protein of *IMS2* (AT5G23020), a potential second T‐DNA insertion site. Sequencing of the T‐DNA flanking region by GABI KAT (www.gabi.kat.da) produced a sequence with partial homology to *IMS2*. However, this insertion was not confirmed. The proposed position of the T‐DNA is indicated. **(D)** Genotyping PCR with primers binding in the *IMS2* sequence (fwI and rvI) and the sequencing primer (seq) used by GABI KAT. **(E)** Western blots against AKIN10 in Col‐0 wt and *akin10* plants. Antibodies against AKIN10 (top) and AMPKα‐pT172, which recognizes both AKIN10 and AKIN11 phosphorylated in the T‐loop, (bottom) were used. The position of AKIN10 and phospho‐AKIN10/11 is indicated. Left: 5 week‐old plants grown on soil in a 12h/12h light/dark cycle. Total proteins were extracted from rosettes harvested after 6h of light (L) or extended night (EN). Right: 2 week‐old seedlings in liquid ½ MS medium. Total proteins were extracted from seedlings treated with 6h of extended night in the presence (+) or absence (‐) of 1% sucrose. **(F)** Comparison of 8 week‐old wt and *akin10* plants. The plants were grown under short day conditions (8h/16h light/dark). The number of leaves, fresh weight (FW), dry weight (DW), and the ratio between FW and DW was determined. The bars represent the mean ± SD of 19‐20 replicates. T‐tests did not produce significant differences between wt and mutant. **(G)** Mild flowering phenotype of *akin10*. Col‐0 wt and *akin10* plants were grown under SD conditions and the number of leaves at bolting time were counted. The bars represent the mean ± SD of 15 and 11 replicates, respectively. The p‐value of an unpaired *t*‐test with unequal variance was < 0.05 as indicated by *.**DOI:**
http://dx.doi.org/10.7554/eLife.05828.044

We also tested the *expression levels of AKIN10 target genes* in response to six hours extended night treatments in this *akin10* line (Figure 11). After six hours of extended night (EN), we observed a clearly impaired induction of *ASN1* (*DIN6*) and *ProDH*. *DIN10* (raffinose synthase) was not affected under this conditions, but this time point might already be a bit too late, as we observed clear changes in the induction of all three genes already starting after two hours EN in the *bzip63* mutant (Figure 6). Therefore we selected an earlier time point (4 h) for the analysis of bZIP63 target gene expression in the complementation lines. In general, it is important to consider that AKIN10 and AKIN11 seem to act in a dynamic system, in which AKIN11 can take over the function of AKIN10 after some time. This becomes evident at different levels: firstly, the viability of the *akin10* mutant; secondly, the level of target gene expression (double knockout is required to abolish induction in response to starvation); thirdly, the remaining background in target phosphorylation (i.e. bZIP63) as, for example, seen in the phos‐tag analysis (see Figure 12).

Author response image 2.Expression levels of the proposed AKIN10/bZIP63 target genes *ASN1*, *DIN10*, and *ProDH* in wt and the *akin10* mutant line in light (L) and extended night (EN). Col‐0 wt and *akin10* plants were grown on soil in a 12h/12h light/dark regime for 5 weeks and harvested after 6h of light and extended night. RNA was extracted from whole rosettes, reverse transcribed and used for qPCR of *ASN1* (ASPARAGINE SYNTHASE 1; AT3G47340), *DIN10* (DARK INDUCIBLE 10; AT5G20250), and *ProDH* (PROLINE DEHYDROGENASE; AT3G30775). *TBP2* (TATA BINDING PROTEIN 2; AT1G55520) and *MHF15* (Thioredoxin superfamily protein; AT5G06430) were used as reference genes. Raw data were analyzed with LinRegPCR (version 2014.02, [43]). The mean N0 values from two technical replicates were normalized using the geometric mean of the values for *TBP2* and *MHF15*. The bars represent the mean ± SD of 3 biological replicates normalized to the expression in wt in light. P‐values from *t*‐test < 0.05 are indicated by *.**DOI:**
http://dx.doi.org/10.7554/eLife.05828.045

2) In vivo phosphorylation status of bZIP63 detected in phos‐tag gels:

Reviewer 1 and 2 raised several questions about the phosphorylation status of bZIP63 as detected by Western blotting of phos‐tag gels. As reviewer 1 pointed out, so far no transcription factor has rigorously been shown to represent a direct phosphorylation target of SnRK1 in plants. Therefore, we think that the phos‐tag data, which show the in vivo phosphorylation status of bZIP63, are crucial for this manuscript and we feel that we need to explain these better in different conditions, treatments and of AKIN10.

In order to better illustrate the different phosphorylation status and to allow a better comparison of the different experiments, we prepared a figure that shows a direct comparison of all phos‐tag gels from the different experiments (Figure 12).

As control for the non‐phosphorylated bZIP63, the recombinant protein is shown in the very left panel. For all other samples we have now labelled the bands from 1‐7 according to an increase in their phosphorylation status. This is the maximum of different isoforms we could identify on different phos‐tag gels and corresponds to the seven identified in vivo phosphorylation sites from the combined MS analyses. Of course, not all isoforms are always visible on the gel as they depend on the phosphorylation level, and furthermore, the phosphorylation status is also highly dynamic, which complicates the picture. Nevertheless, the main point we want to make in the manuscript – and we think this is clearly evident from the figures – is that the level of bZIP63 phosphorylation changes in response to the different treatments and is furthermore depending on AKIN10.

Author response image 3.Comparison of bZIP63 phosphorylation on phos‐tag gels in response to different treatments and in different genetic backgrounds. GFP/YFP‐tagged bZIP63 was detected by western blotting with an antibody against GFP (top). The Coomassie Brilliant Blue (CBB) stained membranes are given below. Recombinant bZIP63‐YFP is shown as a control for non‐phosphorylated bZIP63. On the left, phosphorylation of bZIP63 after 6h of light (L) or extended night (EN) is shown. Proteins were extracted from whole rosettes of 5 week‐old plants grown on soil in a 12h/12h light/dark cycle. On the right, phosphorylation of bZIP63 in response to sugar treatment is depicted. Seedlings were grown for 1 week in liquid ½ MS in the presence of 0.5% sucrose, followed by 1 week without sucrose. They were then treated with 6h of EN in the presence (+) or absence (‐) of 1% sucrose and harvested. The plant lines used are two bZIP63 over‐expressor lines – *35S::bZIP63‐GFP* in wt and *UBI10::bZIP63‐YFP* in *akin10* – and two genomic complementation lines expressing YFP‐tagged bZIP63 – wt bZIP63 (GY11) and bZIP63 harboring S/A mutations in all three AKIN10 target sites (AY14) – under its endogenous promoter in the *bzip63* background. All bands (except unspecific bands and degradation bands in the two blot on the right) are indicated by black and grey dots. The color of the dot indicates the strength of the band in the blot. The scheme on the right shows the numbers assigned to each band in the other figures.**DOI:**
http://dx.doi.org/10.7554/eLife.05828.046

In the first panel from the left, it can be seen that, in the light, bZIP63 is already highly phosphorylated – which is not surprising considering seven identified in vivo phosphrylation sites and several potential kinases. However, the majority of bZIP63 is present in bands 5 and 6. In response to starvation (6 hrs extended night EN), band 7 appears as higher phosphorylated form. This new band 7 is barely detectable in the EN samples from the *akin10* mutant. This might be considered as a small difference but these data should be seen in the overall context. In the genomic complementation, lines bZIP63 are almost exclusively detected in bands 5 and 6 in the light (so the same as before), and now most of the bZIP63 protein is shifted to the most phosphorylated form in band 7 in the extended night (EN). Strikingly, in the line complemented with the version of bZIP63 in which all three AKIN10 target sites are mutated to an alanine (S/A), ZIP63 is detected only in the bands 1 and 2 (so almost non‐phosphorylated) under all conditions. This makes clear that 1) the 3 mutated sites are the main in vivo phosphorylation sites and 2) the increased phosphorylation in EN happens at one of these sites.

As reviewer 1 pointed out, the band pattern in this line (GAY14) is not the same as in the *akin10* line. This is indeed true, however, considering again that we identified more than one kinase as potential upstream regulators of bZIP63 this is not surprising and does *not* mean that AKIN10 does not phosphorylate bZIP63 in vivo. As mentioned before, one likely explanation for the differences in phosphorylation is that AKIN11 might take over part of the function of AKIN10 in the *akin10* mutant. This would also explain the lack of a strong phenotype in the *akin10* line (see Figure 1 and Baena-González et al., 2007) compared to the *akin10 akin11* double ko. Although AKIN11 does not interact well with bZIP63 in the Y2H and BiFC assays, it might still interact with bZIP63 via SNF4 and form a stable complex in vivo. Moreover, other kinases (like the identified CKII or CDPKs) could also phosphorylate one or more of the three target sites of AKIN10. We will discuss this aspect more clearly in a revised version.

Reviewer #1:

Plant SnRK1 kinases are structural and functional homologs of yeast Snf1 and animal AMP-activated kinases that play a central role in sensing energy depletion and maintaining cellular energy homeostasis. In this paper a multinational consortium set out to demonstrate that Arabidopsis SnRK1, carrying the AKIN10 catalytic subunit, phosphorylates in vivo the C-group bZIP63 transcription factor. bZIP63 has been identified to control transcription of some SnRK1-regulated target genes, involving ASPARGINE SYNTHASE 1 (ASN1/DIN6), PROLINE DEHYDROGENASE (PDH1/PRODH), RAFFINOSE SYNTHASE (DIN10) and SENESCENCE 1 (SEN1). To justify the statements in the Abstract and Introduction that bZIP63 is a key regulator of starvation response, the paper starts with limited characterization of a bzip63 T-DNA insertion mutant, and two bZIP63-ox overexpression lines.

Molecular characterization and description of the *bZIP63* ko and ox lines was provided in Figure 6–figure supplement 6A-C as well as in the Methods section on plant lines in the previous version of the manuscript. To make the information more accessible, we moved the characterization of the plant lines to a new figure in the beginning of the manuscript (Figure 1—figure supplement 1) and included some additional information on the characterization of the *bzip63* line, as well as some plant pictures.

Furthermore, we have now also included a number of different assays to clearly show, that the observed phenotype is indeed dark-induced senescence (DIS) caused by energy depletion (see qPCR data) and that this phenotype can be rescued by addition of sugar.

*The authors use a phenotypic assay by placing plants for 9 days into the dark and monitoring rosette leaf senescence/bleaching by ImageJ scanning of green versus bleached/yellow leaf areas. Whereas in wild type (wt) the first two leaves (numbered 3 and 4 in*
Figure 1*) show bleaching, they remain greenish in the bzip63 mutant (defined as stay-green phenotype), and 3-5 more leaves undergo bleaching in the bZIP63-ox plants, suggesting enhanced senescence response*.

We would like to correct the statement of the reviewer. As indicated in the text, all leaves used for senescence analysis were *true* leaves. Cotyledons were not included. Therefore the leaves numbered with 3 and 4 are not leaf 1 and 2 but really leaves 3 and 4.

*When grown in the light, the* bzip63 *and bZIP63-ox plants do not differ from wt*.

*Next, the authors compare metabolites entering or derived from glycolysis and citrate cycle in light-grown (L) and dark exposed (elongated night, EN) plants and find that the levels of* “*most*” *free amino acids is increased in the* bzip63 *mutant and decreased in bZIP63-ox plants, whereas sugar levels are stated to show an opposite trend of changes. Based on these data the authors conclude that bZIP63 is a key regulator of starvation responses (see Abstract and Introduction)*.

*When inspecting the metabolite profiling data, it is clear that the discussion of the results is rather superficial. For example, raffinose levels are decreased, whereas trehalose levels are increased in both mutant and ox lines, similarly to glucose in the dark. In general, we do not learn much from the metabolomics data. The actual conclusion is that* “*misregulation of bZIP63*” *(in the subsection “bZIP63 controls dark induced senescence and primary metabolism”) has an effect on metabolism*.

We agree with this and have intensively studied the literature and found a number of relevant links of our metabolite data to starvation and senescence. We have therefore completely re-written this part and are fully confident that the data now do clearly support the DIS phenotype and the role of the AKIN10-bZIP63 module in regulating primary metabolism in the low-energy response.

*On the other hand, we do not find any additional data (such as quantitative measurement of chlorophyll levels, degradation products, or other senescence markers, such as enzymes or transcripts), which would provide a further insight documenting how alteration of bZIP63 levels modulate the starvation-induced and/or senescence responses*.

We now include quantitative chlorophyll measurements of plants before and after 9 days in darkness (Figure 1—figure supplement 2). Results are similar to the “green leaf area” data presented throughout our manuscript, which justifies the use of green leaf area as a crude measure for chlorophyll content. In this context we would like to mention that senescing leaves often have reduced water content. Therefore, when normalizing the chlorophyll content to fresh weight this can lead to overestimation of the chlorophyll content in strongly senescing plants. We thus believe that the “green leaf area” method produces more reliable data for estimating the chlorophyll content during dark-induced senescence.

Further, to confirm that the phenotype we observe is related to dark-induced and not age-induced senescence we did RT-qPCR of marker genes for dark-induced and natural senescence (Figure 1—figure supplement 3). Early time points up to day 4 were chosen as transcriptional response to darkness is faster than visible chlorophyll loss. CHLOROPHYLL A/B BINDING PROTEIN 2 (*CAB2*), as a marker for active photosynthesis was quickly downregulated, while SENESCENCE 1 (*SEN1*) and SENESCENCE ASSOCIATED GENE 103 (*SAG103*), were strongly upregulated. Both genes are known to be regulated by dark-induced senescence. In contrast, YELLOW-LEAF-SPECIFIC 3 (*YLS3*), which is induced by natural senescence, was not increased. We could not observe any strong differences in expression of these genes between wt and *bzip63* plants, indicating that bZIP63 is not involved in the signaling pathways upstream of these senescence markers.

Moreover, as mentioned above, the delayed senescence in the *bzip63* plants could be related to the metabolic phenotype we observed. Amino acids can serve as alternative energy under starvation conditions ([51], Proline: a multifunctional amino acid; [52], The role of mitochondria in leaf nitrogen metabolism). Therefore, the increased amount of total amino acids in the bZIP63 ko could lead to a delayed onset of visible senescence effects. We have discussed this possibility in the revised version of our manuscript.

*Whether 9 days of dark treatment can be considered as starvation or general stress response remains a matter of discussion*.

Dark induced senescence was suggested to be a result of sugar/carbon depletion ([5], Comparative transcriptome analysis reveals significant differences in gene expression and signalling pathways between developmental and dark/starvation-induced senescence in *Arabidopsis*). We therefore tested whether the dark-induced senescence phenotype is energy related and could be complemented by the addition of sugars. We found that glucose can indeed almost completely rescue the dark-induced senescence phenotype of bZIP63 mutant in seedlings (Figure 1). This points towards an involvement of bZIP63 in sugar/starvation signaling. Also, seedlings grown on plates with sugar showed enhanced growth and reduced or no chlorophyll loss as compared to the control plates without sugar, indicating that energy/carbon depletion is the major reason for senescence under these conditions.

*By performing 2D and Phos-tag gel westerns (*Figure 2*), the authors detect multiple phosphorylated (lambda phosphatase sensitive) isoforms of bZIP63-GFP immunoprecipitated from overexpressing plants. On the Phos-tag gel, they show a rather weak band termed* “*supershift*” *of bZIP63, which appears in dark-treated plants, and detect somewhat weaker bZIP63-GFP bands with altered pattern plants grown with 1% sucrose. They use the latter result, without explicit explanation or demonstration, to suggest that the observed phosphorylation changes reflect inhibition of SnRK1 by 1% sucrose*.

Although it is tempting to suggest a connection between the reduced phosphorylation of bZIP63 in the presence of sucrose and the activity of SnRK1, we do *not* claim at any point in the text that this is the case. We merely suggest that sugar feeding counters the starvation resulting from cultivation in extended dark conditions.

However, it is possible that some of the difference in bZIP63 phosphorylation between seedlings incubated in the presence and absence of sucrose is due to altered SnRK1 activity. T6P levels increase and decrease in parallel to sucrose levels and are rapidly depleted in the presence of sucrose (Smeekens et al., 2010, Sugar signals and molecular networks controlling plant growth; Lunn et al., 2014, Trehalose metabolism in plants) and it has been shown that in seedlings and young leaves T6P has an inhibitory effect on SnRK1 activity (Zhang et al., 2009, Inhibition of SNF1-related protein kinase 1 activity and regulation of metabolic pathways by trehalose-6-phosphate).

*By mass spectrometry analysis, seven phosphorylated residues were then identified in the immonoprecipitated bZIP63-GFP protein. Subsequent in-gel kinase tests indicate that bZIP63 is phosphorylated by multiple protein kinases showing apparent molecular masses of 50-60 and 35-38 kDa. In*
Figure 2*, it should be clearly indicated that the plant extract used for in-gel kinase assays derived from hydroponic root cultures (see “Kinase assays”, third paragraph), along with the conditions used for root culture in the case of the experiment in*
Figure 2.

We assume the reviewer means 3A in the original manuscript (Figure 4 in the revised manuscript). We have specified the use of roots and root cell culture in the figure legend as well as in the main text. The growth conditions are described in the Methods section under “plant growth”. We further added more detailed information on the plant material used in each of the four affinity purification/kinase identification experiments that were conducted in the excel spreadsheet containing information on which kinase was identified in which sample (first tab of [Supplementary-material SD3-data]).

*To isolate the kinases detected in the in-gel assay, GST-bZIP63 was immobilized on GST-Sepharose and plant protein extracts made from root cultures or root-derived cell suspension (as the latter is maintained in the presence of sucrose, please specify which of these sample is shown in*
Figure 2*) were subjected to affinity binding followed by NaCl elution. The eluted proteins resolved between 30 and 60 kDa were isolated from in-gel kinase PAGE (if I understood well), and identified by mass spectrometry. From 28 candidates only 7 kinases (including the SnRK1 catalytic AKIN10, AKIN11 and activating subunit SNF4, as well as casein kinase subunits CKA1, CKA2, CKB1 and CKL2) were identified by 2-9 peptides. It should be noted in the Materials and methods how the excess of His6-bZIP63 was filtrated out during the mass spectrometry analysis*.

The reviewer misunderstood the use of in-gel PAGE for protein identification. Proteins were isolated from a non-substrate gel. This is shown in the scheme in Figure 4 and described in the Methods subsection “Identification of proteins and in vivo phosphorylation sites by LC-MS/MS”. As this seems not to be clear enough, we clarified the use of a normal SDS PAGE gel for MS analysis in the main text. As the reviewer pointed out correctly, using a gel with substrate would indeed strongly hamper identification of other proteins by MS/MS, if not prevent it completely.

*In addition, it would be worth knowing whether the GST-alone matrix had retained any of these kinases*.

Unfortunately, we never tested whether any of the kinases phosphorylating bZIP63 binds to a GST-coupled column. Affinity purification was employed in the first place to reduce the complexity of the sample and at the same time enrich for bZIP63 binding proteins before MS analysis. For kinase identification we cut out only those regions giving a signal in the in-gel kinase assay and considered only kinases with the correct molecular weight as a further restrictive step.

It is possible that one of the kinases identified by MS binds to GST and not bZIP63. However, for the “hot candidates” among the kinases we conducted additional assays (Y2H, BiFC) to confirm their interaction with bZIP63 and we tested phosphorylation of bZIP63 by these kinases. Results for SnRK1 subunits can be found (among others) in Figure 5 (including Figure 5—figure supplement 4 and Figure 5—figure supplement 5) and Figure 6 (including Figure 6—figure supplement 1 and Figure 6—figure supplement 2). We are therefore confident that AKIN10 and AKIN11 were not retained by GST, but by bZIP63.

*Furthermore, αAKIN10 western showing an enrichment of AKIN10 on the GST-bZIP63 matrix versus GST would be more convincing*.

We did not do a western blot as we did not have an antibody against AKIN10 at the time we undertook the affinity purifications (2008). However, we are routinely doing proteomics analysis in our department and SnRK1 peptides are usually not detected in total protein extracts with shotgun proteomics. As we repeatedly found a large number of peptides for AKIN10 and AKIN11 in our affinity purification samples, this is a very good indication that these kinases have been enriched.

*From here on, the authors concentrate on the analysis of bZIP63 phosphorylation by the SnRK1 catalytic subunits AKIN10, which according to Baena-Gonzalez et al. and Jossier et al. (Plant J 59: 316, 2009), is at least 10-20 times more active/abundant compared to its homolog AKIN11 in* Arabidopsis*.*
Figure 3
*illustrates that GST-AKIN11 purified from* E. coli *also phosphorylates bZIP63 less efficiently compared to GST-AKIN10 in vitro. It is strange that no autophosphorylation of AKIN10 was ever detected by the authors either in normal or in-gel kinase assays. Any explanation?*

We assume the reviewer means 4B in the original manuscript (Figure 5 in the revised manuscript).

In fact, auto-phosphorylation was detected. However, it is quite weak compared to substrate phosphorylation and can therefore not be seen well on the autoradiography.

*As a key argument, the authors use the* akin10 *GABI_579E09 T-DNA insertion mutant to demonstrate that AKIN10 is indeed responsible for in vivo phosphorylation of bZIP63. Unfortunately, according to the reviewer's knowledge, this mutant has not yet been characterized by others, so, to a certain extent, this must be performed by the authors in this paper*.

Unfortunately we were not aware of the fact that this mutant had never been described before. We had of course characterized the *akin10* mutant line before use. We added a detailed molecular characterization of the mutant in Figure 5—figure supplement 1. Additionally we show some data on the phenotypic appearance of the mutant and expression of selected AKIN10 target genes in the mutant in light and extended night (Figure 5—figure supplement 2).

*In*
Figure 4*, they show that one of the phosphorylated bZIP63 bands around 55-57 kDa is missing in their in-gel kinase assay with* akin10 *mutant extracts, and they fail to detect AKIN10 by western with the Agrisera αAKIN10 antibody in roots, but observe a lower Mw weakly cross-reacting protein band in leaf extracts. According to these results, the GABI_579E09 insertion mutant appears to produce no immunologically detectable AKIN10. Yet, the authors could successfully cultivate the mutant, which means that it must be fertile. If AKIN10 is a central regulator of starvation responses, and shows an order of magnitude higher activity compared to AKIN11 in* Arabidopsis*, based on the published striking AKIN10 RNAi senescence phenotype by Bhaena-Gonzalez et al., one would expect a dramatic growth phenotype. Consequently, it is absolutely necessary to show the phenotype of the* akin10 *mutant both in the light and dark that we can compare that to that of the* bzip63 *mutant*.

The reviewer confuses the plant lines described in Baena-Gonzalez et al. (2007, A central integrator of transcription networks in plant stress and energy signaling). The mutant showing the strong phenotype in the Baena-Gonzalez paper is an *akin10/akin11* double mutant (VIGS of *AKIN11* in *AKIN10* RNAi background). They also showed *akin10* and *akin11* single mutants, which did not have a striking phenotype. Single mutants of *AKIN10* (RNAi) were not smaller than wt and flowered normally, which is probably due to the action of AKIN11. Therefore, the lack of a strong phenotype in our *akin10* T-DNA insertion line is not in contrast to the data published in Baena-Gonzalez, but rather supports the findings described there. A molecular and phenotypic characterization of the GABI-KAT *akin10* line can be found in Figure 5—figure supplement 1 and Figure 5—figure supplement 2.

*Furthermore, the analysis of transcription of bZIP63 target genes should provide further clear evidence that post-translational regulation of bZIP63 is indeed impaired in the GABI_579E09 T-DNA insertion mutant*.

Not many bZIP63 target genes are known. We tested the expression of *ASN1*, *DIN10*, and *ProDH* in the *akin10* line after 6h of light and extended night (Figure 5—figure supplement 2). *ASN1* and *ProDH* showed reduced expression under both conditions and in extended night, respectively. Expression of the genes is not completely abolished, but this is not surprising considering that AKIN11 seems to have redundant functions (as can be seen from the phenotype of single and double mutants).

*This is because*
Figure 4
*does not really show a very convincing change in the Phos-tag pattern of phosphorylated bZIP63 despite a lower level of* “*supershift*” *indicated by asterisks (see*
Figure 4
*and Figure 4–figure supplement 3)*.

It is true that there is no difference in bZIP63 phosphorylation in wt and *akin10* background in the light and in extended night. Only the upper band is reduced (Figure 5, left) and we can see how this may seem disappointing at first sight. However, we identified more than one kinase and it can therefore not be expected that the phosphorylation of bZIP63 vanishes completely when one kinase is missing. We do see a clear reduction in the phosphorylation of bZIP63 under extended night conditions in the *akin10* mutant line, which tells us that AKIN10 phosphorylates bZIP63 only under specific conditions. The fact that only one band is vanishing and not all means that probably only one site is less phosphorylated, while others, which are targeted by other kinases, are not.

As it is possible that AKIN11 phosphorylates bZIP63 stronger in the absence of AKIN10, we now knocked down both AKIN10 and AKIN11 using VIGS and looked at the bZIP63 phosphorylation pattern. We saw that again in light there was no difference but the reduction of the “supershift” band in extended night was even stronger in these plants (Figure 5, right). In our eyes, this confirms that both SnRK1 kinases phosphorylate bZIP63 under extended night conditions. Thus, we are fully confident that with the additional Phos-tag data, particularly those from the genomic constructs and the VIGS experiment, the effect is absolutely convincing.

Figure 4
*show yeast 2-hybrid and Agrobacterium infiltrated N. tabacum protein-protein interaction assays indicating that that unlike AKIN11, the AKIN10 catalytic and SNF4 activating subunits of SnRK1 interact with bZIP63*. *Given the high level of amino acid homology between AKIN10 and AKIN11, it is surprising that bZIP63 fails to bind AKIN11 and provide a possible hint about the location of possible interaction sites. Did you map these?*

No, we did not map the interaction sites. We observed a very weak interaction with AKIN11 in tobacco and no detectable interaction in yeast. This probably reflects the affinity of the kinase subunit alone to the bZIP and fits to the observed reduction in phosphorylation of bZIP63 by AKIN11. However, in vivo AKIN11 probably still plays a role in bZIP63 phosphorylation. SnRK1 forms a trimeric complex, which always includes SNF4 ([11], SnRK1 from *Arabidopsis thaliana* is an atypical AMPK). As SNF4 interacts strongly with bZIP63, it might act as a link to bring bZIP63 and AKIN11 together.

*On the other hand, it is remarkable that the kinase-substrate interaction is so strong that this can be clearly detected by both assays. However, as these are heterologous assays based on overexpression of proteins, it is absolutely necessary to present co-immunoprecipitation data using the later described bZIP63 GENE construct (and not 35S ox lines) and αAKIN10 western from wt and akin10 mutant plants*.

We tried co-IPs from the genomic complementation lines. However, we could not successfully pull down AKIN10. One possible explanation for this is that, as the reviewer stated above, interactions between a kinase and its substrate are mostly not very strong and often not very long lived. We can therefore expect that only a fraction of bZIP63 is bound to AKIN10 at any given time and that there is not enough bound AKIN10 to be detected in the western blot. This analysis is furthermore complicated because bZIP63 is expressed at very low levels and very hard to detect (already in plant extracts), therefore it is again not surprising that we could not detect it in a co-IP where most likely only a fraction will bind very transiently to the kinase.

*According to*
Figure 4—figure supplement 1*, the putative upstream activating kinase SnAK2 enhances the activity of AKIN10 (and may be that of AKIN11). SnAK2 is supposed to phosphorylate the T-loop of both AKIN10 and AKIN11. Yet, we do not see enhanced phosphorylation of GST-AKIN10/11 in this figure. Any explanation?*

The phosphorylation of AKIN10 and AKIN11 in the presence of SnAK2 is enhanced, as can be seen in Figure 5—figure supplement 4. However, the amount of AKIN10 and SnAK2 used in this kinase assay is much lower than the amount of bZIP63. Therefore, the phosphorylation band of AKIN10 is weak in relation to the bZIP63 band and cannot be seen very well on the gel. In Figure 5—figure supplement 4 AKIN10 and AKIN11 are always together with SnAK2, so the increased AKIN10 phosphorylation cannot be seen in this figure. Comparing phosphorylation intensities between figures is not possible.

*As GST-AKIN10/11, GST-bZIP63 and GST-SnAK2 have closely similar masses, it would be necessary to resolve them in a lower % of SDS-PAGE*.

The bands are close together, but still distinguishable. Only SnAK2 is very close in size to bZIP63. We provide all controls to show that the substrate, and not the kinases, is highly phosphorylated.

To better separate the bands, it would be necessary to run the gels longer, which we would like to avoid as otherwise radioactive nucleotides run into the running buffer.

*In vitro phosphorylation of bZIP63 by AKIN10 and AKIN11 looks convincing in*
Figure 4—figure supplement 1*, so this approach should have provided a simple means to determine both AKIN10 and AKIN11 phosphorylation sites comparatively in bZIP63 by mass spectrometry. Yet, in*
Figure 5*, a complicated intuitive combination of alanine replacements of in vivo observed seven serine phosphorylation sites is used for demonstration that GST-AKIN10 probably phosphorylates only S29, S294 and S300 residues of bZIP63*.

Kinase assays with non-phosphorylatable mutants of potential kinase target sites are a standard technique to pin down the likely kinase target sites. We do not think that this strategy is complicated and unusual.

Furthermore, as was mentioned in our manuscript, it is extremely difficult to cover the total protein with MS as some areas do not produce suitable tryptic peptides. Therefore an MS based approach was not our first choice.

*The referred* “*consensus SnRK1 sites*” *in*
Figure 5
*require a corresponding reference either in the text or figure legend*.

The reference was added in the figure legend.

*The promoter activation assays in protoplast support the conclusion that S29 is the critical AKIN10 phosphorylation site, but in combination with the S294A and S300A amino acid exchanges, AKIN10-S29A shows lower transactivation of* ASN1*/*DIN6-GUS *and* ProDH-GUS *reporters. Whereas sequential phosphorylation might indeed trigger threshold-dependent conformational change of TFs, it is premature to conclude anything about possible order of phosphorylation sites (in the subsection “AKIN10 phosphorylates three conserved and functionally important serine residues in bZIP63“).*

We removed speculations about the order of phosphorylation.

*In the subsection “The AKIN10 target sites play an important role for bZIP63 function in planta“ it is stated that:* “*The phosphorylation status of the wt construct after 6h of light and extended night and in the presence or absence of sugar was similar to the one observed in ox#3 plants (*Figure 6*;*
Figure 6—figure supplement 1*).*” *Concerning the reproducibility and similarity of Phos-tag gels, would it be possible to run on the same gel the reactions shown in*
Figure 2
*and/or*
Figure 4
*with*
Figure 6*, as well as Figure 4–figure supplement 3 and*
Figure 6—figure supplement 1*, to have an idea about the identity/mobility of phosphorylated bands in the respective reactions? Probably, it would also be beneficial to compare the Phos-tag assays on the same gel with the reactions shown in*
Figure 5—figure supplement 2*. This would be necessary because the Phos-tag gel depicted in*
Figure 6
*looks dramatically different for the GY11 line in L and EN compared to the 35S:bZIP63-GFP line in*
Figure 2
*(i.e., despite the authors' statement)*.

Unfortunately, it is impossible to run all samples on one gel due to technical limitations, i.e. loading capacities. As the identity of the bands on the Phos-tag gels seem to be too confusing, and comparison between figures is difficult, we have first aligned all Phos-tag gels and put them in a supplementary figure for convenient comparison (Figure 3—figure supplement 1).

Further, we marked all bands and labeled them persistently with numbers from 1-8 in each of the figures according to an increase in their phosphorylation status. This is the maximum of different isoforms we could identify on different Phos-tag gels and corresponds to the nonphosphorylated form plus the seven identified in vivo phosphorylation sites from the combined MS analyses. Of course, not all isoforms are always visible on the gel as they depend on the phosphorylation level and, furthermore, the phosphorylation status is also highly dynamic, and not every phosphorylation causes a similar strong shift, which complicates the picture. Nevertheless, the main point we want to make in the manuscript – and we think this is clearly evident from the figures – is that the level of bZIP63 phosphorylation changes in response to the different treatments and is furthermore depending on AKIN10.

*In the same subsection, I do not understand the logic of the following sentence:* “*Notably, hyper-phosphorylation of bZIP63 in the extended night was even stronger in the GY lines than in ox#3, probably due to lower bZIP63 expression.*” *Do the authors suggest that AKIN10 activity is limiting in terms of bZIP63 phosphorylation even if AKIN10 was supposed to be activated by extended night?*

The expression of bZIP63 in the genomic line is of course lower than in the ox lines. Therefore the ratio between bZIP63 and AKIN10 is also lower in these plants, which means that there is more AKIN10 per bZIP63 in the wt situation, which corresponds to the Phos-tag gel, where more bZIP63 is shifted to the hyper-phosphorylated form.

*With the limitations mentioned above concerning the necessary parameters of* bzip63 *mutant characterization, the lack of complementation by the triple S/A* bZIP63 *mutant gene is documented. As the mutant gene constructs were generated by PCR of 2+xkb genomic DNA fragments, the Materials and methods should mention that all constructs were verified by sequencing*.

The constructs were of course verified by sequencing. We have added this information in the Materials and methods section.

*In*
Figure 6*, it is rather surprising that the GY11 wt complementing line shows significantly higher induction of* ASN1 *and* DIN10*.*

The line is somewhat over-expressing (see Figure 7—figure supplement 1), which could explain the higher levels of *ASN1*.

*In the fold-induction curve and column diagram to the right, the standard error of Ws-2 is huge at 4h of EN, thus no significant difference between wt and* bzip63 *mutant can be claimed*.

We did the appropriate statistical tests and found significant differences. Looking at standard deviations alone is not sufficient to deduce significant differences. Also, the error bars are the standard deviations and not standard errors.

*In this figure part, either the curves shown to the left should include all four samples (Ws-2,* bzip63*, GY11 and GAY14) or all samples at the four time points should be presented as column diagrams*.

The two halves of the figure serve different purposes. We first did a time course (left) with wt and ko lines to determine the best time point for comparison of the gene expression. This was done with only two lines to reduce sample numbers. The 4h time point was chosen to test also the complementation lines.

*Given the data presented in*
Figure 6—figure supplement 1*, I wonder why did the authors use the low expressing GY11 line for comparison with GAY14*.

The lower expressing line was chosen, because bZIP63 expression is more similar in these lines (Figure 7—figure supplement 1).

*From*
Figure 6—figure supplement 1
*it is also apparent that the GAY4 line shows no expression of triple S/A mutant bZIP63-YFP, so only one bzip63 mutant transformed with the triple phosphor-site mutant of bZIP63 was analysed*.

The RT-qPCR in Figure 7—figure supplement 1 shows clearly that bZIP63 is expressed in this line. The levels are just lower than in the other lines, but in fact expression in this line is most similar to the wt line. The signal in the western blot is therefore also weaker and can only be seen in the extended night sample.

*Ehlert et al. reported that, when overexpressed in plant protoplasts, bZIP63 can heterodimerize with members of the S-group, and in those assays bZIP63 homodimers were barely detectable.*
[22]
*reported later that bZIP63 also interacts with the Arabidopsis response regulator ARR18 involved in cytokinin signaling, whereas alone (i.e. by interacting with S-group partners in heterodimers), bZIP63 functions as negative regulator of osmotic stress responses and seed germination*.

This was reported by Veerabagu et al. (2014, The Interaction of the *Arabidopsi*s Response Regulator ARR18 with bZIP63 Mediates the Regulation of PROLINE DEHYDROGENASE Expression).

*Several heterodimerization partners of bZIP63 are well characterized in terms of their transcription regulatory targets. Therefore, it is apparent that their regulatory functions might be modified by bZIP63, where heterodimerization capability of bZIP63 could be controlled post-translationally, e.g. by phosphorylation through SnRK1 and potentially other protein kinases. This idea should be introduced clearly, with some necessary references to the functions of suspected bZIP63 dimerization partners at the beginning of the manuscript and discussed in this term at the end. For example, the observed effect of bzip63 mutation on the levels of free amino acids could be related to the function of its heterodimerization partner bZIP1. So, probably citing a recent paper from Para et al. (Proc Natl Acad Sci USA. 111: 10371, 2014) may be helpful for discussion of metabolite changes observed in the bzip63 mutant*.

We agree, and have introduced that reference in the Introduction and indeed re-written the entire discussion of the metabolite data and highlighted the similarities and complementary aspects with bZIP1. We are confident that in the revised version the entire story is indeed much more stringent and fits nicely together also with respect to the other published data on bZIP1.

*Evidently, the bzip63 mutation could influence the function of bZIP1, i.e. if we assumed that the bZIP1 functions as heterodimer with bZIP63. It is therefore interesting that overexpression of AKIN10 in protoplasts has only a slight effect on heterodimerization of overexpressed Gal-AD-bZIP1 and Gal-BD-bZIP63. While this artificial protoplast dimerization assay in*
Figure 7
*indicates over 60-fold enhancement of Gal-AD-bZIP11+Gal-BD-bZIP63 dimerization, at the level of GAL-UAS4:GUS reporter, about 15-fold increase is observed, which is reduced to about 2-3-fold by triple mutation of the identified S29/S294/S300 bZIP63 phoshorylation sites. In case of*
Figure 7
*could not fully understand why the authors used two different (i.e. GAL-UAS4:GUS and GAL-UAS4:LUC) reporters in the dimerization and reporter activation assays. This correlative evidence between the dimerization and reporter activation assay should definitely be strengthened by performing similar dimerization assays with bZIP63 carrying the S29A/S294/S300 phospho-site replacements. This appears to be obligatory*.

We agree that mixing the LUC and GUS system is not optimal and have therefore repeated the P2H assays of bZIP63 with bZIP1, 11, and 63 as interaction partners in the GUS system (Figure 8).

While the general trend (addition of AKIN10 leads to stronger dimerization) remained the same, the magnitude of the change – especially for bZIP11 – is lower and fits better to the assays shown in Figure 8. It could be that the LUC system is more sensitive than the GUS system, which led to the discrepancies in the fold change between interaction in the absence and presence of AKIN10. Moreover, we have also noticed that in the previous version we have been mistaken by the strong effect on the interaction with bZIP11. As we discuss now, bZIP11 is a strong activator of transcription and bZIP1 and 63 not, which makes of course a big difference in these assays. This has been corrected and we did also include the requested point mutations, which confirm the strong effect of S29 on the dimerization.

*Although the Discussion takes care to announce that* “*to our knowledge this is the first time that phosphorylation outside the bZIP domain was shown to affect dimerization with different partners in a distinct way*”*, it is rather flat. Somewhere, it should mention that the current study did not answer the question of whether the observed phosphorylation sites would modify the cellular location (i.e. nuclear import) and stability of bZIP63, in addition to its heterodimerization, which should be anyhow confirmed by co-IP studies using native gene constructs*.

We did not observe an effect of S/A mutation on the localization of bZIP63 (also not with other serines than 29, 294, and 300). It is therefore not likely that phosphorylation alters the subcellular localization of bZIP63. We have stated this now clearly in the manuscript.

Concerning DNA binding, we tested whether S/A or S/D mutation of the N- and C-terminal serines would have an effect on binding to the C-box consensus motif as it was shown for S160, 164, and 168 in the DNA binding domain ([26], The role of phosphorylatable serine residues in the DNA-binding domain of *Arabidopsis* bZIP transcription factors). Again, we did not see any difference between wt and mutant bZP63 in these assays, which we also mention in the text.

*Unfortunately, the majority of the Discussion includes high-flying general statements, such as,* “*Here we show that bZIP63 plays an important role in the energy starvation response and metabolic regulation…”. If this is so, the reader will ask whether transcription of bZIP63 and its heterodimerization partners is controlled by AKIN10 and is altered in the akin10 mutant (i.e. a necessary but missing control)*.

We tested whether *bZIP63*, *bZIP1*, and *bZIP11 are* differentially expressed in wt and *akin10* plants by RT-qPCR (Figure 5—figure supplement 3). For *bZIP63* and *bZIP1,* no significant difference between the lines could be observed, neither in the light nor in the extended night. For *bZIP11* we observed a mild increase in *akin10* plants. However, the difference was less than 2-fold. It is therefore unlikely that AKIN10 regulates the transcription of these genes.

Can the enhanced senescence caused by bZIP63-ox compensated by glucose or sucrose supply?

To answer this question we did dark-induced senescence experiments in seedlings on plates with and without sugar. Seedlings were first grown on plates containing 0.5% sucrose and transferred to plates containing 0 or 2% glucose, grown for some days and put in the dark for 7 days. We found that sugar can indeed rescue the DIS phenotype in the ox to some extent (Figure 1).

*Is there any evidence for enhanced ROS production in the bZIP63-ox lines or change or red-ox regulation in the* bzip63 *mutant?*

Due to time constraints this question could not be addressed.

“*We measured strong differences in proline and asparagine levels in the bZIP63 ko and ox […] like altered carbon or nitrogen assimilation or protein turnover*” *(in the subsection “bZIP63 is an important metabolic regulator in the starvation response”)*. *So, how would bZIP63 control these mechanisms? Through controlling genes involved in N or C signaling or uptake?*

As already mentioned before, the Discussion has been rewritten to emphasize that these higher amino acids levels do indeed fit very nicely to starvation and have also been observed in dark-induced senescence. One explanation might be the increased turnover (degradation) of proteins and their role as alternative energy source, particularly under low-carbon conditions. This is now discussed in the manuscript.

*You say:* “*We found that the SnRK1 kinase AKIN10, which was proposed to be a central regulator of transcription in starvation response (*[3]*), is the major kinase responsible for the starvation-induced hyper-phosphorylation of bZIP63. Therefore, bZIP63 presents the first TF acting as direct target of SnRK1 in the transcriptional energy deprivation response*” *(in subsection “bZIP63 function is regulated by SnRK1-dependent phosphorylation”). Yes, but the hyper-phosphorylation of bZIP63 under extended night (called “starvation”, though it was 9 days of dark treatment) caused the appearance of only a weak band marked by asterisks on the images of Phos-tag gels, which represents the only evidence supporting this statement*.

This is a misunderstanding. Phos-tag gels were done after 6h of extended night not after 9 days of darkness. The extended night treatment represents a starvation condition for the plant.

Further, although the hyper-phosphorylation of bZIP63 is sometimes not so strong in the ox plants, it is very strong in the genomic complementation lines, which likely better reflect the phosphorylation state of bZIP63 in the wt. We think that the reduced phosphorylation of this band in *akin10* as well as *snrk1 VIGS* plants after 6h of extended night can be seen as good evidence that bZIP63 is a direct target of SnRK1.

In total, we provide evidence for this statement by:

(i) identifying AKIN10 as kinase for bZIP63 in an unbiased approach after affinity purification;

(ii) a missing band in in-gel kinase assays with extracts from the *akin10* mutant;

(iii) (various) direct interaction and phosphorylation assays;

(iv) a clearly reduced appearance of the hyper-phosphorylated form in the *akin10* mutant;

(v) even clearer results of this effect after VIGS of AKIN10 and AKIN11;

(vi) starvation (low-energy) related phenotypes of the *bzip63* mutants, which cannot be complemented by a bZIP63 version lacking the AKIN10 target sites;

(vii) a missing shift in response to EN treatments in the S29/S294/S300 mutant.

*In the same subsection, you state:* “*Reporter activation assays in protoplasts revealed that these (i.e., AKIN10 phosophorylation) sites are essential for AKIN10-dependent induction of* ASN1 *and* ProDH *by bZIP63*”*. This may be, but AKIN10 alone also activates these reporter genes, probably through other factors, which according to this paper are probably not the S-group bZIPs (i.e. as they appear not to be phosphorylated by AKIN10). It would be fair to mention this somewhere*.

The activation of the reporter genes by AKIN10 and a possible explanation had been mentioned in the text (“Transformation of AKIN10 alone also led to a weak induction of the reporters, which could be explained by the action of endogenous bZIPs”).

*Still in the same subsection, the authors say:* “*The relatively small difference in* ProDH *expression in plants, as compared to the protoplast assay, is probably due to redundancy within the C/S1 group bZIPs…*”*. Does this sentence predict that other C-group bZIPs could be SnRK1 substrates? Would it be not necessary to discuss potential conservation of observed AKIN10 sites in other bZIPs?*

No, this sentence does not imply that other C-group bZIPs are SnRK1 targets (although it is possible). We rather wanted to underline that several bZIPs have been found to regulate *ProDH* and therefore bZIP63 might not be necessarily required for *ProDH* expression.

*In the same part of the text, the authors state:* “*Our data provide compelling evidence that bZIP63, but none of the S1 group bZIPs, is a bona fide in vivo target of SnRK1 in low energy signaling.*” *This should clearly be demonstrated by identical Phos-tag gel patterns of bZIP63 isolated from sucrose-treated or* akin10 *mutant plants and that of the bZIP63 S29A/S294A/S300A protein obtained from* bzip63 *plus/minus sugar or the* akin10 *mutant*.

We do not agree with this statement. We believe that bZIP63 is phosphorylated by several different kinases, which makes the phosphorylation pattern more difficult to interpret.

As analysis of the *akin10* and *akin10/11* mutants showed that bZIP63 is phosphorylated by more than just AKIN10, as phosphorylation was not abolished completely in these mutants (Figure 5).

Furthermore, the S29/294/300A mutant has a much lower phosphorylation than bZIP63 in both of these mutants (Figure 7). This indicates that one or more of the sites can be phosphorylated by another kinase than AKIN10 and AKIN11.

Sugar treatment as well, could affect the activity of other kinases than AKIN10. It can therefore not be concluded that sugar treatment and *akin10* ko should have the same effect.

Taken together, we think that the phosphorylation pattern of bZIP63 is quite complex and depends on different factors (including kinases with potentially overlapping target sites) and cannot be simplified as the reviewer suggests.

Reviewer #2:

*Phos-tag analysis of in vitro phosphorylated bZIP63 samples allows the attribution of each shifted band to the phosphorylation of a specific Ser residue (*Figure 5—figure supplement 2*). In contrast, the same analyses of in vivo samples showed variable banding patterns that are not easy to reconcile (*Figure 2*,*
Figure 4*, Figure 4–figure supplement 3*, *and*
Figure 6*).*

As the comparison of different Phos-tag gels was difficult for several reviewers, we have aligned all Phos-tag gels in one figure (Figure 3—figure supplement 1) and introduced a numbering system to enable easier comparison of band patterns between figures (as has been mentioned before).

*This could be a result of phosphorylation of bZIP63 by other kinase(s) on S59, S102, S160, and S261, four sites that are not phosphorylatable by AKIN10. I would suggest the authors to determine whether the mutation of these four Ser residues to Ala will affect the function of bZIP63. If it does, the data would reveal that bZIP63 is regulated by multiple kinase pathways to carry out its function in the energy transition response. If it does not, the data would demonstrate that AKIN10-phosphorylation on S29, S294 and S300 is the major phosphorylation event that controls the activity of bZIP63*.

We do believe, as the reviewer suggested, that other kinases also phosphorylate bZIP63. However, as we focused on SnRK1 regulation of bZIP63 in this manuscript, we have not analyzed the functionality of these other sites. This would be a time-consuming process and is well beyond the focus of this manuscript. We have therefore not included additional data on the non-AKIN10 target sites.

For S160 it has been shown that it can potentially reduce DNA binding of bZIP63 when phosphorylated ([26], The role of phosphorylatable serine residues in the DNA-binding domain of *Arabidopsis* bZIP transcription factors), but for the other sites, no data on functionality are available.

*By performing Phos-tag analysis of bZIP63-S59A/S102A/S160A/S261A, the authors can provide more convincing results about the in vivo phosphorylation of bZIP63 by AKIN10 kinase*.

We feel that the relatively low gain of information for this line does not justify the large amount of work and time needed to create this line (∼ 1 year). It would be wrong to conclude that only AKIN10 would phosphorylate the remaining three sites in this mutant, which would again complicate the interpretation of the results. This is based on the fact, that AKIN10 ko leads to the vanishing of only one band, while mutation of the three sites leads to a strong reduction in phosphorylation. In fact, in the S29/294/300A mutant, only two bands remain (the non-phosphorylated form and a presumably single phosphorylated form). Therefore, the phosphorylation pattern of S29+294+300 would again be a combined effect from more than one kinase.

To further analyze the phosphorylation of bZIP63 in an *akin10/akin11* background, we added Phos-tag gels from VIGS ko of these two kinases (Figure 5). Also here, the only difference to wt plants was the vanishing of the hyper-phosphorylation in extended night. We therefore believe that the major phosphorylation of bZIP63 by AKIN10 happens in extended night and that maybe only one site is strongly phosphorylated by AKIN10 in vivo. This could be S29, as we showed with a new S294/300A mutant that S29 is phosphorylated in extended night, but not in the light (Figure 8).

Reviewer #3:

*1) Because SnRK and bZIP are well known proteins in plant biology, and because phosphorylation of bZIPs has been reported before, the true novelty in the manuscript is in my opinion the phosphorylation-dependent differential dimerization leading potentially to a change in transcriptional specificity plus activity toward distinct target genes (summarized in*
Figure 7
*– model). Data in*
Figure 7
*rely on a protoplast two-hybrid assay, allowing a transcriptional readout (equal protein amount in transfected cells has not been included here)*.

We have added a western blot to show the amounts of bZIP1, bZIP11, bZIP63, and AKIN10 in the P2H assay shown in Figure 8 (Figure 8—figure supplement 1).

To more prominently support the key message of the manuscript will have to address:

*i) Differential dimerization on protein level: conducting for example transient co-localization studies based on (multi-color) fluorescent proteins/BiFC (*[60]*), or using biochemical means to compare phosphorylation-dependent bZPI63/bZIP63 with bZIP63/bZIP11 formation, and in case even include bZIP S29D (or other) phospho-mimic constructs*.

We did pick up this suggestion by the reviewer, particularly as we noticed that the protoplast two-hybrid data do not allow a conclusion on preferred dimerization of different dimers because that assay is based on a transcriptional readout and leads to over-estimation of the effect on bZIP11 as mentioned before and also discussed in the revised manuscript. Therefore – as suggested by the reviewer – we have conducted a multi-color BiFC experiment to analyze the dimerization preferences of bZIP63 in a comparative in vivo assay (Figure 9).

As exemplary dimerization partners for bZIP63, we have chosen bZIP1 and bZIP11. It is important to note here that of course many other bZIPs will also affect dimerization under the ‘real’ conditions in the wt. All three dimerization partners (bZIP1, bZIP11, and bZIP63) were expressed under control of the 35S promoter from the same plasmid to ensure that all transformed cells express all three partners. A scheme of the BiFC experiment is shown in Figure 9.

Dimerization of bZIP63 with bZIP1 reconstructs the VENUS construct and leads to ‘yellow’ fluorescence. Dimerization of bZIP63 with bZIP11 reconstructs the CFP construct and leads to ‘cyan’ fluorescence. Homo-dimerization of bZIP63 or dimerization of any of the three partners with an endogenous (untagged) bZIP does not lead to a signal.

To see if there is a difference between bZIP63 dimerization with bZIP1 and 11, we co-transformed AKIN10 and compared the resulting signals obtained with the wt and the S29/294/300A version of bZIP63. This experiment allowed us to study the dimerization preference of phosphorylated and non-phosphorylated bZIP63. Accordingly, the bZIP63-11 hetero-dimer will generate a blue (cyan) signal and the bZIP63-1 hetero-dimer a yellow signal, whereas the bZIP63 homo-dimer cannot be detected. A quantification of the ratio of the VENUS/CFP signal obtained from more than 100 nuclei did clearly show that the co-transformation of AKIN10 did shift the signal towards the VENUS emission, thus showing that AKIN10 does preferentially trigger the formation of bZIP63-1 hetero-dimers in a competitive in vivo assay.

At first glance, this seems to stand in contrast to the data obtained in the protoplast two-hybrid assays, but as already pointed out, these assays cannot be compared. As we have been mistaken by not realizing this problem before, we had also to revise our model accordingly (Figure 9). However, we would like to emphasize that the multicolor BiFC experiment is indeed a first in vivo evidence for the proposed phosphorylation triggered shift in dimerization, mediated by AKIN10. We are well aware that other factors (protein stability, expression levels etc.) will also strongly influence this process, but in our view there is currently no experimental possibility to address these points as well in one assay.

*ii) Show differential target gene specificity:*
Figure 7
*contains as marker for transcriptional activity a gene which is target to all three bZIPs (1, 11, 63). Even more convincing will be phosphorylation/heterodimerization-dependent activity toward a distinct bZIP11- or bZIP63-controlled gene, respectively*.

Yes, we agree, this would indeed be a nice experiment and we would like to do that. Unfortunately, we do not know yet any target gene, which would be specific for only one of the different bZIP dimers. Microarray data are available for bZIP1 and bZIP11, but not for bZIP63. Moreover, also from these data, such genes cannot be deduced as the other different dimerization partners are still present. Maybe ChIP experiments would be a solution here, but that would clearly be beyond the scope of this study.

*2) The primary identification of* “*in vivo*” *phosphorylation sites is conducted on bZIP-GFP overexpressing lines on material harvested in both light and prolonged dark. Therefore, basic constitutive bZIP phosphorylation (and the involvement of other kinases than AKIN10) cannot be excluded. Ideally, one may ask for the degree of conditional in vivo phosphorylation (in particular at S29 of endogenous bZIP) performing MRM/SRM phosphoproteomics*.

To address this question we generated a genomic complementation line expressing the S294/300A version of bZIP63 in the ko background. In this line, S29 can be phosphorylated, but none of the other potential AKIN10 target sites. Due to the short time we had for the revision, this plant line is still heterozygous and cannot be used for phenotypic analysis at this point. However, we could use the line for Phos-tag gels and we here we saw indeed a striking effect: Compared to the triple mutant, that did not show any change in the phosphorylation pattern in response to starvation, the S294/S300 double mutant did still show a strong shift in the extended night. As S29 is the only AKIN10 target site left in this version, it must indeed be S29 that is phosphorylated by AKIN10 under starvation conditions. We think therefore we did add really significant novel information, and more is simply not possible in the few months we had for the revision.